# A Survey on Generative Modeling with Limited Data, Few Shots, and Zero Shot

**Milad Abdollahzadeh**[*§]                                     *milad@betterdata.ai*
*Singapore University of Technology and Design*
*Betterdata AI, Singapore*

**Guimeng Liu**[*]                                     *guimeng_liu@mymail.sutd.edu.sg*
*Singapore University of Technology and Design*

**Touba Malekzadeh**[†]                                     *touba_malekzadeh@sutd.edu.sg*
*Singapore University of Technology and Design*

**Christopher T.H Teo**[†§]                                     *christopher.teo@sap.com*
*Singapore University of Technology and Design*
*SAP, Singapore*

**Keshigeyan Chandrasegaran**[†§]                                     *keshik@cs.stanford.edu*
*Singapore University of Technology and Design*
*Stanford University*

**Ngai-Man Cheung**[‡]                                     *ngaiman_cheung@sutd.edu.sg*
*Singapore University of Technology and Design*

**Reviewed on OpenReview:** *https://openreview.net/forum?id=u7GTHazuRp*

## Abstract

Generative modeling in machine learning aims to synthesize new data samples that are statistically similar to those observed during training. While conventional generative models such as GANs and diffusion models typically assume access to large and diverse datasets, many real-world applications (*e.g.* in medicine, satellite imaging, and artistic domains) operate under limited data availability and strict constraints. In this survey, we examine **Generative Modeling under Data Constraint (GM-DC)**, which includes limited-data, few-shot, and zero-shot settings. We present a unified perspective on the key challenges in GM-DC, including overfitting, frequency bias, and incompatible knowledge transfer, and discuss how these issues impact model performance. To systematically analyze this growing field, we introduce two novel taxonomies: one categorizing GM-DC tasks (*e.g.* unconditional vs. conditional generation, cross-domain adaptation, and subject-driven modeling), and another organizing methodological approaches (*e.g.* transfer learning, data augmentation, meta-learning, and frequency-aware modeling). Our study reviews over 230 papers, offering a comprehensive view across generative model types and constraint scenarios. We further analyze task-approach-method interactions using a Sankey

---

[*]Equal first-author contribution
[†]Equal second-author contribution
[‡]Corresponding author
[§]Work was done while the authors were at SUTD.

diagram and highlight promising directions for future work, including adaptation of foundation models, holistic evaluation frameworks, and data-centric strategies for sample selection. This survey provides a timely and practical roadmap for researchers and practitioners aiming to advance generative modeling under limited data. Project website: https://sutd-visual-computing-group.github.io/gmdc-survey/.

# 1 Introduction

Generative modeling is a field of machine learning that focuses on learning the underlying distribution of the training samples, enabling the generation of new samples that exhibit similar statistical properties to the training data. Generative modeling has profound impacts in various fields including computer vision (Ramesh et al., 2022; Karras et al., 2020b; Brock et al., 2019; Rombach et al., 2022; Peebles & Xie, 2023; Guo et al., 2025), natural language processing (Yu et al., 2017; Gulrajani et al., 2017; Van Den Oord et al., 2017; Gat et al., 2024; Nie et al., 2025) and data engineering (Antoniou et al., 2017; Karras et al., 2020a; Tran et al., 2021; Wang et al., 2023d; Hou et al., 2024). Over the years, significant advancements have been made in generative modeling. Innovative approaches such as Generative Adversarial Networks (GANs) (Goodfellow et al., 2014; Karras et al., 2019; Brock et al., 2019; Arjovsky et al., 2017; Kang et al., 2023; Huang et al., 2024b), Variational Autoencoders (VAEs) (Kingma & Welling, 2014; Vahdat & Kautz, 2020; Van Den Oord et al., 2017), and Diffusion Models (DMs) (Rombach et al., 2022; Song et al., 2020; Dhariwal & Nichol, 2021; Nichol & Dhariwal, 2021; Peebles & Xie, 2023; Esser et al., 2024; Chandrasegaran et al., 2025) have played a pivotal role in enhancing the quality and diversity of generated samples. The advancements in generative modeling have fueled the recent disruption in generative AI, unlocking new possibilities in various applications such as image synthesis (Nellis, 2024; Metz, 2025a), text generation (Jamali, 2025; Metz, 2025b), music composition (Bakare, 2025; Lamba & Sophia, 2025), genomics (Schiff et al., 2024), and more (Han et al., 2024a; Chen et al., 2024b). The ability to generate realistic and diverse samples has opened doors to creative applications and novel solutions (Shearing, 2025; Tong & Hu, 2025).

Research on generative modeling has been mainly focusing on setups with sizeable training datasets. Style-GAN (Karras et al., 2019) learns to generate realistic and diverse face images using Flickr-Faces-HQ (FFHQ), a high-quality dataset of 70k human face images collected from the photo-sharing website Flickr. More recent text-to-image generative models and diffusion models (DMs) are trained on millions of image-text pairs, e.g. Stable Diffusion (Rombach et al., 2022) is trained on LAION-400M with 400 million samples (Schuhmann et al., 2021). However, in many domains (*e.g.* medical), the collection of data samples is challenging and expensive.

**In this paper**, we survey Generative Modeling under Data Constraint (GM-DC). This research area is important for many domains/ applications where challenges and constraints in data collection exist. We conduct a thorough literature review on learning generative models under limited data (given 50-5000 training samples), few shots (1-50 samples), and zero shot (no samples). *Our survey is the first to provide a comprehensive overview and detailed analysis of all types of generative models, tasks, and approaches studied in GM-DC, offering an accessible guide on the research landscape* (Fig. 1). We cover the essential backgrounds, provide detailed analysis of unique challenges of GM-DC, discuss current trends, and present the latest advancements in GM-DC.

**Our Contributions:** i) Trends, technical evolution, and statistics of GM-DC (Fig. 3; Fig. 4; Sec. 7.1); ii) New insights on GM-DC challenges (Sec. 4.2); iii) Two novel and detailed taxonomies, one on GM-DC tasks (Sec. 4.1) and another on GM-DC approaches (Sec. 5); iv) A novel Sankey diagram to visualize the research landscape and relationship between GM-DC tasks, approaches, and methods (Fig. 1); v) An organized summary of individual GM-DC works (Sec. 5), critical analysis, and empirical comparison (Sec. 6), vi) Discussion of future directions (Sec. 7.3). We further provide a project website with an interactive diagram to visualize GM-DC landscape. Our survey aims to be an accessible guide to provide fresh perspectives on the current research landscape, organized pointers to comprehensive literature, and insightful trends on the latest advances of GM-DC.

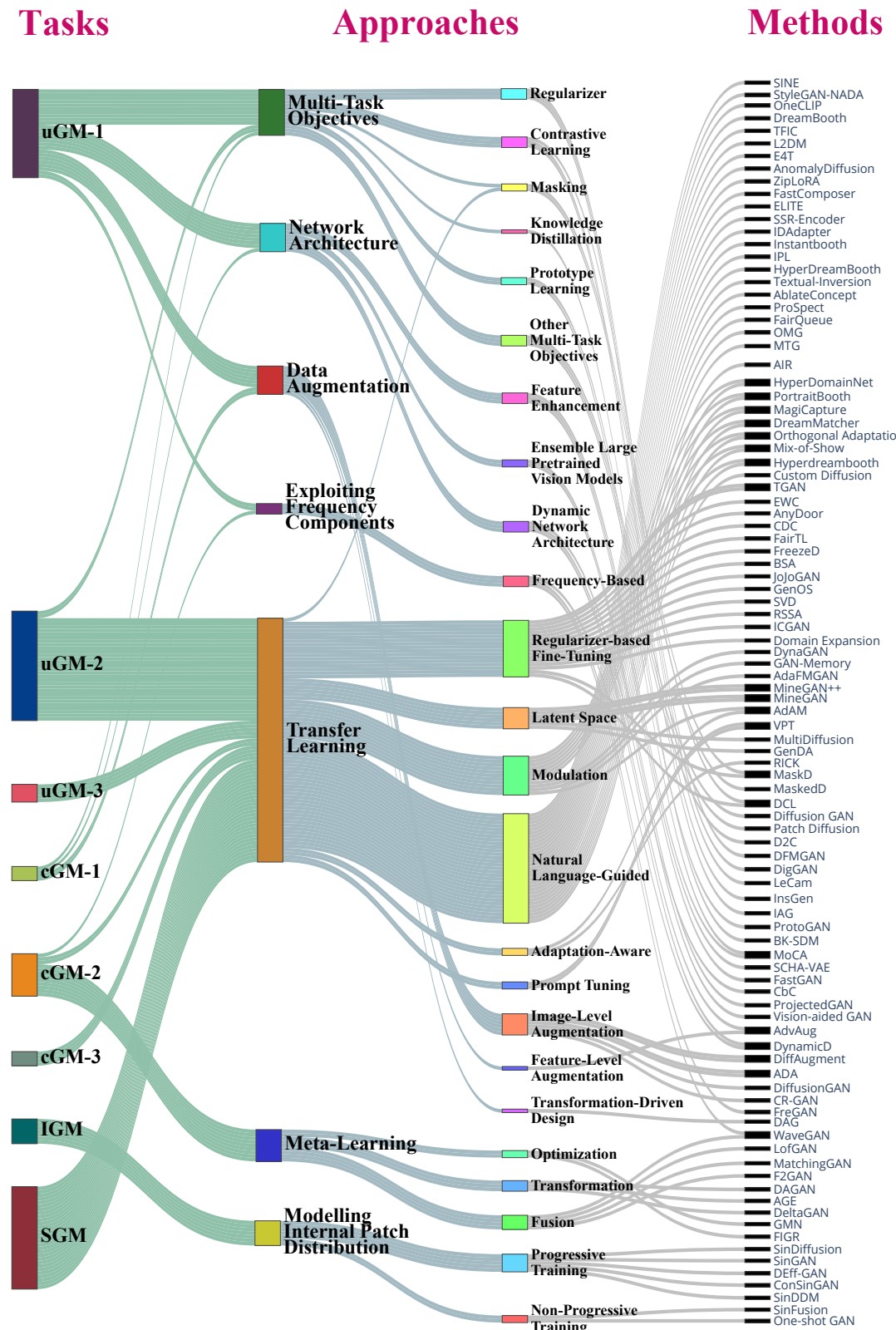

Figure 1: **Research Landscape of GM-DC.** The figure shows the interaction between GM-DC tasks and approaches (main and sub categories), and representative GM-DC methods. Tasks are defined in our proposed taxonomy in Tab. 2, and approaches in our proposed taxonomy in Tab. 3. An interactive version of this diagram is available at our project website. Best viewed in color and with zoom.

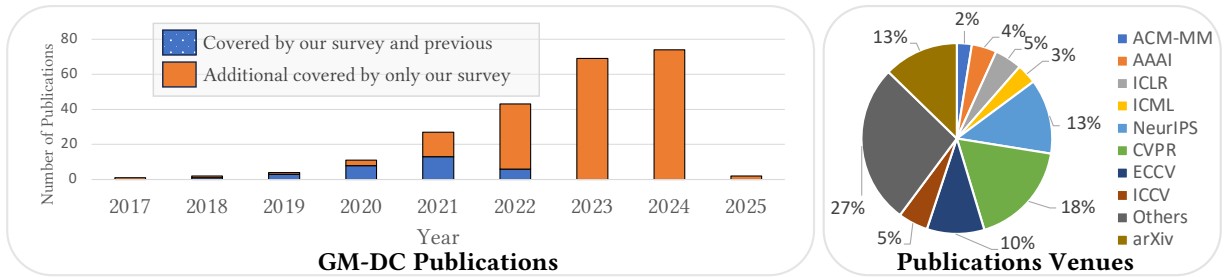

Figure 2: Overall publications statistics in GM-DC. **GM-DC Publications (Left):** GM-DC publication trends indicate rising interest in this area. We remark that the previous survey (Li et al., 2022d) only covers ∼13% of publications discussed in our survey. **Publication Venues (Right):** The distribution of publications in major machine learning and computer vision venues, other venues, and arXiv. Best viewed in color.

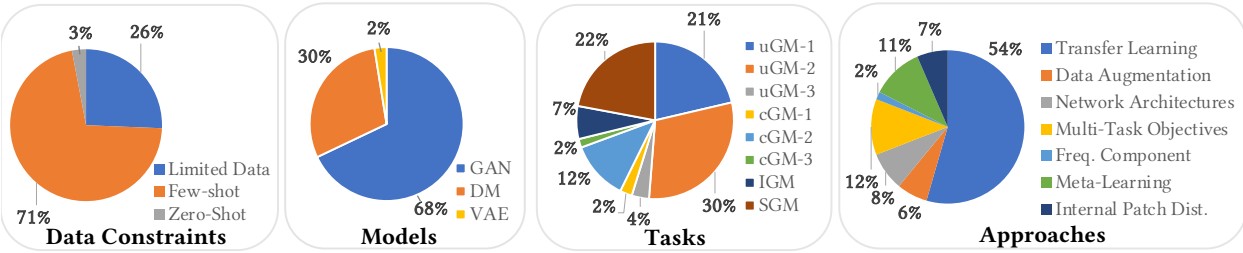

Figure 3: Analysis of publications in GM-DC. **Data Constraints:** Different types of data constraints studied in GM-DC. See Sec. 3 for more details on setups. **Models:** Different types of models are studied including Generative Adversarial Network (GAN), Diffusion Model (DM), and Variational Auto-Encoder (VAE). **Tasks:** Different GM-DC tasks that are studied; See Sec. 4.1, and Tab. 2 for details on task definitions in our proposed task taxonomy. **Approaches:** Different approaches that are applied for addressing GM-DC; More details on our proposed taxonomy of approaches can be found in Sec. 5 and Tab. 3. Best viewed in color.

## 1.1 Paper Selection and Search Strategy

**Search Strategy.** Our search strategy is based on an extensive list of keywords related to GM-DC. We began with a set of seed keywords derived from well-known papers in the field and selected several representative works. We then carefully examined these papers and their related work, expanding our initial list to form the main set of keywords used to search for relevant research papers for the GM-DC survey. The final list of 17 keywords is as follows: *Transfer learning for generative models*; *Transfer learning in GANs/DMs* (2 keywords); *Fine-tuning generative models*; *Fine-tuning GANs/DMs/VAEs* (3 keywords); *Generative model adaptation*; *Adaptation of GANs/DMs/VAEs* (4 keywords); *Few-shot image generation*; *Few-shot adaptation of generative models*; *Data-efficient generative modeling*; *Training GANs/DMs with limited data* (2 keywords). To ensure comprehensive coverage, we searched for these keywords across seven major repositories commonly used for machine learning and computer vision research: Google Scholar, OpenReview, CVF Open Access, IEEE Xplore Digital Library, ACM Digital Library, ScienceDirect (Elsevier), and SpringerLink.

**Study Selection Criteria.** Among the collected papers, we applied three main criteria for inclusion, focusing on the problem setup, modality, and task type:

- Problem Setup: We examined all experimental results (both qualitative and quantitative) in each paper to ensure that at least one of them satisfies a data-constrained setup ($0 \sim 5000$ available samples), i.e., the proposed approach can operate under such conditions.

- Modality: Our survey focuses exclusively on the image modality; therefore, we discarded studies dealing with other modalities such as sculpture, 3D mesh, or point cloud data.

- Task Type: We strictly focused on the image generation task and excluded papers addressing other tasks (e.g., image editing).[1]

**Second Round of Search.** After the initial selection, we conducted a second round of review by carefully examining the related work and experimental setups of the selected papers to ensure that all relevant studies were included. In this step, we also added some works that did not explicitly mention the keywords in their title or abstract but nonetheless satisfied all inclusion criteria.

**The rest of the paper is organized as follows.** In Sec. 2 we discuss related work. In Sec. 3 we provide the necessary background. In Sec. 4, we discuss GM-DC tasks and unique challenges. In Sec. 5, we analyze GM-DC approaches and methods. In Sec. 6, we summarize, for each class of approaches, the key factors contributing to their success or failure, their fundamental limitations, and the deeper insights on their design principles. We further provide empirical comparison. In Sec. 7, we discuss open research problems and future directions. Sec. 8 concludes the survey.

## 2 Related Work

**Discriminative Modeling with Limited Data.** A conceptually similar task to GM-DC is discriminative learning under data constraints. Approaches in this research direction aim to learn classification, regression, or even reinforcement learning models using limited and sometimes few-shot data, often through techniques such as meta-learning (Finn et al., 2017; Vinyals et al., 2016; Snell et al., 2017) or knowledge transfer from a powerful, pretrained model (Radford et al., 2021; Sun et al., 2019; Tan et al., 2018). This line of research is commonly referred to as few-shot learning for simplicity, and there are several surveys that cover the key concepts and methodologies in this area in considerable detail (Wang et al., 2020a; Hospedales et al., 2021; Gharoun et al., 2024; Zeng & Xiao, 2024; Song et al., 2023). However, despite some conceptual overlap in addressing the challenge of learning with limited data, these works focus on discriminative tasks, which are fundamentally different from the generative learning tasks discussed in this survey. Consequently, the existing surveys in this domain do not cover the concepts, approaches, or taxonomies that are the focus of this work.

**Generative Modeling.** A broad line of research in machine learning has focused on developing powerful generative models that can synthesize high-quality and diverse data samples. Key paradigms include variational autoencoders (Kingma & Welling, 2014; Vahdat & Kautz, 2020), generative adversarial models (Goodfellow et al., 2014; Karras et al., 2019; Kang et al., 2023), flow-based Models (Ho et al., 2019; Gat et al., 2024), diffusion models (Song et al., 2020; Rombach et al., 2022; Esser et al., 2024). Several surveys provide comprehensive overviews of these approaches and their advancements (Pouyanfar et al., 2018; Jabbar et al., 2021; De Souza et al., 2023; Hu et al., 2023b; Bie et al., 2024; Li et al., 2025a). These works highlight the remarkable progress in fidelity, diversity, and controllability. However, they typically assume access to abundant training data, and their performance can degrade significantly when trained with limited samples, manifesting in problems such as overfitting, mode collapse, or reduced generalization. Consequently, while these surveys form the foundation of modern generative modeling, they do not directly address the unique challenges of learning under data-constrained settings, which are the primary focus of this work.

**Generative Modeling with Limited Data.** Previous survey on GM-DC (Li et al., 2022d) has focused on only a subset of GM-DC papers, studying only works with GANs as a generative model and a subset of technical tasks/ approaches. Our survey differentiates itself from Li et al. (2022d) in: i) Scope - Our survey is the first to cover all types of generative models and all GM-DC tasks and approaches (Fig. 3); ii) Scale - Our study includes 233 papers and covers broad GM-DC works, while previous survey (Li et al., 2022d) covers only ≈13% of works discussed in our survey (Fig. 2); iii) Timeliness - Our survey collects and surveys the most up-to-date papers in GM-DC; iv) Detailedness - Our paper includes detailed visualizations (Sankey diagram, charts) and tables to highlight interactions and important attributes of GM-DC literature; v) Technical evolution analysis - Our paper analyzes the evolution of GM-DC tasks and approaches, providing new

---

[1] Note that some recent works, such as DreamBooth, can handle both image generation and image editing tasks. Since these methods support image generation and not only editing, they are included in our study.

perspectives on recent advances; vi) Horizon analysis - Our paper discusses distinctive obstacles encountered in GM-DC and identifies avenues for future research.

# 3 Background

In this section, we first define 'domain' and 'generative modeling', then we discuss common approaches of generative modeling and data constraints studied in GM-DC.

**Domain.** In this survey, a *domain* consists of two components: i) a sample space $\mathcal{X}$, and ii) a marginal probability distribution $P_{data}$, which models the probability of samples from $\mathcal{X}$ (Pan & Yang, 2009). This is written as $\mathcal{D} = \{\mathcal{X}, P_{data}\}$, and $x \sim P_{data} \in \mathcal{X}$ denoting a sample in this space. An example of a domain is the domain of image of human faces: $\mathcal{D}^h = \{\mathcal{X}, P_{data}^h\}$. Here $\mathcal{X}$ is the sample space of images, and $P_{data}^h$ is the probability distribution of human faces.

**Generative Modeling.** Given a set of training sample $x$ of a domain $\mathcal{D} = \{\mathcal{X}, P_{data}\}$, i.e., with an underlying probability distribution $P_{data}$, generative modeling aims to learn to capture $P_{data}$ —sometimes also denoted as $P(x)$ in literature. The result of generative modeling is a *generative model $G$* encoding a probability distribution $P_{model}$. The learning objective is to have $P_{model}$ similar to $P_{data}$ statistically. After the training, $G$ can generate samples following $P_{model}$. For example, generative modeling with a training set of human face images aims to learn to capture $P_{data}^h$, thereby the resulting $G^h$ can generate human face images that are statistically similar to samples from $P_{data}^h$. We also refer to the domain of training samples as *target domain*.

**Conditional vs Unconditional Sample Generation.** After learning the underlying distribution of data $P_{data}$, the generative model can generate new samples by sampling from the learned distribution $P_{model}$. Typically, generation starts with sampling a random vector $z$ —also called latent code— as input. Then, this input is passed into the generative model $G$ to transform the latent code into a new sample $G(z) \sim P_{model}$. Ideally, a good generator is able to capture the characteristics, quality and diversity of the training dataset, *i.e.* , $P_{model}$ is similar to $P_{data}$ statistically. If an additional condition $c$ (like a class label or attribute) is used alongside with the latent code to steer the sample generation towards $c$, the sample generation is called conditional generation: $G(z, c)$.

## 3.1 Approaches for Generative Modeling

Earlier works on generative models study Gaussian Mixture Models (Reynolds et al., 2009), Hidden Markov Models (Phung et al., 2005), Latent Dirichlet Allocation (Chauhan & Shah, 2021) and Boltzmann Machines (Ackley et al., 1985). With the introduction of deep neural networks, recent works study powerful generative models, particularly those for image generation, which most GM-DC works focus on.

**Variational Auto Encoders (VAE)** (Kingma & Welling, 2014). VAE is a variant of Auto-Encoder (AE) (Zhai et al., 2018), where both consist of the encoder and decoder networks. AE focuses on dimensional reduction. The encoder in AE learns to map an input $x$ into a latent (compressed) representation, $z = E(x)$. Then, the decoder aims to reconstruct the image from that latent representation, $\hat{x} = D(z)$. Model parameters are optimized with the following reconstruction loss:

$$\mathcal{L}_{rec} = ||x - D(z)||_2 \tag{1}$$

AEs are notorious for latent space irregularity making them improper for sample generation (Kingma et al., 2019). VAE aims to address this problem by enforcing $E$ to return a normal distribution over latent space. Assuming a distribution $z \sim \mathcal{N}(\mu, \sigma^2)$ for latent space, this is done by adding the KL-divergence term to the loss function:

$$\mathcal{L} = ||x - D(z)||_2 + KL(\mathcal{N}(\mu, \sigma^2), \mathcal{N}(0, I)) \tag{2}$$

Due to the challenges of direct maximization of the likelihood in pixel space, Vector-Quantized VAE (VQ-VAE) proposes *tokenization* where a codebook $\mathbf{e}_k$, $k \in 1, \ldots, K$ is used to quantize the embeddings $E(x)$ into visual tokens (indices), acting like a lookup table. In addition, a latent prior of the visual tokens is predicted (usually using a transformer), and the decoder is modified to map the visual tokens into the image space.

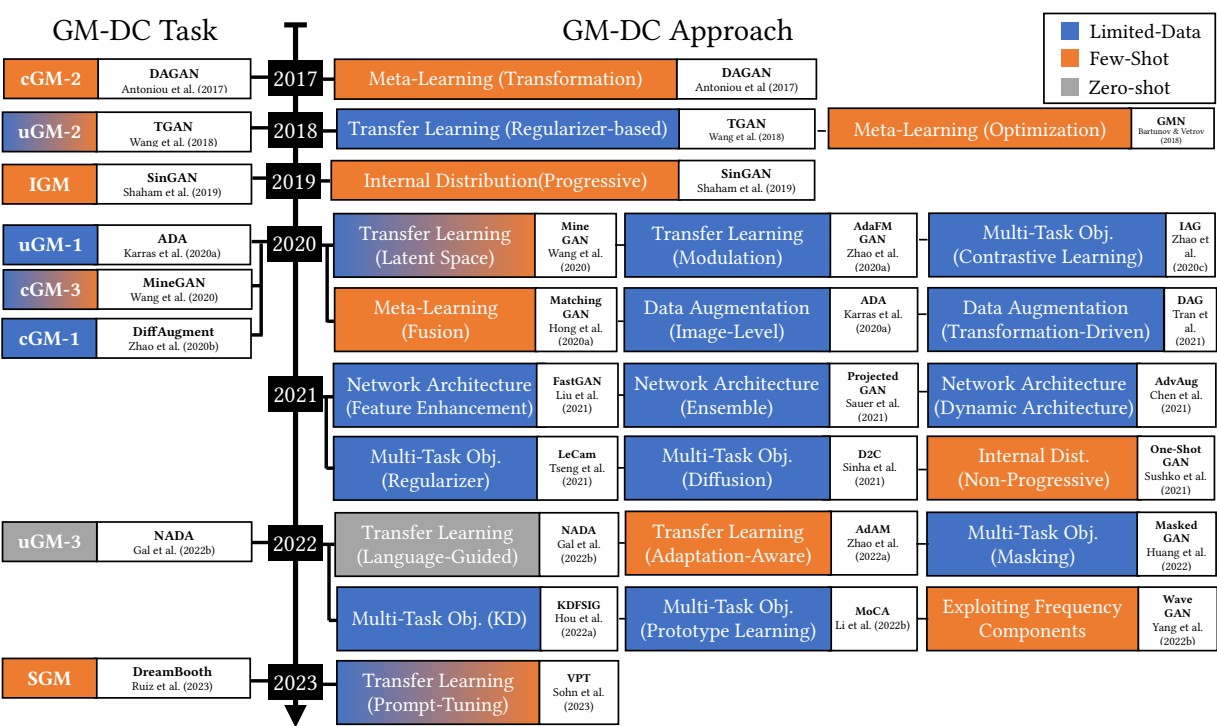

Figure 4: Illustration of the **timeline** when a GM-DC task/approach was introduced based on our proposed taxonomies: task taxonomy (details in Sec. 4.1, and Tab. 2), and approach taxonomy (details in Sec. 5, and Tab. 3). Best viewed in color.

**Generative Adversarial Models (GAN)** (Goodfellow et al., 2014). GAN applies an adversarial approach to learn the distribution of data $P_{data}$. It consists of a generator $G$ and a discriminator $D$ playing a min-max game. Specifically, given the latent code $z$, the $G$ learns to generate the images $G(z)$, $z \sim P_z$, where $P_z$ is usually a Gaussian distribution. Then, $D$ learns to distinguish the real images $x \sim P_{data}$ from the generated ones $G(z) \sim P_{model}$. The $D$ and $G$ are optimized by respectively maximizing and minimizing the following value function:

$$\mathcal{V}(D, G) = \mathbb{E}_{x \sim p_{data}}[\log D(x)] + \mathbb{E}_{z \sim p_z}[\log(1 - D(G(z)))] \tag{3}$$

**Flow-based Models** (Ho et al., 2019). The flow-based model includes a series of invertible yet differentiable functions $f$, between latent distribution $P_z$, and data distribution $P_{data}$. The following log-likelihood function is maximized to train $f(.|\theta)$:

$$\max_{\theta} \sum_{i=1}^{K} \log P_z(f(x^{(i)}|\theta)) + \log|\det Df(x^{(i)}|\theta)| \tag{4}$$

For ease of discussion, we simplify the model as a single flow and denote the training samples with $\{x^{(i)}\}_{i=1}^{K}$, and the Jacobian of $f(x)$ as $Df(x)$. We remark that, unlike VAEs that estimate the lower bounds of the log-likelihood, flow-based models evaluate the exact log-likelihood in their loss function.

**Diffusion Models (DM)** (Ho et al., 2020). DM leverages the concept of the diffusion process from stochastic calculus and consists of forward diffusion and reverse diffusion processes. In the forward diffusion process, based on the foundations of Markov chains, the noise $\epsilon \sim \mathcal{N}(0, I)$ is iteratively added to data samples until it approaches an isotropic Gaussian distribution. Then, in the backward process, the DM learns to denoise the noisy vector $x_T$ and reconstruct the data samples $x_0$. This is done by learning the noise estimation model $\epsilon_\theta$ with minimizing the following loss function (Ho et al., 2020):

$$\mathcal{L} = \mathbb{E}_{t, x_0, \epsilon}[||\epsilon - \epsilon_\theta(\sqrt{\bar{\alpha}_t}x_0 + \sqrt{1 - \bar{\alpha}_t}\epsilon, t)||_2] \tag{5}$$

Table 1: List of common datasets used in GM-DC works. Number of samples (# Samples) refers to the sample size of the entire dataset. In GM-DC experiments, usually, only a subset of the dataset is used. We remark that ○/●denotes the absence/presence of the dataset under the data constraint settings: **LD**: Limited-Data, **FS**: Few-Shot and **ZS**: Zero-Shot, and Labels indicate if training labels are available (but not necessarily used).

| Dataset | Description | # Samples | Resolution | LD | FS | ZS | Labels |
|---|---|---|---|---|---|---|---|
| Flickr-Faces-HQ (FFHQ) (Karras et al., 2019) | Images with human faces, containing variation in terms of age, ethnicity, and image background. | 70K | 1024×1024 | ● | ● | ● | ○ |
| Large-scale Scene Understanding (LSUN) (Yu et al., 2015) | Images with large-scale scene containing 10 scene and 20 object categories. | 3M | 256×256 | ● | ● | ○ | ● |
| MetFace (Karras et al., 2020a) | Images depicting paintings, drawings, and statues of human faces | 1336 | 1024×1024 | ● | ● | ○ | ○ |
| BreCaHAD (Aksac et al., 2019) | Images of breast cancer histopathology. | 162 | 1360×1024 | ● | ○ | ○ | ○ |
| Animal FacesHQ (AFHQ) (Choi et al., 2020) | Images of animal faces in the domains of cat, dog, and wildlife. | 15K | 512×512 | ● | ● | ○ | ○ |
| CIFAR-10 (Krizhevsky et al., 2009) | Images including objects and animals. | 60K | 32×32 | ● | ○ | ○ | ● |
| CIFAR-100 (Krizhevsky et al., 2009) | A dataset similar to CIFAR-10, but with 100 classes | 60K | 32×32 | ● | ○ | ○ | ● |
| 100-shot Obama/ Gumpy Cat/Panda (Zhao et al., 2020b) | Colored images of Obama/Gumpy Cat/Panda | 100 | 256×256 | ● | ○ | ○ | ○ |
| Sketches (Wang & Tang, 2008) | Face sketches in frontal pose, normal lighting, and neutral expressions | 606 | 256×256 | ○ | ● | ○ | ○ |
| Sunglasses (Ojha et al., 2021) | Images of human faces wearing sunglasses. | 2700 | 256×256 | ○ | ● | ○ | ○ |
| Babies (Ojha et al., 2021) | Images of baby faces. | 2500 | 256×256 | ○ | ● | ○ | ○ |
| Artistic-Faces (Yaniv et al., 2019) | Images containing 160 artistic portraits of 16 different artists. | 160 | 256×256 | ○ | ● | ○ | ○ |
| Haunted houses (Yu et al., 2015) | Images of haunted houses | 1K | 256×256 | ○ | ● | ○ | ○ |
| Wrecked cars (Yu et al., 2015) | Images of wrecked cars | 1K | 256×256 | ○ | ● | ○ | ○ |

Then, during the generation process, DM first samples a noise $x_T \sim \mathcal{N}(0, I)$, and utilizes the learned noise function $\epsilon_\theta$ to iteratively apply the following denoising process (Ho et al., 2020):

$$x_{t-1} = \frac{1}{\sqrt{\alpha_t}}(x_t - \frac{1-\alpha_t}{\sqrt{1-\bar{\alpha}}}\epsilon_\theta(x_t, t)) + \sqrt{\beta_t}\epsilon, \quad t \in [0, T] \tag{6}$$

Here, $x_t$ is the generated sample at step $T - t$, $\beta_t$ is variance scheduler, $\alpha_t = 1 - \beta_t$ and $\bar{\alpha}_t = \prod_{s=1}^{t} \alpha_s$.

**Remark.** We remark that among discussed models, only GANs, DMs, and VAEs are adopted in the context of GM-DC.

## 3.2 Data Constraints and Commonly Used Datasets

In GM-DC, three data constraints have been considered in most works: (i) *Limited data (LD)*, when 50 to 5,000 training samples are given; (ii) *Few-Shot (FS)*, when 1 to 50 training samples are given; (iii) *Zero-Shot (ZS)*, when no training samples are given. These ranges reflect the experimental setups repeatedly adopted in the literature. For example, representative LD works such as StyleGAN2-ADA and FastGAN report results on datasets with roughly $10^2$–$10^3$ samples (e.g., 60 to 1.9k images, up to 5k images), and later studies frequently follow similar scales. Works in the FS setting (e.g., EWC, CDC, AdAM, JoJoGAN) explicitly assume only 1–50 samples from the target domain. More recent ZS methods (e.g., NADA, IPL, AIR) assume no target-domain samples and instead rely on external guidance. We adopt these ranges to align with these widely used regimes. Training under such constraints often leads to issues such as overfitting and mode collapse. Tab. 1 lists the most common datasets used in GM-DC with related details.

Table 2: **Our proposed taxonomy for tasks in GM-DC**. For each task, we extract their key characteristics. [Attributes] **C**: Conditional generation, **P**: Pre-trained generator given, **I**: Images (as input), **TP**: Text-Prompt (as input), **X**: X(Cross)-domain adaptation; [Data Constraint] **LD**: Limited-Data, **FS**: Few-Shot, **ZS**: Zero-Shot. ○/● denotes the absence/presence, respectively. Best viewed in color.

| Task | Attributes | | | | | Data Constraint | | | Task Illustration |
|---|---|---|---|---|---|---|---|---|---|
| | **C** | **P** | **I** | **TP** | **X** | **LD** | **FS** | **ZS** | |

| | C | P | I | TP | X | LD | FS | ZS |
|---|---|---|---|---|---|---|---|---|
| **uGM-1** | ○ | ○ | ● | ○ | ○ | ● | ○ | ○ |

**uGM-1** — **Description:** Given $K$ samples from a domain $\mathcal{D}$, learn to generate diverse and high-quality samples from $\mathcal{D}$. **Example:** ADA (Karras et al., 2020a) learns a StyleGAN2 using 1k images from AFHQ-Dog

**uGM-2** — C○ P● I● TP○ X● | LD● FS● ZS○. **Description:** Given a pre-trained generator on a source domain $\mathcal{D}_s$ and $K$ samples from a target domain $\mathcal{D}_t$, learn to generate diverse and high-quality samples from $\mathcal{D}_t$. **Example:** CDC (Ojha et al., 2021) adapts a pre-trained GAN on FFHQ to Sketches using 10 samples

**uGM-3** — C○ P● I○ TP● X● | LD○ FS○ ZS●. **Description:** Given a pre-trained generator on a source domain $\mathcal{D}_s$ and a text prompt describing a target domain $\mathcal{D}_t$, learn to generate diverse and high-quality samples from $\mathcal{D}_t$. **Example:** NADA (Gal et al., 2022b) adapts pre-trained GAN on FFHQ to the painting domain using *'Fernando Botero Painting'* as input

**cGM-1** — C● P○ I● TP○ X○ | LD● FS○ ZS○. **Description:** Given $K$ samples with class labels from a domain $\mathcal{D}$, learn to generate diverse and high-quality samples conditioning on the class labels from $\mathcal{D}$. **Example:** CbC (Shahbazi et al., 2022) trains conditional generator on 20 classes of ImageNet Carnivores using 100 images per class

**cGM-2** — C● P● I● TP○ X○ | LD○ FS● ZS○. **Description:** Given a pre-trained generator on the seen classes $C_{seen}$ of a domain $\mathcal{D}$ and $K$ samples with class labels from unseen classes $C_{unseen}$ of $\mathcal{D}$, learn to generate diverse and high-quality samples conditioning on the class labels for $C_{unseen}$ from $\mathcal{D}$. **Example:** LoFGAN (Gu et al., 2021) learns from 85 classes of Flowers to generate images for an unseen class with only 3 samples

**cGM-3** — C● P● I● TP○ X● | LD● FS● ZS○. **Description:** Given a pre-trained generator on a source domain $\mathcal{D}_s$ and $K$ samples with class labels from a target domain $\mathcal{D}_t$, learn to generate diverse and high-quality samples conditioning on the class labels from $\mathcal{D}_t$. **Example:** VPT (Sohn et al., 2023) adapts a pre-trained conditional generator on ImageNet to Places365 with 500 images per class

**IGM** — C○ P○ I● TP○ X○ | LD○ FS● ZS○. **Description:** Given $K$ samples (usually $K = 1$) and assuming rich internal distribution for patches within these samples, learn to generate diverse and high-quality samples with the same internal patch distribution. **Example:** SinDDM (Kulikov et al., 2023) trains a generator using a single image of Marina Bay Sands, and generates variants of it

**SGM** — C○ P● I● TP● X○ | LD○ FS● ZS○. **Description:** Given a pre-trained generator, $K$ samples of a particular subject, and a text prompt, learn to generate diverse and high-quality samples containing the same subject. **Example:** DreamBooth (Ruiz et al., 2023) trains a generator using 4 images of a particular backpack and adapts it with a text-prompt to be in the *'grand canyon'*

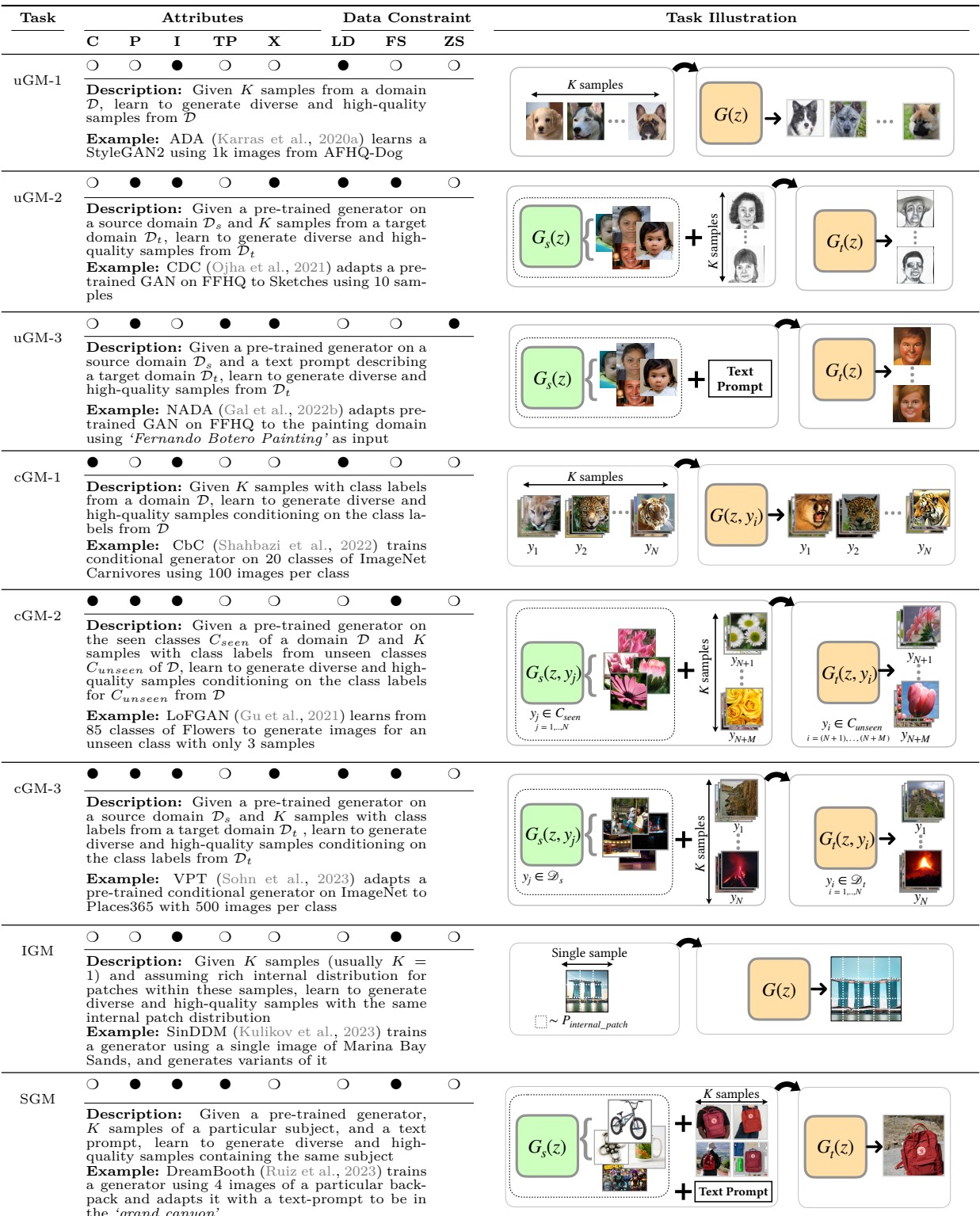

# 4 Generative Modeling under Data Constraint: Task Taxonomy, Challenges

In this section, first, we present our proposed taxonomy on different GM-DC tasks (Sec. 4.1) highlighting their relationships and differences based on their attributes, e.g. unconditional or conditional generation. Then, we present the unique challenges of GM-DC (Sec. 4.2), including new insights such as domain proximity, and incompatible knowledge transfer. Later, in Sec. 5, we present our proposed taxonomy on approaches for GM-DC, with a detailed review of individual work organized under our proposed taxonomy.

## 4.1 Generative Modeling under Data Constraint: A Taxonomy on Tasks

The goal of GM-DC is to learn to generate diverse and high-quality samples given only a small number of training samples. A number of GM-DC setups have been studied in different works (Fig. 4). In this section, we propose a **GM-DC task taxonomy** to categorize setups in different works. Tab. 2 tabulates our GM-DC task taxonomy.

1. **Unconditional generative modeling under data constraint (uGM-1).**

   **Definition 1** (uGM-1)**.** *Given $K$ samples from domain $\mathcal{D}$, learn to generate diverse and high-quality samples from $\mathcal{D}$.*

   Without leveraging other side information, existing work has studied uGM-1 under limited samples ranging from 100 to several thousands. uGM-1 is an important task, especially for a domain that is distant from common domains, e.g. medical images which are distant from common personal photos in terms of content and characteristics. In such scenarios, leveraging from common domains would not provide any advantage. We provide a comprehensive quantitative comparison of representative methods in Tab. 5.

2. **Unconditional generative modeling under data constraint with pre-trained generator and cross-domain adaptation** (uGM-2).

   **Definition 2** (uGM-2)**.** *Given a pre-trained generator on a source domain $\mathcal{D}_s$ (with numerous and diverse samples) and $K$ samples from a target domain $\mathcal{D}_t$, learn to generate diverse and high-quality samples from $\mathcal{D}_t$.*

   uGM-2 is similar to uGM-1, except that a pre-trained generator on another source domain $\mathcal{D}_s$ is additionally given. uGM-2 is a major task in GM-DC and has been studied in many works. In most works, close proximity in semantic between $\mathcal{D}_s$ and $\mathcal{D}_t$ is assumed, *e.g.* $\mathcal{D}_s$ is photos of human faces, $\mathcal{D}_t$ is sketches of human faces. For uGM-2, transfer learning has been a popular approach to tackle this task driving GM-DC into the few-shot regime, *e.g.* only 10 samples from $\mathcal{D}_t$ are given (Li et al., 2020) (See Sec. 5 for the taxonomy of GM-DC approaches). Recent work has started to look into the challenging setup when $\mathcal{D}_s$ and $\mathcal{D}_t$ are more semantically apart (Zhao et al., 2022a), *e.g.* $\mathcal{D}_s$ is photos of human faces, $\mathcal{D}_t$ is photos of cat faces. We provide a comprehensive quantitative comparison of representative methods in Tab. 6. See Sec. 4.2 for further discussion on domain proximity in GM-DC.

3. **Unconditional generative modeling under data constraint with pre-trained generator and cross-domain adaptation, using text prompt (uGM-3).**

   **Definition 3** (uGM-3)**.** *Given a pre-trained generator on a source domain $\mathcal{D}_s$ (with numerous and diverse samples) and a text prompt describing a target domain $\mathcal{D}_t$, learn to generate diverse and high-quality samples from $\mathcal{D}_t$.*

   uGM-3 is similar to uGM-2, except that a text prompt is provided to describe $\mathcal{D}_t$ instead of samples from $\mathcal{D}_t$. Particularly, this task requires generating samples from $\mathcal{D}_t$ without seeing any sample from that domain, *i.e.* zero-shot domain adaptation. Important work to tackle this task leverages recent large vision-language models to provide textual direction to guide the adaptation of the pre-trained generator to $\mathcal{D}_t$ (Gal et al., 2022b).

4. **Conditional generative modeling under data constraint (cGM-1).**

   **Definition 4** (cGM-1)**.** *Given $K$ samples with class labels from a domain $\mathcal{D}$, learn to generate diverse and high-quality samples conditioning on the class labels from $\mathcal{D}$.*

cGM-1 is similar to uGM-1 but focuses on conditional generation, *i.e.* inputs to the generator include a random latent vector and a class label. Conditional generative models such as BigGAN (Brock et al., 2019) could achieve high-quality image generation when they are trained on large-scale datasets *e.g.* ImageNet. However, under limited data, it is challenging to achieve diverse and high-quality conditional sample generation. As a natural extension of uGM-1, data augmentation has been studied for cGM-1 among other approaches, see Sec. 5.

5. **Conditional generative modeling under data constraint with pre-trained generator (cGM-2).**

**Definition 5** (cGM-2). *Given a pre-trained generator on the seen classes $C_{seen}$ of a domain $\mathcal{D}$, and $K$ samples with class labels from unseen classes $C_{unseen}$ of $\mathcal{D}$, learn to generate diverse and high-quality samples conditioning on the class labels for $C_{unseen}$ from $\mathcal{D}$.*

cGM-2 is similar to cGM-1, except that a pre-trained generator on the seen classes $C_{seen}$ is additionally given. Note that in cGM-2, $C_{seen}$ and $C_{unseen}$ contain disjoint classes, but both of them are from the same domain $\mathcal{D}$. For example, Shahbazi et al. (2021) studies the setup when CIFAR100 (Krizhevsky et al., 2009) is partitioned into 80 seen classes for the pre-trained generator and 20 unseen classes as the target, with 100 samples per unseen class given for training. Meta-learning and transfer learning (regularizer-based fine-tuning, etc.) have been effective approaches for cGM-2, see Sec. 5.

6. **Conditional Generative Modeling under data constraint with pre-trained generator and cross-domain adaptation (cGM-3).**

**Definition 6** (cGM-3). *Given a pre-trained generator on a source domain $\mathcal{D}_s$ (with numerous and diverse samples) and $K$ samples with class labels from a target domain $\mathcal{D}_t$, learn to generate diverse and high-quality samples conditioning on the class labels from $\mathcal{D}_t$.*

cGM-3 is similar to uGM-2 as cross-domain adaptation is required in both tasks, but cGM-3 focuses on conditional generation while uGM-2 focuses on unconditional generation. Furthermore, cGM-3 is similar to cGM-2, but seen classes and unseen classes are from different domains in cGM-3. For example, Shahbazi et al. (2021) has studied the setup when a pre-trained generator on ImageNet is adapted to generate samples for several classes from Places365 (Zhou et al., 2017). Transfer learning is one of the effective approaches for cGM-3, see Sec. 5.

7. **Internal patch distribution Generative Modeling (IGM).**

**Definition 7** (IGM). *Given $K$ samples and assuming rich internal distribution for patches within these samples, learn to generate diverse and high-quality samples with the same internal patch distribution.*

IGM aims to capture the internal distribution of patches within the samples. With the model capturing the samples' patch statistics, it is then possible to generate high quality, diverse samples with the same content as the given training samples. In most works, $K = 1$, and IGM focuses on images (Shaham et al., 2019), learning to generate new images with significant variability while maintaining both the global structure and fine textures of the training image.

8. **Subject-driven Generative Modeling (SGM).**

**Definition 8** (SGM). *Given $K$ samples of a particular subject and a text prompt, learn to generate diverse and high-quality samples containing the same subject.*

SGM is a recent GM-DC task introduced in Ruiz et al. (2023). Given a few images (3-5 in most cases) of a subject and leveraging a large text-to-image generative model, Ruiz et al. (2023) learns to generate diverse images of the subject in different contexts with the guidance of text prompts. The goals are: i) to achieve natural interactions between the subject and diverse new contexts, and ii) to maintain high fidelity to the key visual features of the subject. In Ruiz et al. (2023), a natural language-guided transfer learning approach and a new prior preservation loss have been proposed to achieve SGM.

Table 3: **Our proposed taxonomy for approaches in GM-DC.** For each approach, the addressed GM-DC tasks (see Tab. 2 for task definitions) and the data constraints are indicated. A detailed list of methods under each sub-category is also tabulated (some methods are under multiple categories). ○/● denotes the absence/presence of the tasks commonly addressed by each approach, and the data constraints usually considered: **LD**: Limited-Data, **FS**: Few-Shot and **ZS**: Zero-Shot.

---

### Transfer Learning (Sec. 5.1)

**Description:** Improve GM-DC on target domain by knowledge of a generator pre-trained on source domain (with numerous and diverse samples).

**Task:** uGM-1 ○ uGM-2 ● uGM-3 ● cGM-1 ○ cGM-2 ● cGM-3 ● IGM ○ SGM ● **Data constraint:** LD ● FS ● ZS ●

**1) Regularizer-based Fine-Tuning:** Explore regularizers to preserve source generators' knowledge.
*Methods:* TGAN (Wang et al., 2018), BSA (Noguchi & Harada, 2019), FreezeD (Mo et al., 2020), EWC (Li et al., 2020), CDC (Ojha et al., 2021), cGANTransfer (Shahbazi et al., 2021), $W^3$ (Grigoryev et al., 2022), $C^3$ (Lee et al., 2021), DCL (Zhao et al., 2022b), RSSA (Xiao et al., 2022), fairTL (Teo et al., 2023), GenOS(Zhang et al., 2022b) , SVD (Robb et al., 2020), $D^3$-TGAN (Wu et al., 2023), JoJoGAN (Chong & Forsyth, 2022), KDFSIG (Hou et al., 2022a), CtlGAN (Wang et al., 2022c), ICGAN (Casanova et al., 2021), MaskD (Zhu et al., 2022a), $F^3$ (Kato et al., 2023b), ICGAN (Casanova et al., 2021), DDPM-PA (Zhu et al., 2022b), DWSC (Hou et al., 2022b), CSR (Gou et al., 2023), ProSC (Moon et al., 2023), DOGAN (Hu et al., 2024a), AnyDoor (Chen et al., 2024a), SmoothSim (Sushko et al., 2023), TAN (Wang et al., 2023b), FPTGAN (Zhang et al., 2023e), CLCR (Zhang et al., 2023b), FastFaceGAN (Kato et al., 2023a), FDDC (Hu et al., 2023a), StyleDomain (Alanov et al., 2023), DomainExpansion (Nitzan et al., 2023), CVD-GAN (Jiang et al., 2023a), Def-DINO (Zhou et al., 2024a), HDA (Li et al., 2024c), FAGAN (Cheng et al., 2024), SSCR (Israr et al., 2024), DPMs-ANT (Wang et al., 2024), SACP (He et al., 2024), DATID-3D (Kim & Chun, 2023)

**2) Latent Space:** Explore latent space of source generator to identify suitable knowledge for adaptation.
*Methods:* MineGAN (Wang et al., 2020b), MineGAN++ (Wang et al., 2021), GenDA (Yang et al., 2021b), $TF^2$ (Yu et al., 2024), LCL (Mondal et al., 2023), SoLAD (Mondal et al., 2024), WeditGAN (Duan et al., 2023b), CRDI (Cao & Gong, 2025), SiSTA (Thopalli et al., 2023), MultiDiffusion (Bar-Tal et al., 2023), DiS (Everaert et al., 2023)

**3) Modulation:** Leverage trainable modulation weights on top of frozen weights of the source generator.
*Methods:* AdaFMGAN (Zhao et al., 2020a), GAN-Memory (Cong et al., 2020), CAM-GAN (Varshney et al., 2021), AdAM (Zhao et al., 2022a), DynaGAN (Kim et al., 2022b), HyperDomainNet (Alanov et al., 2022), NICE (Ni & Koniusz, 2023), $A^3FT$ (Moon et al., 2022), LFS-GAN (Seo et al., 2023), OKM (Zhang et al., 2024f), CFTS-GAN (Ali et al., 2025), HyperGAN-CLIP (Anees et al., 2024), DPH (Li et al., 2024d), DoRM (Wu et al., 2024), Mix-of-Show (Gu et al., 2023), Orthogonal Adaptation (Po et al., 2024), DreamMatcher (Nam et al., 2024), PortraitBooth (Peng et al., 2024), DisenDiff (Zhang et al., 2024b), RealCustom (Huang et al., 2024a)

**4) Natural Language-guided:** Use the feedback of vision-language models to adapt the source generator with text prompts.
*Methods:* StyleGAN-NADA (Gal et al., 2022b), MTG (Zhu et al., 2022c), HyperDomainNet (Alanov et al., 2022), DiFa (Zhang et al., 2022a), OneCLIP (Kwon & Ye, 2023), IPL (Guo et al., 2023), SVL (Jeon et al., 2023), AIR (Liu et al., 2025a), StyleGAN-Fusion (Song et al., 2024a), UniHDA (Li et al., 2024b), ITI-GEN (Zhang et al., 2023a), FairQueue (Teo et al., 2024b), SINE (Zhang et al., 2023f), DreamBooth (Ruiz et al., 2023), Custom Diffusion (Kumari et al., 2023b), Textual-Inversion (Gal et al., 2022a), SpecialistDiffusion (Lu et al., 2023), BLIP-Diffusion (Li et al., 2023a), AblateConcept (Kumari et al., 2023a), StyO (Li et al., 2024a), HyperGAN-CLIP (Anees et al., 2024), DPH (Li et al., 2024d), ELITE (Wei et al., 2023), E4T (Gal et al., 2023), MoMA (Song et al., 2025), SSR-Encoder (Zhang et al., 2024c), MultiGen (Wu et al., 2025), MasterWeaver (Wei et al., 2025b), Lego (Motamed et al., 2025), CGR (Jin et al., 2025), DreamBlend (Ram et al., 2025), Cross Initialization (Pang et al., 2024), SAG (Chan et al., 2024), ZipLoRA (Shah et al., 2025), RealCustom (Huang et al., 2024a), PALP (Arar et al., 2024), InstantBooth (Shi et al., 2023b), IDAdapter (Cui et al., 2024), Domain gallery (Duan et al., 2024), LogoSticker (Zhu et al., 2025), ProSpect (Zhang et al., 2023d), Dreambooth-CL (Zhu & Yang, 2024), AnomalyDiffusion (Hu et al., 2024b), ComFusion (Hong et al., 2025), SuDe (Qiao et al., 2024), LFS-Diffusion (Song et al., 2024b), $L^2DM$ (Sun et al., 2024), Omg (Kong et al., 2025), TFIC (Li et al., 2025b), MagiCapture (Hyung et al., 2024), Mix-of-Show (Gu et al., 2023), Orthogonal Adaptation (Po et al., 2024), DBLoRA (Pascual et al., 2024), HybirdBooth (Guan et al., 2025), T2IRL (Wei et al., 2025a), HyperDreamBooth (Ruiz et al., 2024), FastComposer (Xiao et al., 2024), PortraitBooth (Peng et al., 2024), DreamMatcher (Nam et al., 2024), DisenDiff (Zhang et al., 2024b), CII (Jeong et al., 2023), DETEX (Cai et al., 2024)

**5) Adaptation-Aware:** Preserve the source generator's knowledge that is important to the adaptation task.
*Methods:* AdAM (Zhao et al., 2022a), RICK (Zhao et al., 2023a), OKM (Zhang et al., 2024f)

**6) Prompt Tuning:** Freeze the source generator and add/ generate visual prompts to guide generation for the target domain.
*Methods:* VPT (Sohn et al., 2023)

---

### Data Augmentation (Sec. 5.2)

**Description:** Improve GM-DC by increasing coverage of the data distribution by applying various transformations on the given samples.

**Task:** uGM-1 ● uGM-2 ○ uGM-3 ○ cGM-1 ○ cGM-2 ○ cGM-3 ○ IGM ○ SGM ○ **Data constraint:** LD ● FS ○ ZS ○

**1) Image-Level Augmentation:** Apply data transformations on image space.
*Methods:* ADA (Karras et al., 2020a), DiffAugment (Zhao et al., 2020b), IAG (Zhao et al., 2020c), DiffusionGAN (Wang et al., 2023d), bCR (Zhao et al., 2021), CR-GAN (Zhang et al., 2020), APA (Jiang et al., 2021), PatchDiffusion (Wang et al., 2023c), ANDA (Zhang et al., 2024d), DANI (Zhang et al., 2024e), AugSelf-GAN (Hou et al., 2024)

**2) Feature-Level Augmentation:** Apply data transformations on the feature space.
*Methods:* AdvAug (Chen et al., 2021a), AFI (Dai et al., 2022), FSMR (Kim et al., 2022a)

**3) Transformation-Driven Design:** Leverage the information of individual transformations to design an efficient learning mechanism.
*Methods:* DAG (Tran et al., 2021), SSGAN-LA (Hou et al., 2021)

---

### Network Architectures (Sec. 5.3)

**Description:** Design specific architecture for the generator to improve its learning under data constraints.

**Task:** uGM-1 ● uGM-2 ○ uGM-3 ○ cGM-1 ● cGM-2 ○ cGM-3 ○ IGM ○ SGM ○ **Data constraint:** LD ● FS ○ ZS ○

**1) Feature Enhancement:** Design additional modules/ layers to enhance/ retain the feature maps of the generator for better generative modeling.

*Methods:* FastGAN (Liu et al., 2021), cF-GAN (Hiruta et al., 2022), MoCA (Li et al., 2022b), DFSGAN (Yang et al., 2023a), DM-GAN (Yan et al., 2024), FewConv (Liu et al., 2025b), SCHA-VAE (Giannone & Winther, 2022)

**2) Ensemble Large Pre-trained Vision Models:** Improve architecture by integrating pre-trained vision models to enable more accurate GM-DC.
*Methods:* ProjectedGAN (Sauer et al., 2021), SPGAN (Hiruta et al., 2022), Vision-aided GAN (Kumari et al., 2022), P2D (Chong et al., 2024), DISP (Mangla et al., 2022)

**3) Dynamic Network Architecture:** Improve generative learning with limited data by evolving the generator architecture during training.
*Methods:* CbC (Shahbazi et al., 2022), PYP (Li et al., 2024e), DynamicD (Yang et al., 2022a), AdvAug (Chen et al., 2021a), Re-GAN (Saxena et al., 2023), RG-GAN (Saxena et al., 2024), AutoInfoGAN (Shi et al., 2023a)

### Multi-Task Objectives (Sec. 5.4)

| **Description:** | Introduce additional task(s) to extract generalizable representations that are useful for all tasks, to reduce overfitting under data constraints. |
|---|---|
| **Task:** | uGM-1 ● uGM-2 ● uGM-3 ○ cGM-1 ● cGM-2 ○ cGM-3 ○ IGM ○ SGM ○ **Data constraint:** LD ● FS ● ZS ○ |

**1) Regularizer:** Add an additional task objective as a regularizer to prevent an undesirable behaviour during training generative model.
*Methods:* LeCam (Tseng et al., 2021), RegLA (Hou, 2023), DigGAN (Fang et al., 2022), MICGAN (Zhai et al., 2024), CHAIN (Ni & Koniusz, 2024), MDL (Kong et al., 2022), DFMGAN (Duan et al., 2023a)

**2) Contrastive Learning:** Introduce a pretext task to enhance the learning process of the generative model.
*Methods:* InsGen (Yang et al., 2021a), FakeCLR (Li et al., 2022c), DCL (Zhao et al., 2022b), $C^3$ (Lee et al., 2021), ctlGAN (Wang et al., 2022c), IAG (Zhao et al., 2020c), CML-GAN (Phaphuangwittayakul et al., 2022), RCL (Gou et al., 2024)

**3) Masking:** Mask a part of the image/ information to increase the task hardness and prevent learning the trivial solutions.
*Methods:* MaskedGAN (Huang et al., 2022), MaskD (Zhu et al., 2022a), DMD (Zhang et al., 2023c)

**4) Knowledge Distillation:** Add a task objective that enforces the generator to follow a strong teacher.
*Methods:* KD-DLGAN (Cui et al., 2023), KDFSIG (Hou et al., 2022a), BK-SDM (Kim et al., 2025)

**5) Prototype Learning:** Emphasize learning prototypes for samples/ concepts within the distribution as an additional task objective.
*Methods:* ProtoGAN (Yang et al., 2023b), MoCA (Li et al., 2022b)

**6) Other Multi-Task Objectives:** Apply other types of multi-task objectives including co-training, patch-level learning, and diffusion.
*Methods:* GenCo (Cui et al., 2022), PatchDiffusion (Wang et al., 2023c), AnyRes-GAN (Chai et al., 2022), DiffusionGAN (Wang et al., 2023d), D2C (Sinha et al., 2021b), AdaptiveIMLE (Aghabozorgi et al., 2023), RS-IMLE (Vashist et al., 2024), FSDM (Giannone et al., 2022), SpiderGAN (Asokan & Seelamantula, 2023)

### Exploiting Frequency Components (Sec. 5.5)

| **Description:** | Exploit frequency components to improve learning the generative model by reducing frequency bias. |
|---|---|
| **Task:** | uGM-1 ● uGM-2 ○ uGM-3 ○ cGM-1 ○ cGM-2 ● cGM-3 ○ IGM ○ SGM ○ **Data constraint:** LD ● FS ● ZS ○ |

*Methods:* FreGAN (Yang et al., 2022c), WaveGAN (Yang et al., 2022b), MaskedGAN (Huang et al., 2022), Gen-co (Cui et al., 2022), FAGAN (Cheng et al., 2024), SDTM (Yang et al., 2023c)

### Meta-Learning (Sec. 5.6)

| **Description:** | Learn meta-knowledge from seen classes to improve generator learning for unseen classes. |
|---|---|
| **Task:** | uGM-1 ○ uGM-2 ○ uGM-3 ○ cGM-1 ○ cGM-2 ● cGM-3 ○ IGM ○ SGM ○ **Data constraint:** LD ○ FS ● ZS ○ |

**1) Optimization:** Learn initialization weights from the seen classes as meta-knowledge to enable quick adaptation to unseen classes.
*Methods:* GMN (Bartunov & Vetrov, 2018), FIGR (Clouâtre & Demers, 2019), Dawson (Liang et al., 2020), FAML (Phaphuangwittayakul et al., 2021), CML-GAN (Phaphuangwittayakul et al., 2022)

**2) Transformation:** Learn sample transformations from the seen classes as meta-knowledge and use them for sample generation for unseen classes.
*Methods:* DAGAN (Antoniou et al., 2017), DeltaGAN (Hong et al., 2022a), Disco (Hong et al., 2022b), AGE (Ding et al., 2022), SAGE (Ding et al., 2023), HAE (Li et al., 2022a), LSO (Zheng et al., 2023), TAGE (Zhang et al., 2024a), CDM (Gupta et al., 2024), ISSA (Huang et al., 2021), MFH (Xie et al., 2022)

**3) Fusion:** Learn to fuse the samples of the seen classes as meta-knowledge, and apply learned meta-knowledge to generation for unseen classes.
*Methods:* MatchingGAN (Hong et al., 2020a), F2GAN (Hong et al., 2020b), LofGAN (Gu et al., 2021), WaveGAN (Yang et al., 2022b), AMMGAN (Li et al., 2023b), MVSA-GAN (Chen et al., 2023), EqGAN (Zhou et al., 2023), SDTM (Yang et al., 2023c), SMR-CSL (Xiao et al., 2025), SAGAN (Aldhubri et al., 2024), F2DGAN (Zhou et al., 2024b)

### Modeling Internal Patch Distribution (Sec. 5.7)

| **Description:** | Learn the internal patch distribution within one image to generate diverse samples with the same visual content (patch distribution). |
|---|---|
| **Task:** | uGM-1 ○ uGM-2 ○ uGM-3 ○ cGM-1 ○ cGM-2 ○ cGM-3 ○ IGM ● SGM ○ **Data constraint:** LD ○ FS ● ZS ○ |

**1) Progressive Training:** Train a generative model progressively to learn the patch distribution at different scales/ noise levels.
*Methods:* SinDiffusion (Wang et al., 2022a), SinDDM (Kulikov et al., 2023), Deff-GAN (Kumar & Sivakumar, 2023), BlendGAN (Kligvasser et al., 2022), SinGAN (Shaham et al., 2019), ConSinGAN (Hinz et al., 2021), CCASinGAN (Wang et al., 2022b), PromptSDM (Park et al., 2024), LatentSDM (Han et al., 2024b), SD-SGAN (Yildiz et al., 2024), SA-SinGAN (Chen et al., 2021b), ExSinGAN (Zhang et al., 2021), TcGAN (Jiang et al., 2023b), RecurrentSinGAN (He & Fu, 2021)

**2) Non-progressive Training:** Train a generative model on the same scale/ noise but with changes to the model's architecture.
*Methods:* SinFusion (Nikankin et al., 2023), One-Shot GAN (Sushko et al., 2021), PetsGAN (Zhang et al., 2022c)

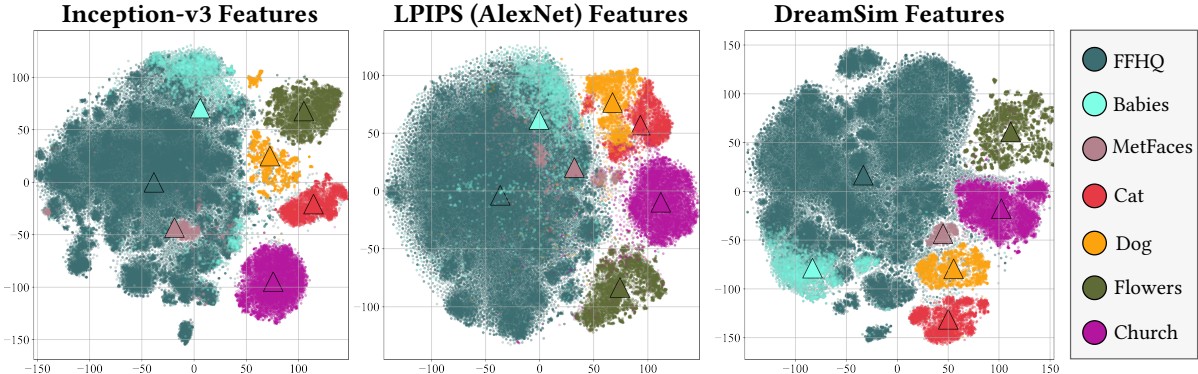

Figure 5: **Source-target domain proximity visualization indicates that distant/ remote target domains have not been explored in GM-DC setups and are very challenging.** We use FFHQ (Karras et al., 2019) as the source domain. We show source-target domain proximity qualitatively by visualizing **Inception-v3 (Left)** (Szegedy et al., 2016), **LPIPS (Middle)** (Zhang et al., 2018) and **DreamSim (Right)** (Fu et al., 2023) features. For feature visualization, we use t-SNE (van der Maaten & Hinton, 2008) and show centroids ($\triangle$) for all domains. We clearly show using feature visualizations that additional setups – Flowers (Nilsback & Zisserman, 2008) and Church (Yu et al., 2015) – represent target domains that are remote from the source domain (FFHQ) compared to target domains used in the literature. This indicates that the exploration of distant/ remote target domains under GM-DC setups has not been pursued and poses notable challenges (Fig. 6). Best viewed in color.

## 4.2 Generative Modeling under Data Constraint: Challenges

### 4.2.1 Challenges for Training Generative Models under Data Constraint

Data constraints typically introduce additional challenges and amplify existing ones when training generative models. Here, we delve into the challenges of training GM-DC. These limitations include pervasive issues of overfitting and frequency bias which are commonly observed across various approaches. Additionally, knowledge transfer between domains brings forth specific problems including the proximity between source and target domains and the transfer of incompatible source knowledge. As shown in Fig. 3, a lot of works directly rely on knowledge transfer as a mainstream method to tackle GM-DC, and a number of works propose methods based on other approaches that are compatible with transfer learning.

**Overfitting to Training Data.** In machine learning, overfitting is a common issue when powerful models start to memorize the training data instead of learning the generalizable semantics (Santos & Papa, 2022). In generative modeling, the overfitting problem exacerbates under data constraints due to the high capacity of current generative models (Noguchi & Harada, 2019; Liu et al., 2021; Karras et al., 2020a). When limited training data is available, generative models may simply remember the training data (Li et al., 2020; Ojha et al., 2021) and learn to generate the exact training samples (Zhao et al., 2022a) instead of capturing the data distribution. Furthermore, under data constraints, generative modeling is more prone to mode collapse (Tran et al., 2021), i.e., the generators learn only a limited set of modes and fail to capture other modes of the data distribution, resulting in limited diversity in generated samples (Yu et al., 2022; Nguyen et al., 2023).

**Frequency Biases.** Generative models are notorious for their spectral bias (Rahaman et al., 2019; Khayatkhoei & Elgammal, 2022), *i.e.* tendency to prioritize fitting low-frequency components while disregarding high-frequency components within a data distribution (Durall et al., 2020; Tancik et al., 2020; Chandrasegaran et al., 2021). The exclusion of these high-frequency components which encode intricate image details (Gonzales & Wintz, 1987) can significantly impact the quality of generated samples, *i.e.* , accurate modeling of high-frequency details is critical in various fields including medical imaging (X-rays, CT-scans, MRIs), satellite/ aerial imaging, astrophotography, and art restoration. This issue becomes more severe under limited data (Yang et al., 2022c;b) as even advanced network structures tailored for such scenarios (Liu et al., 2021) struggle to maintain the desired level of details in generated samples.

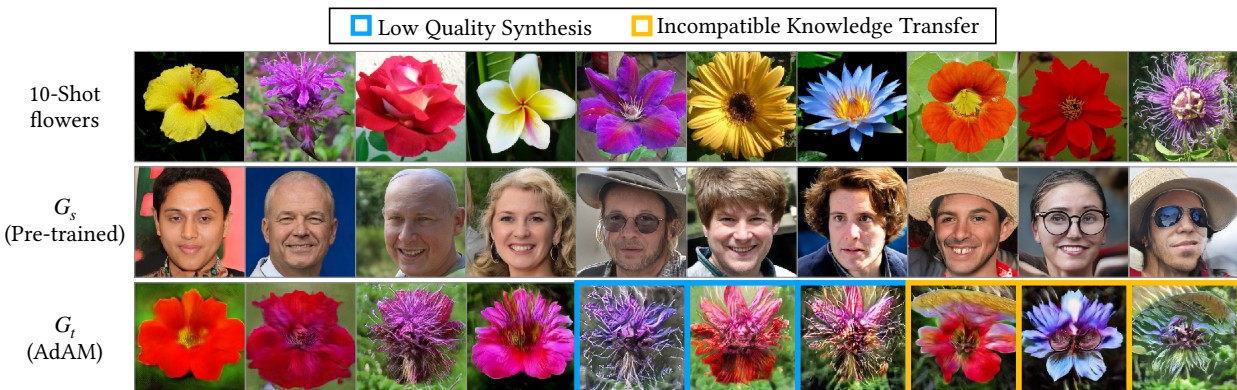

Figure 6: **Knowledge transfer under distant/ remote target domain (Human face → Flowers) suffers from low synthesis quality and incompatible knowledge transfer.** We show 10-shot adaptation results for FFHQ → Flowers using AdAM (Zhao et al., 2022a). The FID for the 10-shot adaptation using AdAM is 124.46. We highlight multiple instances of low synthesis quality and incompatible knowledge transfer (*i.e.* glasses, hat from FFHQ to flowers), showing that GM-DC modeling of remote target domains poses significant challenges. Best viewed in color.

**Modeling Distant/ Remote Target Domains under GM-DC Setups.** Substantial number of GM-DC tasks rely on the transfer learning principle (uGM-2, uGM-3, cGM-2, cGM-3, SGM), which aims to enhance the generative capabilities for a target domain by leveraging the knowledge of a generator pre-trained on a large and diverse source domain (See Fig. 1). A significant amount of research has been focused on target domains that are semantically/ perpetually similar to the source domain, *e.g.*, learn to generate Baby faces using a pre-trained generator trained on Human faces (Gal et al., 2022b; Li et al., 2024c; Liu et al., 2025a). In particular, when dealing with GM-DC setups involving significant domain shifts between the source and target domains (Human Faces→Animal Faces), many proposed methods fail to outperform a basic finetuning approach (Zhao et al., 2022a). This is due to these methods prioritizing knowledge preservation from the source domain/ task, overlooking the adaptation step to the target domain (Zhao et al., 2022a). Adaptation-aware algorithms have characterized source→target domain proximity (Zhao et al., 2022a) and addressed GM-DC setups with pronounced domain shifts between the source and target domains (Human Faces→Animal Faces) (Zhao et al., 2022a; 2023a). To understand the concept of distant/ remote target domains, we additionally introduce two remote target domains that further exhibit a considerable degree of domain shifts: i) Human Faces (FFHQ) (Karras et al., 2019)→ Flowers (Nilsback & Zisserman, 2008), ii) Human Faces (FFHQ) (Karras et al., 2019)→Church (Yu et al., 2015). Domain proximity visualization is shown in Fig. 5. In particular, we conducted a GM-DC experiment (uGM-2) to adapt a pre-trained Human face (FFHQ) generator to Flowers under 10-shot setup using AdAM (Zhao et al., 2022a), obtaining a FID value of 124.46. Adaptation results are shown in Fig. 6. As one can observe, multiple instances of low quality synthesis are observed in AdAM (Zhao et al., 2022a). In summary, we remark that modeling distant/ remote target domains remains an important and challenging area for GM-DC.

**Identifying and Removing Incompatible Knowledge Transfer.** Another challenge with leveraging source domain's knowledge for GM-DC tasks is incompatible knowledge transfer, which is discovered in Zhao et al. (2023a). In particular, many methods may transfer knowledge that is incompatible with the target domain, *e.g.* hat from source domain FFHQ to target domain flowers, significantly degrading the realisticness of the generated samples. In Fig. 6, we show multiple examples of incompatible knowledge transfer using AdAM for 10-shot flower adaptation. Although some recent effort has been invested in identifying and proactively truncating incompatible knowledge transfer (Zhao et al., 2023a) in Human Faces → Animal Faces adaptation setups, it is worth noting that identifying and removing incompatible knowledge remains a critical and demanding area in GM-DC.

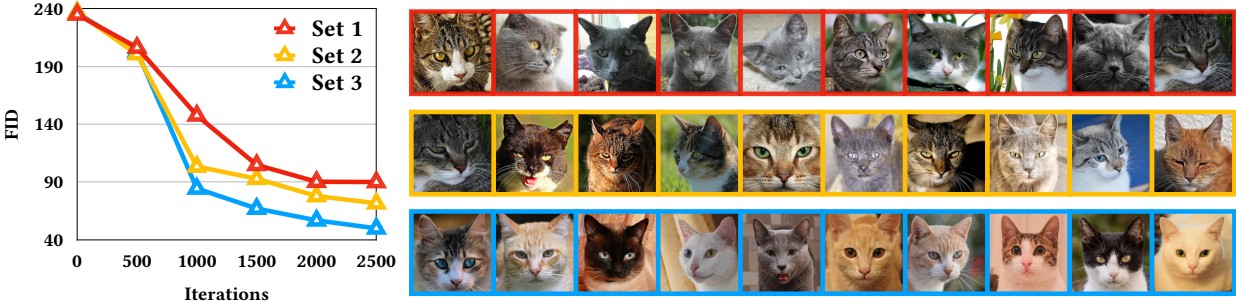

Figure 7: **Sample selection for GM-DC remains challenging and relatively unexplored.** We use AdAM (Zhao et al., 2022a) to adapt a pre-trained StyleGAN2 on FFHQ to AFHQ-Cat dataset (Choi et al., 2020) using 3 random sets of 10-shot data *(Right)*. We report FID results during training *(Left)* for these sets of data. Following Zhao et al. (2022a), FID is measured between 5000 generated samples and the entire AFHQ-Cat dataset consisting of 5115 samples. We use clean-FID library (Parmar et al., 2022), obtaining FID values of 90.0, 71.6 and 49.9 for Sets 1, 2, and 3 respectively at iteration=2500. As indicated by FID trends, the generative capabilities of GM-DCs are drastically influenced by sample selection. Best viewed in color.

### 4.2.2 Challenges on Selecting Samples for GM-DC

Although considerable research effort has been invested in developing algorithms for GM-DC, the task of sample selection for GM-DC remains a challenging and relatively unexplored area. It is essential that the samples selected for GM-DC should represent the target domain. In particular, we observe significant variation in performance with different selection of target samples as the training datasets in GM-DC. We perform a 10-shot *data-centric* GM-DC experiment using AdAM (Zhao et al., 2022a) to emphasize the importance of sample selection in GM-DC. Following Zhao et al. (2022a; 2023a), we use AFHQ-Cat dataset (Choi et al., 2020) and select 3 random sets of 10-shot cat data for GM-DC. Data and 10-shot adaptation FID results are shown in Fig. 7. We obtain FID values of 90.0, 71.6 and 49.9 for Sets 1, 2 and 3 respectively (iteration=2500). This study provides evidence that sample selection plays a vital role in determining the capabilities of GM-DC. Specifically, due to cost/ privacy concerns, the role of sample selection is critical in applications including biomedical imaging, satellite/ aerial imaging and remote sensing. In summary, sample selection for GM-DC holds significant importance and remains an area with limited investigation thus far.

### 4.2.3 Challenges in Evaluating Generative Models under Data Constraint

The assessment of generative modeling capabilities presents lots of challenges, encompassing both objective and subjective evaluation (Kynkäänniemi et al., 2023). These issues are aggravated under low-data regimes resulting in the evaluation of GM-DC to be challenging and an active topic of research. In contemporary GM-DC literature, sample quality and diversity are used as the main attributes for evaluating generation capability. A summary of prominent metrics for GM-DC is included in Tab. 4.

Existing GM-DC evaluation metrics present multiple challenges: i) Statistical measures including FID, KID, IS, $FID_{CLIP}$ lose their significance when dealing with setups where there is an extreme scarcity (Few-shots) or complete absence (Zero-shot) of target domain data. For example, when the reference distribution contains only 10 real images, the mean and trace components of FID are not statistically significant. ii) Although human judgment/ user feedback is used for the subjective evaluation of GM-DC, the absence of a unified framework/ protocol for such evaluation strategy results in inadequacy when comparing the generative capabilities of different GM-DC models. iii) The over-reliance on objective GM-DC measures on deep features extracted from pre-trained networks remains challenging and relatively unexplored. For example, FID, KID, and IS use features extracted from an Inception model trained on ImageNet-1K (Deng et al., 2009); LPIPS, and Intra-LPIPS, use features extracted from models trained on BAPPS (Zhang et al., 2018) dataset. Although these pre-trained models effectively function as general-purpose feature extractors, their ability to capture properties/ attributes of out-of-domain data to objectively quantify the capabilities

Table 4: Common metrics used for evaluating GM-DC works (Zhao et al., 2022a; Teo et al., 2023; Liu et al., 2025a). **LD**: Limited-Data, **FS**: Few-Shot, **ZS**: Zero-Shot. ○/●denotes the absence/presence, respectively.

| Metric | LD | FS | ZS |
|---|---|---|---|
| FID (Heusel et al., 2017)/ FID$_{\text{CLIP}}$ (Kynkäänniemi et al., 2023) | ● | ● | ○ |
| KID (Bińkowski et al., 2018) | ● | ● | ○ |
| IS (Salimans et al., 2016) | ● | ● | ○ |
| Intra-LPIPS (Ojha et al., 2021) | ○ | ● | ● |
| SIFID (Shaham et al., 2019) | ○ | ● | ● |
| Image/ Text Similarity (Gal et al., 2022a) | ○ | ● | ● |
| User Feedback | ● | ● | ● |

of GM-DC requires more investigation, *e.g.* medical images. In summary, the area of evaluation measures for GM-DC cannot be overstated, as it remains critical and challenging.

## 5 Comprehensive Review

In this section, first, we will present our proposed taxonomy of approaches for GM-DC which systematically categorizes and organizes GM-DC methods under seven approaches (Tab. 3) based on the principal ideas of these methods. Then, we will discuss individual GM-DC methods organized under our proposed taxonomy.

**Our Proposed Taxonomy of Approaches for GM-DC** categorizes GM-DC methods into seven groups:

1. **Transfer Learning:** In GM-DC, transfer learning (TL) aims to improve the learning of the generator for the target domain using the knowledge of a generator pre-trained on a source domain (with numerous and diverse samples). For example, some methods under this category use the knowledge of a Stable Diffusion Rombach et al. (2022) pre-trained on LAION-400M Schuhmann et al. (2021) to learn to generate diverse and high-quality samples of particular subject(s), given only a few images of the subject(s) Ruiz et al. (2023); Gal et al. (2022a); Kumari et al. (2023b). Major challenges for TL-based GM-DC are to identity, select, and preserve suitable knowledge of the source generator for the target generator. Along this line, there are six subcategories: i) *Regularization-based Fine-tuning*, explores regularizers to preserve suitable source generator's knowledge to improve learning target generator; ii) *Latent Space*, explores transforming/ manipulating the source generator's latent space to identify suitable knowledge for adaptation; iii) *Modulation*, freezes and transfers weights of the source generator to the target generator and adds trainable modulation weights on top of frozen weights to increase the adaptation capability to the target domain; iv) *Natural Language-guided*, uses natural language prompt and supervision signal from language-vision models to adapt source generator to target domain; v) *Adaptation-Aware*, identifies and preserves the source generator's knowledge that is important to the adaptation task; vi) *Prompt Tuning*, is an emerging idea that freezes the weights of the source generator and learns to generate visual prompts (tokens) to guide generation for the target domain.

2. **Data Augmentation:** Augmentation aims to improve GM-DC by increasing coverage of the data distribution with applying various transformations $\{T_k\}_{k=1}^K$ to available data. For example, within this category, some works augment the available limited data to train an unconditional StyleGAN2 (Karras et al., 2020b) using the 100-shot Obama dataset or train a conditional BigGAN (Brock et al., 2019) with only 10% of the CIFAR-100 dataset. A major challenge of these approaches is augmentation leakage, where the generator learns the augmented distribution, *e.g.* , generating rotated/ noisy samples. There are three representative categories: i) *Image-Level Augmentation*, applies the transformations on the image space; ii) *Feature-Level Augmentation*, applies the transformations on the feature space; iii) *Transformation-Driven Design*, leverages the information on each individual transformation $T_k$ to design an efficient learning mechanism.

3. **Network Architectures:** These approaches design specific architectures for the generators to improve their learning under data constraints. Some works in this category design shallow/ sparse generators to

prevent overfitting to training data due to over-parameterization. The primary challenge is that when endeavoring to design a new architecture, the process of discovering the optimal hyperparameters can be laborious. There are three major types of architectural designs for GM-DC: i) *Feature Enhancement*, introduces additional modules to enhance/ retain the knowledge within feature maps; ii) *Ensemble Large Pre-trained Vision Models*, leverages large pre-trained vision models to aid more accurate generative modeling; iii) *Dynamic Network Architecture*, evolves the architecture of the generative model during training to compensate for data constraints.

4. **Multi-Task Objectives:** These approaches modify the learning objective of the generative model by introducing additional task(s) to extract generalizable representations and reduce overfitting under data constraints. As an example, some works define a pretext task based on contrastive learning (He et al., 2020) to pull the positive samples together and push negative ones away in addition to the original generative learning task. The efficient integration of the new objective with the generative learning objective could be challenging under data constraints. These works can be categorized into several approaches: i) *Regularizer*, adds an additional learning objective as a regularizer to prevent an undesirable behavior during training a generative model under data constraints. Note that this category is different from regularizer-based fine-tuning, as the latter aims to preserve source knowledge, but the former is for training without a source generator; ii) *Contrastive Learning*, adds the learning objective related to a pretext task to enhance the learning process of the generative model using an additional supervision signal from solving this pretext task; iii) *Masking*, introduces alternative learning objective by masking a part of the image/ information to improve generative modeling by increasing the task hardness and preventing learning the trivial solutions; iv) *Knowledge Distillation*, introduces an additional learning objective that enforces the generator to follow a strong teacher; v) *Prototype Learning*, emphasizes learning prototypes for samples/ concepts within the distribution as an additional objective; vi) *Other Multi-Task Objectives*, include co-training, patch-level learning, and using diffusion to enhance generation.

5. **Exploiting Frequency Components:** Deep generative models exhibit frequency bias tending to ignore high-frequency signals as they are hard to generate (Schwarz et al., 2021). Data constraints can exacerbate this problem (Yang et al., 2022c). The approaches in this category aim to improve frequency awareness of the generative models by leveraging frequency components during training. For instance, certain approaches employ Haar Wavelet transform to extract high-frequency components from the samples. These frequency components are then fed into various layers using skip connections, aiming to alleviate the challenges associated with generating high-frequency details. Despite its effectiveness, utilizing frequency components for GM-DC has not been thoroughly investigated. The performance can be enhanced by incorporating more advanced techniques for extracting frequency components.

6. **Meta-Learning:** These approaches create sample generation tasks with data constraints for the seen classes, and learn the meta-knowledge—knowledge that is shared between all tasks—across these tasks during meta-training. This meta-knowledge is then used in improving generative modeling for the unseen classes with data constraints. For instance, some studies, as meta-knowledge, learn to fuse the samples from the seen categories $C_{seen}$ of the Flowers dataset (Nilsback & Zisserman, 2008) for sample generation. This meta-knowledge enables the model to generate new samples from unseen classes $C_{unseen}$ within the same dataset ($C_{seen} \cap C_{unseen} = \emptyset$) by fusing only 3 samples from each class. Note that as these works employ episodic learning within a generative framework, the training stability can be impacted. Approaches within this line can be classified into three categories: i) *Optimization*, initializes the generative model with weights learned on the seen classes as meta-knowledge, to enable quick adaptation to unseen classes with limited steps of the optimization; ii) *Transformation*, learns cross-category transformations from the samples of the seen classes as meta-knowledge and applies them to available samples of the unseen classes to generate new samples; iii) *Fusion*, learns to fuse the samples of the seen classes as meta-knowledge, and applies learned meta-knowledge to sample generation by fusing samples of the unseen classes.

7. **Modeling Internal Patch Distribution:** These approaches aim to learn the internal patch distribution within one image (in some cases a few images), and then generate diverse samples that carry the same visual content (patch distribution) with an arbitrary size, and aspect ratio. As an example, some works train a Diffusion Model using a single image of the "Marina Bay Sands", and after training, the Diffusion Model can generate similar images, but include additional towers topped by the similar "Sands Skypark".

However, a major limitation of these methods lies in the fact that for every single image, usually a separate generative model is trained from scratch, neglecting the potential for efficient training through knowledge transfer in this context. Approaches proposed along this line can be categorized into two major groups: i) *Progressive Training*, progressively trains a generative model to learn the patch-distribution at different scales or noise levels; ii) *Non-Progressive Training*, learns a generative model at a single scale by implementing additional sampling techniques or new model architectures.

In what follows we delve into detailed descriptions of the approaches within each category.

## 5.1 Transfer Learning

Transfer Learning (TL) is a major approach for GM-DC. Given a source generator $G_s$ (for GANs, both $G_s$ and $D_s$) pre-trained on a large and diverse source domain $\mathcal{D}_s$, these approaches aim to learn an adapted generator to the target domain $G_t$ by initializing the weights to those of the source generator.

### 5.1.1 Regularizer-based Fine-Tuning

**Early evidence of transfer learning benefits.** Early works in this category explore the effectiveness of transfer learning in the context of generative modeling with limited data, analyzing different aspects such as the effect of source and target domain distance, the size of the target domain dataset, and even the statistics of the distribution on which the source model was trained. TGAN (Wang et al., 2018) is the first systematic study to evaluate transfer learning in GANs. TGAN shows that transfer learning reduces the convergence time and improves generative modeling under limited data. The knowledge transfer is performed by using the source GAN for initializing the weights of the target GAN, followed by fine-tuning the weights on target data. TGAN (Wang et al., 2018) demonstrates that: i) transferring $D$ is more important than $G$, while transferring both $G$ and $D$ gives the best results; ii) transfer learning performance degrades by increasing the distance between source and target domains or decreasing the number of samples from target domain; iii) to select a pre-trained GAN for a target domain, in addition to a smaller distance, more dense source domains are preferable. As an example, for the Flower (Nilsback & Zisserman, 2008) target domain, surprisingly, a GAN pre-trained on semantically unrelated LSUN Bedrooms (Yu et al., 2015) is shown to be among the best sources (Wang et al., 2018). $W^3$ (Grigoryev et al., 2022) revisits the transfer learning in GANs with a modern structure—StyleGAN2-ADA (Karras et al., 2020b;a) instead of WGAN-GP (Gulrajani et al., 2017) used in TGAN. Results in Grigoryev et al. (2022) suggest that for SOTA GANs, it is beneficial to transfer the knowledge from sparse and diverse sources (pre-trained StyleGAN2 on ImageNet) rather than dense and less diverse ones. A major limitation of TGAN is that under limited data, simply fine-tuning the whole generator destroys a considerable portion of the general knowledge obtained on the source domain. Almost all of the following works aim to address this by using different approaches to preserve the knowledge of the source generator.

**Preserving source knowledge via parameter-efficient fine-tuning.** Approaches immediately after TGAN performed this knowledge preservation through parameter-efficient fine-tuning, where a small set of parameters was fine-tuned using data from the target domain while the remaining parameters were kept frozen. BSA (Noguchi & Harada, 2019) only updates scale and shift parameters in batch normalization (BN) layers during fine-tuning to prevent overfitting to limited data. FreezeD (Mo et al., 2020) hypothesizes that as $D$ performs the discriminative task during training a GAN, based on common knowledge in discriminative learning (Yosinski et al., 2014), its early layers encode general knowledge which is shared between source and target domains. Therefore, this general knowledge is preserved during adaptation by freezing early layers of $D$. cGANTransfer (Shahbazi et al., 2021) assumes that the pre-trained $G$ is conditioned on class labels using BN parameters, *i.e.* each class has its own BN parameters (Brock et al., 2019). Then, explicit knowledge propagation from seen classes to unseen classes is enforced by defining the BN parameters of the unseen classes to be the weighted average of the BN parameters of seen classes. SVD (Robb et al., 2020) applies singular value decomposition (Van Loan, 1976), and only updates the singular values that are related to changing entanglement between different attributes within data.

**Constraining adaptation to prevent source-knowledge distortion.** Later, more advanced approaches were proposed by exploring the causes of source knowledge distortion during fine-tuning and then proposing

constraints to prevent it. EWC (Li et al., 2020) aims to preserve the diversity of the source GAN during adapting to a target domain with only a few samples, *e.g.* , 10-shot. The importance of each parameter in source GAN is measured by Fisher Information (FI), and the change on each parameter during adaptation is penalized based on its importance, *e.g.* , change over important parameters is penalized more. CDC (Ojha et al., 2021) aims to keep the diversity of the generated samples using a cross-domain correspondence loss. Specifically, first, a batch of $N+1$ latent codes are sampled for image generation: $\{G(z_0), ..., G(z_N)\}$. Then, using $G(z_0)$ as a reference, the similarity of generated samples to the reference is measured for the generator before and after adaptation, resulting in two $N-way$ probability vectors. The diversity is preserved by adding the KL divergence between these two probability vectors to the standard loss as a regularizer. MaskD (Zhu et al., 2022a) applies random masks to extracted features of $D$, on top of CDC (Ojha et al., 2021), to prevent overfitting. DDPM-PA (Zhu et al., 2022b) uses a pairwise adaptation method similar to CDC for adapting diffusion models to the new domain. RSSA (Xiao et al., 2022) extends the cross-domain consistency idea of the CDC (Ojha et al., 2021) to a more constrained form by preserving the structural similarity of the samples before and after adaptation. ProSC (Moon et al., 2023) extends RRSA by performing a progressive adaptation to the target domain in $N$ iterations instead of a single adaptation to reduce the gap between pairs of domains. CSR (Gou et al., 2023) uses a similar idea to CDC but applies semantic loss directly to the spatial space, *i.e.* , generated images with $G_s$ and $G_t$. DWSC (Hou et al., 2022b) proposes the dynamic weighted semantic correspondence between the source and target generator during adaptation to preserve the diversity. SSCR (Israr et al., 2024) extends CDC by computing more accurate similarity scores by leveraging frozen Siamese networks (Chen et al., 2020).

**Contrastive objectives to preserve diversity.** A line of work employs contrastive learning during fine-tuning to prevent knowledge distortion when adapting generative models to the target domain. For instance, $C^3$ (Lee et al., 2021), DCL (Zhao et al., 2022b), and CtlGAN (Wang et al., 2022c) aim to preserve diversity by applying contrastive learning. Assuming $G_s(z_i)$ as an anchor point, the generated sample for the same latent code with the adapted generator $(G_t(z_i))$ is considered a positive pair, and the generated samples with the adapted generator for other latent code values $(G_t(z_j), i \neq j)$ are considered as negative pairs. Additionally, DCL applies similar contrastive learning to the $D$.Based on the contrastive learning idea of DCL (Zhao et al., 2022b) and LeCam regularizer (Tseng et al., 2021), CLCR (Zhang et al., 2023b) proposes a transfer learning approach for generating diverse and high-quality COVID-19 CT images for a target group.

**Latent inversion for stronger preservation.** Some works invert images from the target domain into the latent space of the pre-trained generator to develop a more accurate regularizer for knowledge preservation during fine-tuning. JoJoGAN (Chong & Forsyth, 2022) addresses one-shot image generation using the style space of StyleGAN2. First, GAN inversion is used to find the corresponding style code of the reference image. Then, style mixing is used to generate a set of style codes, and generated images with these styles are used for GAN adaptation. GenOS (Zhang et al., 2022b) includes entity transfer with some related entity mask using an auxiliary network. $D^3$-TGAN (Wu et al., 2023) first inverts each target sample into the latent code space of the source GAN. Then, the maximum mean discrepancy between the features of the source $G$ for inverted code and features of the adapted GAN for a random latent code is used as a regularizer. $F^3$ (Kato et al., 2023b) proposes a faster method for image generation with features of a specific group. First, a GAN inversion of target images is applied, and then PCA is leveraged as a feature extraction strategy to render features of the target group.

**Diverse and emerging regularization ideas.** More recent advances in regularizer-based fine-tuning employ a wide array of diverse and scattered ideas for knowledge preservation. Focusing on a somewhat different goal, FairTL (Teo et al., 2023; 2024a) adopts transfer learning in GANs to train a fair generative model *w.r.t.* a sensitive attribute (SA) using a limited fair dataset. To model complex distributions like ImageNet, IC-GAN (Casanova et al., 2021) learns data distribution as a mixture of conditional distributions. This enables IC-GAN to generate images from unseen distributions, by just changing the conditioning instances on the target samples. KDFSIG (Hou et al., 2022a) exploits the knowledge distillation idea by treating the source model as a teacher and the target model as a student. DOGAN (Hu et al., 2024a) proposes frequency-based segregation of the generation and the discrimination process. FAGAN (Cheng et al., 2024) introduces two frequency regularizers between the source and adapting generator in a one-shot adaptation setup. AnyDoor (Chen et al., 2024a) proposes the use of a discriminative-ID and frequency-aware

extractor to characterize the subject. SmoothSim (Sushko et al., 2023) proposes to preserve the smoothness of the source GAN in target GAN via a regularizer that estimates the Jacobian matrix of the generator in different layers. DPMs-ANT (Wang et al., 2024) uses KL-divergence between the noisy output of the source and fine-tuned diffusion model at time step $t$ (not final output) to prevent overfitting when adapting a diffusion model to a target domain in a few-shot setup. It also introduces an adversarial noise selection mechanism to reduce training iterations. Focusing on efficient transfer learning in diffusion models, TAN (Wang et al., 2023b) proposes a KL divergence minimization between the gap between the source and target domains. To address the overfitting problem caused by multiple denoising steps in diffusion models, they also propose an adaptive noise selection scheme to reduce the number of denoising steps. FDDC (Hu et al., 2023a) gives more weight for content information in early steps of the denoising process by fusing the content from the images generated by the source generator. To address catastrophic forgetting, FPTGAN (Zhang et al., 2023e) proposes a trust-region optimization to smooth the fine-tuning dynamics by adjusting the noise distribution and a SVD-based approach to detect and mitigate mode collapse. StyleDomain (Alanov et al., 2023) analyzes the structure of StyleGAN2 (Karras et al., 2020b) for lightweight adaptation of a pre-trained model to target domains.

**Beyond single-domain adaptation: expansion and hybrid targets.** Finally, several works extend the setting itself. In Nitzan et al. (2023), domain expansion (DE) of Image Generators is studied. Instead of transforming the entire generator from a source domain to a target domain, DE expands the generator to include new data domains, based on their discovery that traversing the latent space of generative models along some directions changes the image significantly while traversing others has no perceptible effect (the dormant direction). HDA (Li et al., 2024c) studies a task beyond domain adaptation to a certain concept, aiming to adapt to a hybrid domain that integrates attributes from several concepts. Specifically, the hybrid domain is represented by the mean embedding of several domains. They utilized the directional loss and distance loss to fine-tune the pre-trained GAN. Def-DINO (Zhou et al., 2024a) suggests that the features learned in the DINO vision transformer (Caron et al., 2021) via self-distillation are more powerful for one-shot adaptation than the features learned by vision-language models like CLIP. In Jiang et al. (2023a), CVD-GAN, a personalized image generation is proposed for color vision deficiency (CVD). SACP (He et al., 2024) trains a translation module to detect the content and style and then adds a regularizer to preserve the content while adapting the style during fine-tuning a pre-trained GAN. In Kim & Chun (2023), DATID-3D is proposed for domain adaptation tailored for 3D generative models. The method leverages text-to-image diffusion models to synthesize diverse images per text prompt. The dataset is refined with their CLIP and pose reconstruction-based filtering process, and the refined dataset is used to fine-tune the 3D generator.

### 5.1.2 Latent Space

MineGAN (Wang et al., 2020b) trains a miner network $M$ during adaptation, to map the latent space $z$ of the source GAN to another space $u = M(z)$ more appropriate for the target domain. MineGAN++ (Wang et al., 2021) extends MineGAN by only updating important parameters. GenDA (Yang et al., 2021b) proposes a lightweight attribute adaptor in the form of scaling and shifting latent codes to adapt the latent space of the source GAN to the target GAN. TF$^2$ (Yu et al., 2024) aligns the latent codes of the defect-free and defect images after denoising to enable robust transfer of the defects to the defect-free images and improve the diversity of the few-shot defect image generation.

LCL (Mondal et al., 2023) freezes $G$ and learns a network to map the latent codes from the $\mathcal{Z}$ space to the extended intermediate space $\mathcal{W}^+$ of a pre-trained StyleGAN2 during adapting GAN. SoLAD (Mondal et al., 2024) introduces a sample-specific latent mapper to transform the sampled latent code before feeding it into the generator. WeditGAN (Duan et al., 2023b) proposes to learn a constant offset parameter ($\Delta w$) for the target domain in the intermediate latent space of StyleGAN2 to relocate source latent codes to the target domain. After fine-tuning the generator to a target domain, SiSTA (Thopalli et al., 2023) perturbs latent representations of the fine-tuned generator that fall below a threshold, either by replacing them with zero or reverting them back to the pre-trained generator's weights. CRDI (Cao & Gong, 2025) avoids the need for fine-tuning or retraining by proposing reconstruction and diversity enhancements to first estimate the generation path to construct the target few-shot samples, while annealing the noise perturbation scheduler for better diversity.

MultiDiffusion (Bar-Tal et al., 2023) freezes the whole parameters of the source diffusion model and optimizes the latent code as a post-processing method to generate the desired output based on a conditioned input. DiS (Everaert et al., 2023) observed that the style of the images generated by Stable Diffusion is tied to the initial latent code. Therefore, they sample noise from the style-specific latent distribution (which is obtained by encoding the target style images to the VAE latent space) and fine-tune Stable Diffusion.

### 5.1.3 Modulation

In signal processing literature, modulation varies some key attributes of a signal to add the desired information to it (Oppenheim et al., 1997). Similarly, in deep neural networks, modulation is used to add some desired information to a base network by adding modulation parameters to the parameters/ features of the base network. AdaFMGAN (Zhao et al., 2020a) shows that layers closer to the sample (earlier layers in $D$, and later layers in $G$) encoder general knowledge. This general knowledge is conceptually shared between source and target domains and aimed to be preserved by Adaptive Filter Modulation which trains a scale and shift parameter for each $k \times k$ kernel. GAN-Memory (Cong et al., 2020), CAM-GAN (Varshney et al., 2021), LFS-GAN (Seo et al., 2023), and CFTS-GAN (Ali et al., 2025) use similar modulation ideas to modulate a pre-trained GAN for generative continual learning. NICE (Ni & Koniusz, 2023) proposed to prevent overfitting by introducing noise into $D$ to modulate its hidden features. The noise is adaptively controlled by the overfitting degree of $D$, which balance the discriminator's discrimination ability and potential overfitting issue. AdAM (Zhao et al., 2022a) and OKM (Zhang et al., 2024f) uses kernel modulation (Abdollahzadeh et al., 2021) for few-shot generative modeling by aiming to preserve the important wights of a pre-trained GAN during a few-shot adaptation to a target domain. HyperDomainNet (Alanov et al., 2022), HyperGAN-CLIP (Anees et al., 2024), DPH (Li et al., 2024d), and DoRM (Wu et al., 2024) adds an additional modulation to StyleGAN2 (Karras et al., 2020b) for adapting to a new domain. Similarly, Mix-of-Show (Gu et al., 2023), Orthogonal Adaptation (Po et al., 2024), DreamMatcher (Nam et al., 2024), PortraitBooth (Peng et al., 2024), DisenDiff (Zhang et al., 2024b), and RealCustom (Huang et al., 2024a) adapt pre-trained diffusion models to learn customized concepts through modulation. $A^3FT$ (Moon et al., 2022) proposes to learn a time-aware adapter (modulation parameters) on top of frozen weights of a pre-trained diffusion model to enable different outputs in different denoising steps when adapting the diffusion model to a target domain with limited data.

### 5.1.4 Adaptation-Aware

Adaptation-aware transfer learning approaches propose that different parts of the knowledge encoded on a pre-trained generative model could be important based on the target domain. AdAM (Zhao et al., 2022a) proposes a probing step before the main adaptation, where the importance of each kernel for adapting a source GAN to the target domain using a few samples is measured using FI. Then, during the main adaptation, the important kernels are preserved using modulation, and the other kernels are simply fine-tuned. OKM (Zhang et al., 2024f) exactly follows AdAM for importance probing. However, in the probing stage, instead of using standard discriminator loss, they use a relaxed version of the Optimal Transport (Villani et al., 2009) idea to measure the distance of the distribution of the generated images from that of the real images. RICK (Zhao et al., 2023a) shows that incompatible knowledge from a source domain to a target domain is related to the kernels with the least importance to this adaptation, and this knowledge can not be removed by simple fine-tuning. Therefore, RICK proposes a dynamic probing schedule during adaptation where it gradually prunes the kernels with the least importance.

### 5.1.5 Natural Language-Guided

Vision-Language models like CLIP (Radford et al., 2021) are usually trained on large-scale image-text pairs and learn to encapsulate the generic information by combining image and text modalities. This generic information is shown to be helpful in various downstream tasks, including zero-shot and few-shot image generation.

**Leveraging offset alignment in CLIP space for zero-shot image generation.** StyleGAN-NADA (NADA) (Gal et al., 2022b) is the first work that leverages CLIP's joint image and text embedding space for

zero-shot image generation. It proposes to use the embedding offset of the textual description in the CLIP space to describe the difference between source and target domains. Specifically, assuming a text prompt $T_s$ that describes the source domain (*e.g.* "Photo" for a StyleGAN2 pre-trained on the FFHQ), and a given $T_t$ (*e.g.* "Fernando Botero Painting"), CLIP's text encoder $E_T$ is used to find the update direction in the embedding space: $\Delta T = E_T(T_t) - E_T(T_s)$. Similarly, the direction of the update/ change for the images can be computed using generated images with source and target generators: $\Delta I = E_I(G_t(z)) - E_I(G_s(z))$, where $E_I$ denotes CLIP's image encoder. By assuming the text offset and image offset are well-aligned in CLIP space, NADA proposes to update the generator's parameters in a way to align $\Delta I$ and $\Delta T$ leading to the directional loss guidance $\mathcal{L}_{directional} = 1 - \Delta I \cdot \Delta T / (|\Delta I||\Delta T|)$. IPL (Guo et al., 2023) points out that adaptation directions in NADA for diverse image samples are computed from one pair of manually designed prompts, which will cause mode collapse. Therefore, they learns a specific prompt for each generated image, and produce different adaptation directions for each sample. Similarly, to prevent mode collapse, SVL (Jeon et al., 2023) uses embedding statistics (mean and variance) for producing adaptation direction instead of only the mean of embeddings in NADA. While these works assumed a perfect alignment between text and image offset, AIR (Liu et al., 2025a) discovered that the offset misalignment exists in the CLIP embedding space and it increases as the source and target domains are more distant. To mitigate the offset misalignment issue, it proposes to sample anchor points closer to the target iteratively during adaptation.

**Extending directional guidance for one-shot setting.** Directional loss proposed in NADA (Gal et al., 2022b) can be easily extended to one-shot image generation, by replacing $\Delta T$ with the direction obtained by target image $I_t$ and a batch of generated images by the source generator: $\Delta I' = E_I(I_t) - \mathbb{E}_i\{E_I(G_s(z_i))\}$, where $\mathbb{E}_i\{E_I(G_s(z_i))\}$ denotes the mean of the CLIP embedding for a batch of generated images. MTG (Zhu et al., 2022c) extends the idea for one-shot image generation by replacing the mean embedding with the projection of the target image on the source generator denoted as $I_s^*$. Specifically, MTG uses GAN inversion to get the corresponding $z^{ref}$ for $I_t$, and uses it to generate the projected image: $I_s^* = G_s(z^{ref})$. HyperDomainNet improves the performance of the NADA and MTG by freezing the weights of the source generator and training modulation weights for the synthesis part inside the generator. DiFa (Zhang et al., 2022a) adds an attentive style loss to directional loss of NADA (Gal et al., 2022b) as a local-level adaptation which aligns the intermediate tokens of the generated image with source and pre-trained GAN. OneCLIP (Kwon & Ye, 2023) exploits the CLIP embedding for three major modules in one-shot learning: i) inverting sample into latent space, ii) preserving the diversity of the GAN during adaptation, and iii) a patch-wise contrastive learning approach for preserving local consistency.

**Language guidance for diffusion control.** The connection between the image embeddings and natural language is also used for adapting generative models to new domains through techniques like diffusion guidance and prompt learning. StyleGAN-Fusion (Song et al., 2024a) aligns a pre-trained StyleGAN2 (Karras et al., 2020b) with target domains by mapping generated images into the latent diffusion space (Rombach et al., 2022) and optimizing based on text-conditioned denoising loss. Similarly, HyperGAN-CLIP (Anees et al., 2024), UniHDA (Li et al., 2024b), and DPH (Li et al., 2024d) leverage CLIP embeddings of image and text to guide domain adaptation, either via hypernetworks that modulate GAN weights or by computing direction-based latent updates. StyO (Li et al., 2024a) targets one-shot face stylization by learning disentangled tokens for source and target styles, along with image-specific tokens to preserve identity. To address fairness in generation, ITI-GEN (Zhang et al., 2023a) and FairQueue (Teo et al., 2024b) propose prompt learning from a limited number of target images to enhance the representation of under-sampled sensitive attributes in text-to-image diffusion models.

**Language-enabled subject-driven generative modeling.** A recent and highly popular research direction involves leveraging the connection between images and natural language to embed a concept directly into the embedding space of image generators (SGM in our proposed GM-DC task taxonomy in Sec. 4.1; this is also called as personalization of generative models in some recent works). DreamBooth (Ruiz et al., 2023) addresses subject-driven sample generation by fine-tuning a text-to-image diffusion model e.g., Imagen (Saharia et al., 2022), or Stable Diffusion (Rombach et al., 2022). Input images are paired with a text prompt that contains a unique identifier and the subject class (e.g., "A [V] dog"), and the pair is used to fine-tune the model. They further propose a class-specific prior preservation regularizer to encourage diversity and to mitigate *language drift*, i.e., the model progressively loses syntactic and semantic knowledge during fine-

tuning. To support continual subject learning without overwriting prior knowledge, LFS-Diffusion (Song et al., 2024b), and L$^2$DM (Sun et al., 2024) introduce lifelong personalization frameworks based on knowledge distillation. Dreambooth-CL (Zhu & Yang, 2024) extends Dreambooth to learn the differences between various input concepts by leveraging a multimodal contrastive learning loss through the CLIP vision encoder. T2IRL (Wei et al., 2025a) proposes the utilization of the diffusion model as a deterministic policy that can be guided by a learnable reward policy for personalized sample generation. Contrary to DreamBooth (Ruiz et al., 2023), AblateConcept (Kumari et al., 2023a) aims to prevent the generation of specific concepts (e.g., copyrighted content) in diffusion models. It achieves this by introducing a KL-divergence loss that aligns the conditional distribution of the target concept with its core class, effectively erasing distinctive details (e.g., converging "A photo of a Grumpy cat" to "A photo of a cat").

**Efficient tuning for SGM.** Instead of fine-tuning the generator, Textual-Inversion (Gal et al., 2022a) optimizes a word vector for the new subject given a few images and uses that word vector for SGM. DBLoRA (Pascual et al., 2024) uses the same framework as DreamBooth, but only updates LoRA weights instead of fine-tuning the whole generator to reduce the computation and memory requirements. It also uses an objective-specific token and a style token to enable more controlled subject-driven generation. HyperDream-Booth (Ruiz et al., 2024) proposes improved memory and time efficiency to DreamBooth (Ruiz et al., 2023). Specifically, it includes a Hyper-network for efficient approximation of the network-weight, followed by fast fine-tuning for better image fidelity. HybridBooth (Guan et al., 2025) combines optimization-based and regression-based methods in a two-stage process, where an initial domain-agnostic word embedding is refined based on the subject image.

**Tuning-free SGM.** A line of research focuses on learning explicit subject representations through encoder-based methods for fast and controllable SGM. BLIP-Diffusion (Li et al., 2023a), SINE (Zhang et al., 2023f), ELITE (Wei et al., 2023), E4T (Gal et al., 2023), SSR-Encoder (Zhang et al., 2024c), and MasterWeaver (Wei et al., 2025b) leverage encoders or mapping networks to extract subject features, enabling efficient generation without iterative token optimization. Expanding beyond single-modal encoders, recent works like MoMA (Song et al., 2025), MultiGen (Wu et al., 2025), and AnomalyDiffusion (Hu et al., 2024b) incorporate multi-modal inputs—such as text, image, and spatial information—to facilitate fine-grained control and broaden the scope of subject-driven generation. In parallel, another line of work seeks to eliminate test-time fine-tuning by directly learning subject-aware modules. InstantBooth (Shi et al., 2023b) achieves this through three auxiliary networks that encode subject identity and inject relevant features into the UNet. IDAdapter (Cui et al., 2024) follows a similar goal using adapter layers and fused facial features, enabling efficient and identity-consistent generation without iterative optimization.

**Improving localization and disentanglement for controllable SGM.** Some of the recent methods aim to improve subject consistency and region-aware control in SGM. FastComposer (Xiao et al., 2024), PortraitBooth (Peng et al., 2024), DreamMatcher (Nam et al., 2024), and DisenDiff (Zhang et al., 2024b) focus on enhancing subject localization through embedding extraction, attention calibration, or alignment of attention maps. In contrast, CII (Jeong et al., 2023) and DETEX (Cai et al., 2024) address subject disentanglement from pose, background, and other semantics, ensuring consistent and controllable generation across diverse scenarios.

**Balancing identity fidelity with prompt controllability.** A key challenge in SGM work is balancing subject fidelity with prompt controllability. Recent works such as Lego (Motamed et al., 2025), CGR (Jin et al., 2025), DreamBlend (Ram et al., 2025), Cross Initialization (Pang et al., 2024), SAG (Chan et al., 2024), ZipLoRA (Shah et al., 2025), RealCustom (Huang et al., 2024a), and PALP (Arar et al., 2024) focus on disentangling subject identity from other semantic attributes while improving alignment with prompts. Building on the idea of leveraging structured semantic knowledge, SuDe (Qiao et al., 2024) models subjects as subclasses of broader categories, capturing both public and subject-specific attributes through reconstruction and semantic constraints. Similarly, ComFusion (Hong et al., 2025) enhances semantic alignment by integrating scene priors and visual-text matching to fuse personalized subject representations with scene-level descriptions.

**Beyond SGM**, other works explore broader personalization tasks. SpecialistDiffusion (Lu et al., 2023) and Domain Gallery (Duan et al., 2024) generalize diffusion models to hard-to-describe or unseen domains,

while ProSpect (Zhang et al., 2023d) and LogoSticker (Zhu et al., 2025) introduce fine-grained control over generation attributes and identity-aware logo insertion, respectively. Multi-subject and multi-concept customization is tackled by Custom Diffusion (Kumari et al., 2023b), along with modular LoRA-based frameworks such as OMG (Kong et al., 2025), TFIC (Li et al., 2025b), MagiCapture (Hyung et al., 2024), Mix-of-Show (Gu et al., 2023), and Orthogonal Adaptation (Po et al., 2024), which collectively balance scalability with fidelity and modular control.

### 5.1.6 Prompt Tuning

VQ-VAEs (Sec. 3.1) can be broadly categorized into two types from the perspective of predicting the latent prior of visual tokens. AutoRegressive (AR) approaches like DALL·E (Ramesh et al., 2021) and VQ-GAN (Esser et al., 2021), learn an AR predictor that follows a raster scan order and predicts the visual tokens from left to right, line-by-line. Non-AutoRegressive (NAR) approaches like DALL·E2 (Ramesh et al., 2022), MaskGIT (Chang et al., 2022), Latent Diffusion (Rombach et al., 2022), or Imagen (Saharia et al., 2022) usually resort to masking techniques (Devlin et al., 2019) to predict the visual tokens in a series of refinement or denoising steps. VPT (Sohn et al., 2023) is the first work that adopts the prompt tuning idea for image generation with generative knowledge transfer. It uses a VQ-VAE framework where a MaskGIT(Chang et al., 2022)/ VQ-GAN(Esser et al., 2021) on the ImageNet dataset (as an example of NAR/ AR approach) is used as a pre-trained network. Then, during adaptation, all the parameters of the VQ Encoder, VQ decoder, and transformer are frozen, and a generator is learned to minimize the adaptation loss by generating and appending a set of visual tokens to the predicted prior. These visual tokens guide the generation process for the target domain by helping the transformer to predict proper tokens to the VQ decoder.

## 5.2 Data Augmentation

Data augmentation increases the quantity and diversity of the training data which is shown to be beneficial for GM-DC. However, if it is not deployed correctly, augmentations can leak into the generator resulting in generating samples with similar augmentations, *e.g.* noisy or rotated images, which is undesirable.

### 5.2.1 Image-Level Augmentation

Early work CR-GAN (Zhang et al., 2020) and bCR (Zhao et al., 2021) apply various transformations on images and enforce the output of the generator to be the same for original and transformed images. Even though not developed specifically for GM-DC, experimental results in Karras et al. (2020a) show that CR-GAN and bCR are beneficial for limited data scenarios. ADA proposes applying the transformations to real and fake images but with a probability $p < 1$. The central design in ADA (Karras et al., 2020a) is that the strength of the augmentation ($p$) is being adapted based on the training dynamics. Specifically, the portion of the real images that get positive output from the discriminator, *i.e.* , $r = \mathbb{E}[Sign(D)]$, is used as an indicator of the discriminator overfitting ($r = 0$ no overfitting, and $r = 1$ complete overfitting). Then, during training, $p$ is adjusted to keep $r$ low. DiffAugment (Zhao et al., 2020b), and IAG (Zhao et al., 2020c) use a similar idea to ADA, but without the adaptive component ($p = 1$). APA (Jiang et al., 2021) uses the same adaptive augmentation mechanism in ADA, but instead of using transformations like rotation, it randomly labels generated images as pseudo-real ones to prevent an overconfident discriminator.

DiffusionGAN (Wang et al., 2023d) applies the gradual diffusion process on real and generated images during training GAN. DANI (Zhang et al., 2024e), similar to DiffusionGAN (Wang et al., 2023d), prevents $D$ from overfitting by injecting noise into images (exactly like the Diffusion process) to augment both real and fake samples. Similar to NICE (Ni & Koniusz, 2023), the noise is adaptive and controlled by the overfitting degree of $D$. Training starts with real and generated images, and each diffusion step is applied after a certain number of training epochs, making the bi-classification task harder for the discriminator to prevent its overfitting.

PatchDiffusion (Wang et al., 2023c) augments the data during training diffusion models by sampling patches with random locations and random sizes alongside the full image and conditioning the denoising score function (Karras et al., 2022) on the patch size and the location information. To address the problem of generating out-of-distribution samples, ANDA (Zhang et al., 2024d) applies negative data augmentation

(Sinha et al., 2021a) to real data to create out-of-distribution samples as fake samples for the discriminator $D$. AugSelf-GAN (Hou et al., 2024) applies self-supervised learning to learn augmentation-aware information.

### 5.2.2 Feature-Level Augmentation

AdvAug (Chen et al., 2021a) computes the adversarial perturbation $\delta$ for the feature maps of the discriminator and generator using the projected gradient descent (Madry et al., 2018). Denoting the discriminator as $D = D_2 \circ D_1$, the adversarial augmentation is applied on the intermediate feature maps ($D_1$) of both real and generated images, resulting in $D_1(x) + \delta$, and $D_1(G(z)) + \delta$. The adversarial loss is then added to the loss function of $D$ during GAN training to maximize the score of the perturbed real image and minimize the score of the perturbed generated image:

$$\mathcal{L}_D^{adv} := \max_{\|\delta\|_\infty < \epsilon} \mathbb{E}_{x \sim p_{data}}[f_D(-D_2(D_1(x) + \delta))] + \max_{\|\hat{\delta}\|_\infty < \epsilon} \mathbb{E}_{z \sim p_z}[f_D(D_2(D_1(G(z)) + \hat{\delta}))] \tag{7}$$

As AdvAug is performed on the feature level, it is shown to be complementary to image-level augmentations like ADA (Karras et al., 2020a) and DiffAug (Zhao et al., 2020b). AFI (Dai et al., 2022) observes a flattening effect in discriminators with multiple output neurons, and takes advantage of this observation by proposing feature interpolation as implicit data augmentation. Meanwhile, inspired by the improved performance of classifiers by debasing them regarding texture, FSMR (Kim et al., 2022a) aims to improve GM-DC by applying a similar idea to GAN's discriminator. It augments the style of each image in the feature space of discriminator and enforces the prediction for these augmented samples to be similar to original sample.

### 5.2.3 Transformation-Driven Design

DAG (Tran et al., 2021) uses a separate discriminator $D_k$ for discriminating real and fake images that are augmented by transformation $T_k$. A weight-sharing mechanism between all discriminators is used to prevent overfitting. Additionally, DAG provides a theoretical ground for training convergence under augmentation. As mentioned in Goodfellow et al. (2014), for an optimal discriminator $D^*$, optimizing $G$ is equivalent to minimizing the Jensen-Shannon (JS) divergence between the real data distribution $P_{data}$ and generated data distribution $P_{model}$, *i.e.* , $JS(P_{data}, P_{model})$. Denoting $P_{data}^T$, and $P_{model}^T$ as the distribution of the real and generated data under augmentation $T$, Tran et al. (2021) shows that JS divergence between two distributions is invariant under differentiable and invertible transformations:

$$JS(P_{data}, P_g) = JS(P_{data}^T, P_g^T) \tag{8}$$

This means that as long as the augmentation is differentiable and invertible, training convergence is guaranteed. SSGAN-LA (Hou et al., 2021) extends DAG by merging all discriminators to a single discriminator and augmenting the label space of the discriminator, *i.e.* , asking $D$ to detect the type of augmentation in addition to conventional real/ fake detection.

## 5.3 Network Architectures

### 5.3.1 Feature Enhancement

FastGAN (Liu et al., 2021) proposes a light-weight GAN structure —shallower $G$ and $D$ compared to SOTA GANs like StyleGAN2— to decrease the risk of overfitting. Inspired by skip connections (He et al., 2016), and squeeze-and-excitation module (Hu et al., 2018), FastGAN fuses features with different resolutions in $G$ through proposed skip-layer excitation modules. An additional reconstruction task is defined for $D$. cF-GAN (Hiruta et al., 2022) is a typical CGAN advanced on FastGAN to perform cGM-1. MoCA (Li et al., 2022b) learns some prototypes for each semantic concept within a domain, *e.g.* , railroad, or sky in a photo of a train. Then, by attending to these prototypes during image generation, some residual feature maps are produced to improve image generation. DFSGAN (Yang et al., 2023a) proposes to preserve the content and layout information in intermediate layers of $G$ by extracting channel-wise and pixel-wise information and using them to scale corresponding feature maps. DM-GAN (Yan et al., 2024) proposes an encoder-decoder-based design for the generator, which consists of CNNs and ViTs for efficiently capturing both

global and local information and enhancing image generation under limited data. FewConv (Liu et al., 2025b) proposes to replace traditional convolutions in GANs with a new design to independently learn the spatial and channel information and therefore reduce the amount of information that needs to be learned for generative modeling. SCHA-VAE (Giannone & Winther, 2022) extend latent variable models for sets to a fully hierarchical approach and propose Set-Context-Hierarchical-Aggregation VAE for few-shot generation.

### 5.3.2 Ensemble Pre-trained Vision Models

ProjectedGAN (Sauer et al., 2021) proposes to project real and generated images into the feature space of a pre-trained vision model to enhance $D$'s performance in discriminating real and fake images by adding two modules. First, the output from multiple layers is used with separate discriminators. Then a *random projection* is used to dilute the features and encourage the discriminator to focus on a subset of the features. SPGAN (Hiruta et al., 2022), built on ProjectedGAN, improves the generation quality of uGM-1 by simply introducing style mapping of StyleGAN and Skip Layer Excitation of FastGAN. Vision-aided GAN (Kumari et al., 2022) uses an ensemble of the original discriminator $D$ and additional discriminators $\{\hat{D}_n\}_{n=1}^N$ to perform the classification task. The additional discriminators $\{\hat{D}_n\}_{n=1}^N$ have a set of pre-trained feature extractors $\mathcal{F} = \{F_n\}_{n=1}^N$ (extracted form pre-trained vision models) with a small trainable head $C_n$ added on top: $\hat{D}_n = F_n \circ C_n$. P2D (Chong et al., 2024), similar to Vision-aided GAN (Kumari et al., 2022), ensembles multiple pre-trained vision models to improve $D$. It proposed to include an R1 regularizer to prevent the classification heads from overfitting. DISP (Mangla et al., 2022) also follows a similar idea to vision-aided GAN to leverage a pre-trained classifier $C$, but it conditions generation by $G$ on the extracted features of a real image with this classifier ($C(x)$), and enforces $G(z|C(x))$ to be similar to input image $x$ in discriminator's feature space.

### 5.3.3 Dynamic Network Architecture

CbC (Shahbazi et al., 2022) shows that under data constraints, where an unconditional GAN can generate satisfactory performance, training the conditional GANs (cGANs) result in mode collapse. To mitigate this issue, CbC (Shahbazi et al., 2022) starts training from an unconditional GAN and slowly transitions to a cGAN using a transition function $0 \leq \lambda_t \leq 1$. Considering the conditioning variable as $c$, this transition is implemented in $G$ as $G(z, c, \lambda_t) = G(S(z) + \lambda_t \cdot E(c))$, with $S$ and $E$ as neural networks that transform the latent code and the conditioning variable. PYP (Li et al., 2024e), using a similar architecture as CbC (Shahbazi et al., 2022), addresses few-shot image generation by generalizing from large pillar datasets during training. Different from CbC, the class embedding is injected into $G$ in parallel to the style code $w$. It further improves the generation diversity by using directional loss.

DynamicD (Yang et al., 2022a) dynamically reduces the capacity of $D$ by randomly sampling a subset of channels of $D$ during each training iteration to prevent overfitting. Inspired by the lottery ticket hypothesis (Frankle & Carbin, 2019), AdvAug (Chen et al., 2021a) and Re-GAN (Saxena et al., 2023) have shown that a much sparse subnetwork of the original generator can be useful for GM-DC. RG-GAN (Saxena et al., 2024) proposed a new weight pruning method. Beyond standard network pruning, to prevent model from over-pruning, a regeneration step is implemented to reintroduce some weights if they gain importance during training. AutoInfoGAN (Shi et al., 2023a) applies a reinforcement learning-based neural architecture search to find the best network architecture for the generator.

## 5.4 Multi-Task Objectives

### 5.4.1 Regularizer

LeCam (Tseng et al., 2021) uses two moving average values to track $D$'s prediction for real and generated images, denoted by $\alpha_R$ and $\alpha_F$, respectively. Then the distance between the $D$'s prediction for real (fake) images and $\alpha_F$ ($\alpha_R$) is decreased by adding a regularizer to prevent overfitting. Analysis in Tseng et al. (2021) shows that this regularizer enforces WGAN (Arjovsky et al., 2017)/ BigGAN(Brock et al., 2019) to minimize the LeCam-divergence which is beneficial for GM-DC. Reg-LA (Hou, 2023) uses a similar idea to regularize the label-augmented GANs discussed in Sec. 5.2. DigGAN (Fang et al., 2022) shows that the discriminator gradient gap between real and generated images increases when training GANs with limited

data, and adds this gap as a regularizer to prevent this behavior. MICGAN (Zhai et al., 2024) observed the mutual information (MI) decay issue in high-resolution uGM-1, therefore, it proposed to explicitly optimize the MI between the features of each layer. CHAin (Ni & Koniusz, 2024) revisits batch normalization (BN) for training GANs under limited data, and suggests replacing the conventional centering of BN with zero-mean regularization and leveraging Lipschitz continuity constraint (Gouk et al., 2021) for the scaling part of the BN. MDL (Kong et al., 2022) addresses the pre-training free few-shot image generation by adding a regularizer that aims to keep the similarities between the latent codes in $\mathcal{Z}$ space and corresponding generated images in image space. DFMGAN (Duan et al., 2023a) proposes the first defect generation approach with limited data. In the first training stage, StyleGAN2-ADA (Karras et al., 2020a) is trained on defect-free images. Then, in the second stage, defect-aware layers are added on top of it to generate the defect masks and fuse the defect features to the main backbone. A variant of mode-seeking loss (Mao et al., 2019) is proposed as a regularizer to encourage the generation of different defects for similar objects.

### 5.4.2 Contrastive Learning

InsGen (Yang et al., 2021a) uses contrastive learning to improve learning $D$ by introducing a pretext task. The pretext task is defined as instance discrimination, meaning that each sample should be mapped to a separate class. This is done by constructing the query and key from the same sample as positive pair, and all remaining images as negative pair. FakeCLR (Li et al., 2022c) analyze three different contrastive learning strategies, namely instance-real, instance-fake, and instance-perturbation. It is shown that instance-perturbation contributes the most improvement in quality and can effectively alleviate the issue of latent space discontinuity. RCL (Gou et al., 2024) follows a similar idea to DCL (Zhao et al., 2022b) but for training a generator from scratch.

**Remark.** As discussed in Sec. 5.1.1, constrastive learning is also used in works like $C^3$ (Lee et al., 2021), DCL (Zhao et al., 2022b), CtlGAN (Wang et al., 2022c), CML-GAN (Phaphuangwittayakul et al., 2022), and IAG (Zhao et al., 2020c) as a regularizer during adapting a pre-trained source generator to the target domain. However, approaches discussed in this section use contrastive learning to train a generative model from scratch using limited data.

### 5.4.3 Masking

MaskedGAN (Huang et al., 2022) utilizes a masking idea for training GANs under limited data by masking both spatial and spectral information. For spatial masking, they use a patch-based mask to enable random masking of all spatial parts. For spectral masking, they mask each frequency channel (extracted by the Fourier transform) based on the amount of information, *i.e.* , channels with more information are more probable to be masked. MaskD (Zhu et al., 2022a) randomly masks feature maps extracted by $D$ for a few-shot setup. DMD (Zhang et al., 2023c) detects that the discriminator slows down learning and applies random masking to its features adaptively to balance its learning pace with the generator.

### 5.4.4 Knowledge Distillation

KD-DLGAN (Cui et al., 2023) proposes a knowledge distillation (KD) (Hinton et al., 2015; Chandrasegaran et al., 2022a) approach by leveraging CLIP (Radford et al., 2021) as the teacher model to distill text-image knowledge to the discriminator. They propose two designs: aggregated generative knowledge designs a harder learning task, and correlated generative knowledge distillation improves the generation diversity by distilling and preserving the diverse image-text correlation from CLIP. BK-SDM (Kim et al., 2025) prunes several residual and attention blocks (manually defined) from the U-Net of Stable Diffusion Rombach et al. (2022) and use feature distillation to compensate for the decline in performance. As discussed before, KDFSIG (Hou et al., 2022a) also uses KD in the context of transfer learning for few-shot image generation.

### 5.4.5 Prototype Learning

Inspired by the success of learning prototypes in few-shot classification, ProtoGAN (Yang et al., 2023b), aims to improve the fidelity and diversity of the FastGAN under limited data (Snell et al., 2017). ProtoGAN has two main modules: prototype alignment for increasing the fidelity of the generated images, and diversity

loss to improve the generation diversity. MoCA (Li et al., 2022b) also learns prototypes but for different semantic concepts through an attend and replace mechanism on the extracted feature maps of $G$.

### 5.4.6 Other Multi-Task Objectives

Gen-Co (Cui et al., 2022) uses multiple discriminators to extract diverse and complementary information from samples. This *co-training* has two major modules: weight-discrepancy co-training, which trains separate $D$s with different weights, and data-discrepancy co-training which in addition to training separate $D$s also uses different information as inputs, *i.e.*, spatial or frequency information. AdaptiveIMLE (Aghabozorgi et al., 2023) proposes an adaptive version of implicit maximum likelihood estimation (Li & Malik, 2018) to improve the mode coverage by assigning different boundary radii for each sample. RS-IMLE (Vashist et al., 2024) also leverages implicit maximum likelihood estimation but for choosing a different prior so that the samples selected for training have a distribution more similar to those sampled at inference. PathcDiffusion (Wang et al., 2023c) and AnyResGAN (Chai et al., 2022) show the effectiveness of *Patch-Level* learning of the generators. Diffusion-GAN (Wang et al., 2023d) leverages the *diffusion process* to improve training GANs by gradually increasing the task hardness for $D$. D2C (Sinha et al., 2021b) uses a DM to improve the sampling process of VAEs by denoising the latent codes and feeding VAE with a clean latent code for sample generation. FSDM (Giannone et al., 2022) uses an attentive conditioning mechanism and aggregates image patch information using a vision transformer for image generation for unseen classes. SpiderGAN (Asokan & Seelamantula, 2023) uses an image dataset as input for training a GAN under limited data instead of using latent codes. It argues that choosing a low-entropy dataset (instead of high-entropy latent codes from a Normal distribution) helps with faster and better training procedures. They further propose a signed Inception Distance metric to select a closer subset of data to the target domain.

### 5.5 Exploiting Frequency Components

Approaches in this category aim to improve frequency awareness to improve GM-DC. FreGAN (Yang et al., 2022c) extracts high-frequency information ($HF$) of images (related to details in images) using Haar Wavelet transform (Porwik & Lisowska, 2004) and uses three different modules to emphasize learning high-frequency information: high-frequency discriminator uses $HF$ as an additional signal to perform real/fake classification, frequency skip connection feeds the $HF$ information of each feature map to the next one in $G$ to prevent frequency loss, and a frequency alignment loss is used to make sure $G$ and $D$ are learning frequency information in the same pace. WaveGAN (Yang et al., 2022b) uses a similar idea, but in a different setup to address the cGM-2 task. SDTM (Yang et al., 2023c) applies Haar Wavelet transformation to decompose features of $D$, encouraging the model to distinguish high-frequency signals of real images from those of generated samples, therefore mitigating the model's frequency bias. Gen-Co (Cui et al., 2022) extracts some frequency information of the image and feeds it to a separate $D$ in addition to using original real and fake images. MaskedGAN (Huang et al., 2022) masks out some frequency bands of the input during training to enforce the generative model to focus more on under-represented frequency bands. FAGAN (Cheng et al., 2024) introduces two frequency regularizers for one-shot adaptation. It aligns low-frequency components of $G_t$ with $G_s$ to preserve general knowledge and matches high-frequency components of the generated image with the reference image to capture domain-specific details.

### 5.6 Meta-Learning

Meta-learning shifts the learning paradigm from data level to task level to capture across-task knowledge as *meta-knowledge*, and then adapt this meta-knowledge to improve the learning process of unseen tasks in the future. A plethora of recent works adopt meta-learning to tackle few-shot classification (Finn et al., 2017; Snell et al., 2017; Vinyals et al., 2016; Sung et al., 2018; Abdollahzadeh et al., 2021) and few-shot semantic segmentation (Wang et al., 2019). These works usually follow the *episodic learning* setup which matches the way that the model is trained and tested. Considering task distribution $P_{\mathcal{T}}$, a set of training tasks are constructed from seen classes $\mathcal{T}^{train} = \{\mathcal{T}_i^{train}\}$, where $\mathcal{T}_i^{train}$ denotes $i^{th}$ training (meta-training) task. The model is trained on the meta-training tasks and later tested on the meta-test tasks $\mathcal{T}^{test} = \{\mathcal{T}_j^{test}\}$ constructed from unseen classes. Usually, meta-training and meta-testing tasks follow the same distribution $P_{\mathcal{T}}$. Similarly, the approaches in this category use meta-learning to address image generation: train a

generative model on a set of few-shot image generation tasks constructed from seen classes of a domain, then test it on the few-shot image generation tasks from unseen classes of the same domain.

### 5.6.1 Optimization

Optimization-based meta-learning algorithms are used in these approaches for learning meta-knowledge. Generative Matching Network (GMN) proposes a similar attention mechanism used in Matching Networks (Vinyals et al., 2016) for few-shot image generation with variational inference. FIGR (Clouâtre & Demers, 2019) meta-trains a GAN using Reptile (Nichol et al., 2018). Training has an inner loop that adapts the GAN weights based on a few-shot image generation task and an outer loop that updates the meta-knowledge using Reptile. Dawson (Liang et al., 2020) modifies the inner loop training to directly get the gradients for the generator from evaluation data. FAML (Phaphuangwittayakul et al., 2021) uses a similar idea to FIGR, but instead of using the standard GAN structure, it uses an encoder-decoder architecture for the generator. CML-GAN (Phaphuangwittayakul et al., 2022) extends FAML (Phaphuangwittayakul et al., 2021) by leveraging contrastive learning to learn quality representations.

### 5.6.2 Fusion

MatchingGAN (Hong et al., 2020a) learns to generate new images for a category by fusing the available images of that category. A set of encoders is used to estimate the similarity between the embedding of the latent code and input images. Then, these similarities are used as interpolation coefficients by an auto-encoder to extract the embeddings of the training images and fuse them for generating new images. F2GAN (Hong et al., 2020b) uses random coefficients for general information, and an attention module for details. The attention module takes the weighted average of the real image features and the corresponding features from the decoder to produce the image details. LoFGAN (Gu et al., 2021) focuses on local features in the fusion process. Given a batch of images, one sample is selected as a base while the remaining are utilized as a reference set. This set acts as a feature bank for the fusing process. F2DGAN (Zhou et al., 2024b) proposed to match the histogram of feature value instead of matching feature value directly in LoFGAN. It encode and reconstruct the features of real samples with a VAE to improve the diversity of fused features. WaveGAN (Yang et al., 2022b) adds frequency awareness to LofGAN by extracting and feeding the frequency components of feature maps to later layers of the generator. SMR-CSL (Xiao et al., 2025) extends LoFGAN by applying a mask to the fused feature during reconstruction. It further introduces a triplet loss to ensure generated images resemble real ones within the same category. This emphasizes category-specific features while enhancing inter-class distinction. AMMGAN (Li et al., 2023b) utilizes an adaptive fusion mechanism for learning pixel-wise metric coefficients during the fusion. EqGAN (Zhou et al., 2023) balances structural and textural information by fusing multi-scale encoder features through a feature equalization module to improve the generation quality. SDTM (Yang et al., 2023c) proposed to improve the generation diversity by introducing out-of-distribution semantic information to the fused features. MVSA-GAN (Chen et al., 2023) proposes a self-attention module alongside a multi-view feature fusion module to capture contextual information within the image and fuse it at the global and local levels for modeling complex scenes. SAGAN (Aldhubri et al., 2024) introduced an attention-based fusion to facilitate optimal integration of features from different encoder-decoder pairs.

### 5.6.3 Transformation

DAGAN (Antoniou et al., 2017) leverages the task of the learning augmentation manifold task in the GAN learning process. This is modeled as some transformations on the input, and these transformations are applied to the new sample from the unseen classes for sample generation. ISSA (Huang et al., 2021) leverages the idea of implicit autoencoders to learn the transformation across datasets using an unsupervised representation in an adversarial manner, while each dataset distribution is learned using implicit distributions. DeltaGAN (Hong et al., 2022a) learns the difference between images (delta) in the feature space, and then uses this delta concept for diverse sample generation. MFH (Xie et al., 2022) aims to learn category-independent and category-related features during episodic training within different categories. Then, the generation network combines these two features to generate diverse images from a single input image. Disco (Hong et al., 2022b) learns a dictionary based on seen images to encode input images into visual tokens. These tokens are then

fed into the decoder with the style embedding of seen images to generate images from unseen classes. AGE (Ding et al., 2022) uses GAN inversion to invert the samples of a category to $\mathcal{W}^+$ space of StyleGAN2 (Karras et al., 2020b). The mean latent code for all samples of a category is used as a prototype, and all differences are considered as general attributes. These attributes are then used to diversify sample generation for unseen classes. TAGE (Zhang et al., 2024a) extends AGE (Ding et al., 2022) by learning a codebook to store category-agnostic features and using it to augment real samples through editing. SAGE (Ding et al., 2023) addresses the class inconsistency in AGE by taking all given samples from unseen classes into account during inference. HAE (Li et al., 2022a) uses a similar idea to AGE (Ding et al., 2022), but hyperbolic space instead of using Euclidean distance, which allows more semantic diversity control. LSO (Zheng et al., 2023) finds a prototype for each class similar to AGE (Ding et al., 2022). Then it adjusts the GAN to produce similar images to target samples using latent samples from the neighborhood of the prototype, followed by updating the prototype in latent space using the adapted GAN. CDM (Gupta et al., 2024) models the distribution of unseen classes in the latent space of Stable Diffusion (Rombach et al., 2022) by leveraging the latent code of the most similar seen classes, it is then used as a conditional input to Stable Diffusion.

### 5.7 Modeling Internal Patch Distribution

### 5.7.1 Progressive Training

SinGAN (Shaham et al., 2019) is the pioneering work that makes use of the internal distribution of the patches within an image to train a generative model. It trains a pyramid of generators $\{G_0, \ldots, G_N\}$ against a pyramid of real images $\{x_0, \ldots, x_N\}$, where $x_n$ is a downsampled version of input image $x$ by a factor of $r^n$. The generator at scale $n$ uses random noise $z_n$ and an upsampled version of the generated image from the lower resolution $\tilde{x}_{n+1}$ as input: $\tilde{x}_n = G_n(z_n, (\tilde{x}_{n+1}) \uparrow^{r_n})$. Similarly, a pyramid of discriminators is used where $D_n$ compares the $\tilde{x}_n$ and $x_n$ in patch-level for real-fake classification. CCASinGAN (Wang et al., 2022b) improves SinGAN by introducing a network block to aggregate global image features, therefore avoiding the training being affected by the outliers in a single image. ConSinGAN (Hinz et al., 2021) stacks the new layers for a bigger scale on top of the previous layers used for a smaller scale instead of using separate generators for each scale. SD-SGAN (Yildiz et al., 2024) enhances SinGAN by adding self-attention for global semantic control and DenseNet blocks to reduce computation while maximizing information flow. SA-SinGAN (Chen et al., 2021b) and TcGAN (Jiang et al., 2023b) use a self-attention mechanism to enable modeling long-range correlations and local information for modeling internal patch distribution. RecurrentSinGAN (He & Fu, 2021) observes kernel similarities across SinGAN's multi-scale generators and replaces them with a single recurrent generator to share parameters across scales. BlendGAN (Kligvasser et al., 2022) and DEff-GAN (Kumar & Sivakumar, 2023) extend previous approaches for learning the internal distribution for $k$ images, thereby allowing for the potential mixing of different image semantics and improving diversity. ExSinGAN (Zhang et al., 2021) compose three modular GANs to learn the structure, semantics, and texture of the internal paths within a single image in a successive manner.

SinDDM (Kulikov et al., 2023) applies the same idea of SinGAN but uses diffusion models with a fully convolutional lightweight denoiser. PromptSDM (Park et al., 2024) enhances SinDDM by incorporating text cross-attention into the diffusion model, where text inputs are generated by captioning blurred images. SinDiffusion (Wang et al., 2022a) addresses artifacts in SinGAN due to progressive resolution growth by applying progressive denoising using a diffusion model architecture. LatentSDM (Han et al., 2024b) accelerates inference by modeling internal patch distribution with a Stable Diffusion (Rombach et al., 2022), where the latent space is obtained by the fused embedding of a VAE and VQ-VAE trained by the given single image.

### 5.7.2 Non-Progressive Training

One-Shot GAN (Sushko et al., 2021) uses a standard generator (single-scale), but multiple paths for the discriminator to enforce learning objects' appearance and how to combine them. Within the discriminator the low-level loss is defined on low-level features, and two different losses are defined to learn the content and the layout in image patches. PetsGAN (Zhang et al., 2022c) utilizes the semantic variation in GAN latent space through GAN inversion to enable large variations in the layout generation. SinFusion (Nikankin et al., 2023) explores learning the internal patch distribution from both single images and videos. SinFusion

extends on DPPM (Ho et al., 2020) and reduces the size of the receptive fields by first removing attention layers, then adopting ConvNext (Liu et al., 2022) blocks in the U-Net (Ronneberger et al., 2015) architecture. To reconstruct videos, a series of images is fed into a series of 3 identical models. The first model predicts the next frame; the second model denoises and removes small artifacts from the generated images; the last model interpolates between the different frames.

## 6 Critical Analysis and Empirical Comparison

### 6.1 Critical Analysis and Design Principles

In this section, we summarize, for each class of approaches, the key factors contributing to their success or failure, their fundamental limitations, and the deeper design principles that emerge from these observations.

#### 6.1.1 Transfer Learning

*Regularizer-based fine-tuning* remains one of the most widely used strategies for leveraging transfer learning in GM-DC. These approaches tend to succeed when the source and target domains are closely related, because the regularizer constrains optimization to preserve useful semantic and structural priors for stabilizing adaptation. However, they often fail when the domain gap becomes large: the same constraint that stabilizes learning in similar domains can over-restrict adaptation in dissimilar ones, causing the generator to retain source-specific biases that are incompatible with the target distribution and hindering the learning of new features. This exposes a fundamental limitation: regularization strength inherently trades off between knowledge preservation and domain adaptability. Determining the optimal level of regularization requires estimating the transfer distance between domains, which is particularly difficult under limited data. A deeper design principle arising from this insight is that successful GM-DC methods should employ adaptive or data-aware regularization schemes that dynamically balance the reuse of transferable priors with the flexibility needed to capture novel characteristics of the target domain.

*Latent-space* methods provide a lightweight and parameter-efficient strategy for GM-DC. These approaches succeed when adaptation occurs within a compact and well-structured latent representation, allowing the model to reuse the pretrained backbone while updating only a small set of parameters. However, they often fail when the semantic gap between source and target domains becomes large. In such cases, the latent space may lack sufficient expressiveness to represent new concepts, leading to misalignment or entanglement of factors that distort the generated outputs. This highlights a fundamental limitation: adaptation performance is highly dependent on the quality of the pretrained latent representation. A deeper design principle emerging from this insight is that effective latent-space adaptation should include mechanisms for latent disentanglement and representation expansion, maintaining efficiency while enabling the model to flexibly capture novel characteristics of the target domain.

*Modulation-based* approaches have emerged as a practical middle ground for GM-DC. They succeed because they inject new information through lightweight modulation layers or parameters, effectively "writing" new concepts without overwriting pretrained base knowledge. However, they often fail when the required transformation between source and target domains is large or complex, as the limited modulation capacity may underfit distant domain adaptations. Performance also depends strongly on the granularity and location of modulation: inappropriate layer selection can lead to insufficient adaptation or degradation of important pretrained representations. This highlights a fundamental limitation: the method's success is tied to the hierarchical representation of the specific model and is sensitive to how modulation interacts with that hierarchy. The deeper design principle is that effective modulation-based adaptation must align modulation granularity and placement with the model's semantic representation hierarchy, ensuring sufficient modulation capacity to inject new concepts effectively while preserving pretrained knowledge.

*Adaptation-aware* methods provide a systematic and data-driven mechanism for selecting and preserving transferable knowledge in GM-DC. They succeed because they explicitly measure the importance or compatibility of model components, such as convolutional kernels or attention heads, often using criteria like Fisher Information. By freezing or pruning components of different importance, these methods retain useful pretrained knowledge while reducing negative transfer from irrelevant or harmful features. However, they

often fail when the importance estimation are unreliable—especially under limited data—leading to the removal of components that are crucial for representing target-domain knowledge. This highlights a fundamental limitation: the quality of adaptation depends on the accuracy and stability of importance estimation under data scarcity. The deeper design principle is that effective adaptation-aware strategies should combine robust importance estimation with dynamic update mechanisms, allowing the model to preserve and refine relevant components during adaptation.

*Natural language-guided* approaches have shown strong performance in zero-shot generative modeling and personalization, especially in diffusion-based models where textual prompts enable flexible concept transfer with minimal training data. They succeed because they leverage the rich alignment between text and visual representations learned by large multimodal foundation models such as CLIP. However, these methods often fail when the textual encoder or base model has inadequate instruction-following ability or weak language grounding, leading to unstable or inaccurate generations. They can also amplify or transfer undesired semantic biases inherited from the pretrained text–image space. This highlights a fundamental limitation: performance depends heavily on the alignment quality and bias characteristics of the underlying multimodal model. The deeper design principle is that effective language-guided transfer requires robust semantic alignment and a reliable, bias-aware base model to ensure that linguistic guidance remains precise, consistent, and faithful to the intended target concept.

*Visual prompt-tuning* provides a highly parameter-efficient strategy for GM-DC and effectively mitigates catastrophic forgetting by keeping the pretrained backbone frozen. These methods succeed when the base model already encodes concepts and structures that overlap with the target domain, allowing lightweight visual tokens or prompts to guide generation without extensive retraining. However, they often fail under large domain shifts or fine-grained structural variations, where the limited capacity of guidance tokens cannot adequately steer the generation process toward new or complex target features. This highlights a fundamental limitation: visual prompts operate within the representational bounds of the pretrained model and have limited ability to extend its conceptual space. The deeper design principle is that effective visual prompt-tuning should incorporate mechanisms that enhance steering capacity while preserving the efficiency and stability benefits of frozen-backbone adaptation.

### 6.1.2 Data Augmentation

*Image-level augmentation* methods improve GM-DC by expanding the coverage of the data distribution through transformations applied directly to the image space. They succeed because augmentations such as flips, color jittering, and affine changes prevent the model from memorizing the limited training samples and enhance the data efficiency of generative models. However, they can fail when the applied transformations alter the semantic meaning of images. This mismatch leads to undesired invariance that weakens feature representation and ultimately affects the generator's ability to model realistic variations. The fundamental limitation is their lack of semantic awareness: augmentations operate purely on pixel-level changes without considering their effect on class or concept meaning. The deeper design principle is that effective augmentation should preserve semantic consistency, using adaptive or learned strategies that respect the underlying data manifold.

*Feature-Level Augmentation* enhance GM-DC by applying transformations in the feature space rather than directly on images. They succeed because the feature manifold, being smoother and more compact than the raw image space, allows meaningful interpolation and mixing that improves regularization and generalization. By combining or interpolating features, these methods encourage invariance to stylistic variations and smooth the training landscape, helping the model learn more robust representations from limited data. However, their effectiveness depends strongly on the quality of the embedding space: if the features are poorly structured or lack semantic alignment, the augmented samples may lose meaning or introduce artifacts. The fundamental limitation is their reliance on the semantic quality and geometry of the learned feature space. The deeper design principle is that successful feature-level augmentation requires well-organized, semantically meaningful embeddings where interpolation preserves semantic consistency.

*Transformation-driven design* approaches leverage knowledge of each individual transformation to construct augmentation strategies that enhance learning without corrupting the true data distribution. They succeed

because, unlike classical augmentation that can mislead the generator into modeling transformed data (e.g., rotated or flipped versions) instead of the original domain, these methods explicitly account for how each transformation affects the data and adjust the training objective accordingly. This enables the generator to benefit from augmented diversity while still learning the correct underlying distribution. However, their success depends heavily on the careful design and calibration of transformation types. They also introduce multiple hyperparameters that require extensive tuning for stability. The fundamental limitation lies in their design complexity and sensitivity to transformation choice. The deeper design principle is that effective transformation-driven augmentation should be guided by a clear understanding of how each transformation interacts with the generative process, balancing diversity enrichment with faithful distribution learning.

### 6.1.3 Network Architecture

*Feature enhancement* approaches aim to improve GM-DC by designing dedicated modules that strengthen or preserve specific features often overlooked by standard training objectives. They succeed because emphasizing such features, such as texture, edge, or structural cues, helps the model capture fine-grained details that would otherwise be lost when data diversity is low. This selective enhancement improves visual fidelity and stability in generation. However, these methods often fail to generalize across diverse domains because the enhanced features are typically task or dataset specific; what benefits one type of data, such as faces, may hinder another, such as natural scenes. This highlights a fundamental limitation: the effectiveness of feature enhancement depends heavily on correctly identifying which features to emphasize and where to apply them within the network, which is difficult to determine consistently across domains. The deeper design principle is that successful feature enhancement should balance specificity and generality.

*Ensemble pre-trained vision model* approaches enhance GMDC by leveraging multiple pre-trained vision networks to strengthen the discriminator with rich, complementary semantic representations. They succeed because these pre-trained models, each specialized in learning specific visual or semantic cues, provide a robust supervisory signal that stabilizes adversarial training and improves the generator's ability to capture fine-grained structures. However, these methods often face practical and conceptual challenges when applied in data-constrained settings. Each pre-trained network requires its own set of trainable layers to align with the generative task, which significantly increases computational and data demands. Moreover, there is no principled way to evaluate or rank pre-trained vision models based on their suitability for a given target domain, leading to potential inefficiencies or suboptimal combinations. This reveals a fundamental limitation: ensemble-based enhancement depends on the compatibility and effective integration of heterogeneous pretrained features within the generative framework. The deeper design principle is that successful ensemble integration requires adaptive selection and alignment mechanisms that automatically identify and fuse the most relevant pretrained representations.

*Dynamic network architecture* approaches adapt the model structure during training to better capture the underlying data distribution. They succeed because dynamically evolving architectures can balance model capacity and data complexity, helping prevent both overfitting and underfitting while promoting efficient knowledge transfer. However, they sometime suffer from instability, as adding or removing layers can disrupt previously learned representations. The fundamental limitation lies in the trade-off between adaptive flexibility and training stability. The deeper design principle is that effective dynamic architectures should incorporate mechanisms that preserve learned knowledge during structural changes and regulate capacity growth in a stable manner.

### 6.1.4 Multi-Task Objective

*Regularizer* approaches address undesirable behaviors that arises when generative models are trained with limited data by introducing additional loss terms that constrain or guide the learning process. They succeed because these regularizers can effectively suppress issues such as overfitting, mode collapse, or distributional drift without adding new parameters or modifying the network architecture, making them computationally lightweight and easy to integrate. However, they often fail to generalize across different architectures or generative paradigms. Moreover, many of these regularizers are tailored to GAN-based frameworks and do not directly translate to diffusion or autoregressive models, limiting their broader applicability. This highlights a fundamental limitation: the effectiveness of regularization depends heavily on the model's internal

design and objective formulation. The deeper design principle is that successful regularizer design should be model-aware and grounded in the underlying training dynamics, ensuring adaptability across architectures while preserving stability and generalization in low-data generative modeling.

*Contrastive learning* approaches enhance generative modeling under limited data by introducing self-supervised pretext tasks that do not require labeled samples. They succeed because contrastive objectives encourage the model to learn discriminative yet semantically meaningful representations, providing an auxiliary supervisory signal that improves the quality and robustness of generated samples. By aligning representations of similar samples and separating dissimilar ones, these methods strengthen the discriminator's feature space and improve the generator's ability to capture fine-grained variations. However, current approaches are fundamentally tied to the adversarial framework, as the contrastive loss is typically applied within the discriminator. This restricts their applicability to GAN-based models and prevents straightforward extension to architectures such as VQ-VAE or diffusion models, which lack an explicit discriminator. The core limitation lies in the architectural dependency of contrastive learning, which confines its potential to a subset of generative paradigms. The deeper design principle is that effective integration of contrastive learning into generative modeling requires rethinking the placement of the contrastive objective, developing architecture-agnostic formulations that preserve its representational benefits while remaining compatible with non-adversarial generative frameworks.

*Masking* approaches improve generative modeling under limited data by intentionally increasing learning difficulty, which forces the model to capture more generalizable and semantically meaningful representations rather than memorizing training samples. They succeed because masking compels the model to infer missing information from contextual cues, strengthening its ability to model global dependencies and enhancing robustness to data scarcity. However, they often fail when the masking strategy is poorly designed. An inappropriate masking ratio can make the task too trivial, allowing shortcut learning, or too difficult, causing optimization instability and degraded generation quality. In addition, misguided choices about what and how to mask may bias the model toward uninformative regions or hide critical semantic content, limiting its capacity to learn coherent structure. The fundamental limitation is the sensitivity of masking-based learning to both the ratio and spatial distribution of masked regions. The deeper design principle is that effective masking should adapt to the semantic structure of the data and balance task difficulty with information retention so that the model learns representations that generalize beyond the observed samples.

*Knowledge distillation* enhances generative modeling with limited data by transferring knowledge from a large, well-trained teacher model to a smaller student model. It succeeds because the teacher provides rich supervisory signals that capture nuanced data statistics and semantic structure, allowing the student model to generalize better even under data scarcity. However, these methods often fail when the distribution of the target data differs significantly from that of the teacher's training data, which leads to inaccurate or misleading feedback that can destabilize learning. The fundamental limitation lies in the dependency on the teacher model's domain alignment. The deeper design principle is that effective knowledge distillation for generative modeling should employ adaptive teacher–student interaction, where the teacher selectively guides the student based on relevance to the target data.

*Prototype learning* enhances generative modeling under limited data by introducing a clustering-based auxiliary task that groups samples according to shared visual or semantic attributes. It succeeds because these prototypes provide additional structural supervision, encouraging the model to learn more coherent and discriminative representations that capture the underlying data manifold. However, the effectiveness of this approach is highly dependent on how prototypes are defined. Coarse definitions, such as categorizing samples simply as real versus fake, offer little additional information, while overly fine-grained or poorly separated prototypes can introduce noise and ambiguity, especially in datasets with diverse or overlapping categories. This reveals a fundamental limitation: prototype learning is sensitive to the granularity and semantic consistency of the prototype definitions, which may not always align with the true data distribution. The deeper design principle is that effective prototype learning should employ adaptive, semantically grounded prototype formation that balances abstraction and specificity.

### 6.1.5 Exploiting Frequency Components

Approaches based on frequency improve GM-DC by explicitly using frequency information, especially high frequency signals that capture fine details such as edges and textures. They succeed because they counter the common tendency of generative models to favor smooth low frequency content and to ignore detailed patterns. By encouraging awareness of both low and high frequency components, these methods improve image fidelity and reduce artifacts, producing more realistic synthesis. However, they can fail when the dataset is extremely small or highly varied, for example only dozens of images with diverse content. In such cases, the learned frequency cues may not generalize, and performance can degrade further when the data are imbalanced or long-tailed. The fundamental limitation is that current works usually rely on simple frequency extraction approaches such as DCT or Haar wavelets, while more advanced representations like multi-wavelet or wavelet packet transforms have been shown to capture frequency information with higher expressiveness and accuracy. This reliance on limited frequency encoders constrains the model's ability to represent complex spatial variations and fine-grained details across diverse data. The deeper design principle is that future frequency-based methods should adopt richer, multi-resolution frequency representations that can capture both global structure and local details, enabling more accurate and semantically coherent image synthesis under limited data conditions.

### 6.1.6 Meta-Learning

*Optimization-based* methods aim to enhance the adaptability of GMDC by using meta-learning strategies that enable quick convergence to new domains with minimal supervision. They succeed because meta-learning provides well-conditioned initialization states for the generator, allowing efficient adaptation to unseen categories without extensive retraining. However, these methods often fail to synthesize realistic and detailed images, as the generated outputs tend to appear blurry or lack coherent structure, as the meta-objective prioritizes fast convergence over capturing high-frequency details and complex semantics. The fundamental limitation lies in their dependence on the trade-off between rapid adaptation and visual fidelity. The deeper design principle is that effective optimization-based transfer should integrate meta-learning with stable and perceptually aligned training objectives to ensure that the generator maintains both adaptability and realism when generalizing to unseen domains with limited data.

*Transformation-based* approaches aim to enhance GM-DC by learning transferable transformation patterns that capture how visual attributes vary within or across categories, and then applying these learned transformations to synthesize novel samples for unseen classes. They succeed because they model intra- and inter-category variations directly in the data space, allowing the generator to reuse structural and appearance-level transformations learned from seen classes. However, they sometimes fail when applied to unseen categories, as the learned transformations may not generalize due to the complex and inconsistent relationships between intra- and inter-category variations. This mismatch can lead to unrealistic or semantically incorrect generations, where the synthesized images lose the distinctive characteristics of the unseen category. A fundamental limitation lies in the assumption that the editing direction for a given attribute is universally shared across categories, which rarely holds true in practice. The deeper design principle is that effective transformation-based learning should incorporate category-aware and context-sensitive modeling of transformations, enabling adaptation to unseen domains while preserving both structural coherence and category identity.

*Fusion-based* approaches aim to improve GM-DC by learning to combine or interpolate feature representations from seen classes to generate samples for unseen ones. They succeed because they exploit shared structures and compositional patterns across categories, allowing the model to synthesize novel data by reusing and blending meaningful visual features learned from rich source domains. However, these methods often fail to produce diverse and high-fidelity images, as the fusion process, typically based on simple feature matching or attention-driven interpolation, can blur distinct attributes and introduce visual artifacts. This limitation arises from their reliance on high similarity among input features and the lack of semantic understanding during fusion, which leads to outputs that are overly smooth or too similar to the conditioning examples. The fundamental limitation lies in their shallow fusion strategy, which captures correlations but not causal or hierarchical relationships among features. The deeper design principle is that effective fusion-based generation requires semantically guided and hierarchically structured fusion mechanisms that

preserve both shared and distinctive features, enabling richer diversity and sharper visual detail in low-data generative modeling.

### 6.1.7 Modeling Internal Patch Distribution

*Progressive Training* methods aim to learn the internal patch distribution of a single image by training a generative model in multiple stages, each responsible for capturing structure at different scales or noise levels. Approaches such as SinGAN achieve impressive results by leveraging self-similarity within an image to generate diverse yet coherent samples that preserve both global layout and fine textures, even from a single example. They succeed because this multi-scale learning effectively prevents overfitting and memorization, allowing realistic variations without requiring multiple training samples. However, these models often fail to capture high-level semantics or meaningful spatial relationships, resulting in incoherent arrangements or distorted objects when generating complex scenes. Their progressive layer-freezing strategy is time-consuming and easy to cause artifacts accumulation. The fundamental limitation lies in the lack of semantic understanding and restricted joint learning across scales. The deeper design principle is that future progressive training frameworks should integrate semantic and structural awareness with multi-scale learning, enabling generative models to achieve both texture fidelity and holistic scene coherence under data-constrained conditions.

*Non-progressive Training* methods train a generative model directly at a fixed scale, often by modifying the model architecture or leveraging a small set of correlated observations that capture natural variations in viewpoint, pose, and texture. These approaches succeed by exploiting the inherent diversity present within such correlated samples, effectively expanding the training distribution without the need for explicit data augmentation. This allows the model to achieve richer and more varied generations, even under one-shot or few-shot conditions. However, they often fail when the available samples exhibit significant spatial inconsistencies or complex non-rigid changes, which can lead to fragmented or incoherent object synthesis. These failures arise because the model lacks semantic understanding—it can capture local correlations but struggles to maintain global structure or coherence across variations. The fundamental limitation is that learning remains primarily appearance-driven, without explicit mechanisms to disentangle motion, shape, or identity. The deeper design principle is that effective non-progressive training should incorporate semantic and structure-aware priors that preserve spatial and conceptual consistency while exploiting natural intra-sample diversity, ensuring both fidelity and coherence in generative modeling under data-limited settings.

## 6.2 Empirical Comparison

In this section, we conduct experimental comparsions across tasks. We provide a comprehensive quantitative comparison of representative methods for a number of GM-DC scenarios. We discuss these scenarios because they attract substantial attention or they capture rapidly growing interest. The visual results were produced using either the official implementations or faithful reproductions that strictly follow the methodological details described in the original papers.

Table 5: FID comparisons of representative methods for task uGM-1. Training datasets are 100-shot Obama/Panda/Grumpy Cat images, and Animal-Face (160 cats and 389 dogs) images.

| Method | Obama | Grumpy Cat | Panda | AFHQ-Cat | AFHQ-Dog |
|---|---|---|---|---|---|
| ADA | 45.69 | 26.62 | 12.90 | 40.77 | 56.83 |
| LeCam | 33.16 | 24.93 | 10.16 | 34.18 | 54.88 |
| GenCo | 32.21 | 17.79 | 9.49 | 30.89 | 49.63 |
| InsGen | 32.42 | 22.01 | 9.85 | 33.01 | 44.93 |
| Diffusion-GAN | 28.55 | 21.87 | 8.69 | 33.18 | 68.15 |
| FakeCLR | 26.95 | 19.56 | 8.42 | 26.34 | 42.02 |
| NICE | 20.09 | 15.63 | 8.18 | 22.70 | 28.65 |
| DANI | 10.08 | 14.92 | 3.04 | 17.72 | 16.81 |

Table 6: Quantitative comparisons of representative methods for task uGM-2. The source model is StyleGAN-2 pretrained on FFHQ for both settings.

| Method | FFHQ-Baby | | AFHQ-Cat | |
|---|---|---|---|---|
| | FID ($\downarrow$) | Intra-LPIPS ($\uparrow$) | FID ($\downarrow$) | Intra-LPIPS ($\uparrow$) |
| TGAN | 101.58 | 0.517 | 64.68 | 0.490 |
| FreezeD | 96.25 | 0.518 | 63.60 | 0.492 |
| EWC | 79.93 | 0.521 | 74.61 | 0.587 |
| CDC | 69.13 | 0.578 | 176.21 | **0.629** |
| DCL | 56.48 | 0.580 | 156.82 | 0.616 |
| AdAM | 48.83 | 0.590 | 58.07 | 0.557 |
| RICK | **39.39** | **0.608** | **53.27** | 0.569 |

**uGM-1.** Tab 5 presents the FID comparison results for task uGM-1. Across all datasets, DANI achieves the best scores, substantially outperforming earlier methods such as ADA, LeCam, and GenCo. This demonstrates DANI's superior ability to generate high-quality and diverse images through more effective normalization and adaptation strategies. In contrast, early approaches like ADA and LeCam produce higher FIDs

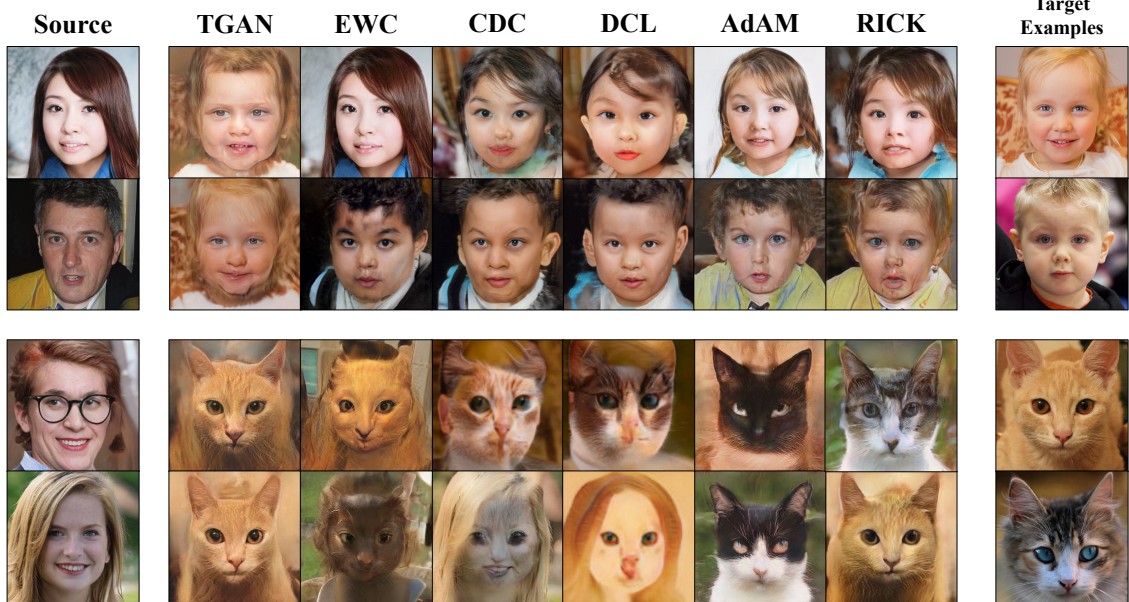

Figure 8: Qualitative comparisons of uGM-2 methods. The source model is StyleGAN-2 pretrained on FFHQ for both settings, and the target domains are 10-shot FFHQ-Baby and 10-shot AFHQ-Cat respectively.

due to limited feature fusion capability and training instability. Intermediate methods such as FakeCLR and NICE benefit from enhanced contrastive learning and normalization, achieving lower FID values but still falling short of DANI, particularly on challenging datasets like AFHQ-Dog.

**uGM-2.** The results in Tab. 6 and Fig. 8 show a clear progression from early transfer-based approaches to more adaptation-aware methods. RICK achieves the best overall performance on both target domains, with the lowest FID and highest Intra-LPIPS, indicating superior adaptation and diversity. Earlier approaches such as TGAN and FreezeD perform worse because they rely heavily on fixed discriminator transfer and lack mechanisms to preserve domain-specific knowledge during adaptation. EWC stabilizes learning by constraining parameter changes but can limit adaptation when domain gaps are large, as reflected in its uneven FID results. CDC and DCL improve diversity through correspondence and contrastive learning but are sensitive to similarity metrics and training stability. AdAM bridges these limitations through adaptive module selection, and RICK refines this by prioritizing useful kernels during adaptation, preventing negative transfer and yielding the best trade-off between fidelity and diversity.

Table 7: Quantitative comparisons of representative methods for task uGM-3. The source model is StyleGAN-2 pretrained on FFHQ for both settings.

| Method | Human → Werewolf | | Photo → Sketch | |
|---|---|---|---|---|
| | CLIP-I (↑) | Intra-LPIPS (↑) | CLIP-I (↑) | Intra-LPIPS (↑) |
| NADA | 0.653 | 0.430 | 0.639 | 0.419 |
| IPL | 0.680 | 0.439 | 0.600 | 0.429 |
| SVL | 0.600 | 0.431 | 0.591 | **0.448** |
| AIR | **0.757** | **0.441** | **0.687** | 0.426 |

Table 8: Quantitative comparisons of representative methods for task cGM-1. Experimental setting follows CbC to use a subset of the datasets with 20 classes.

| Method | Food101 | AFHQ |
|---|---|---|
| ADA | 111.65 | 90.11 |
| DiffAug | 28.70 | 25.09 |
| CbC | **20.12** | **16.25** |

**uGM-3.** In the comparison among uGM-3 methods in Tab. 7, AIR achieves the highest CLIP-I and competitive Intra-LPIPS scores, demonstrating strong semantic alignment and image diversity. NADA provides an effective baseline using CLIP-guided text–image offsets but struggles with mode collapse due to reliance on single-prompt direction. IPL and SVL mitigate this issue by introducing sample-specific or distributional guidance, improving diversity (higher Intra-LPIPS), though their assumptions about perfect alignment in CLIP space limit performance on distant domains. AIR further advances this by explicitly

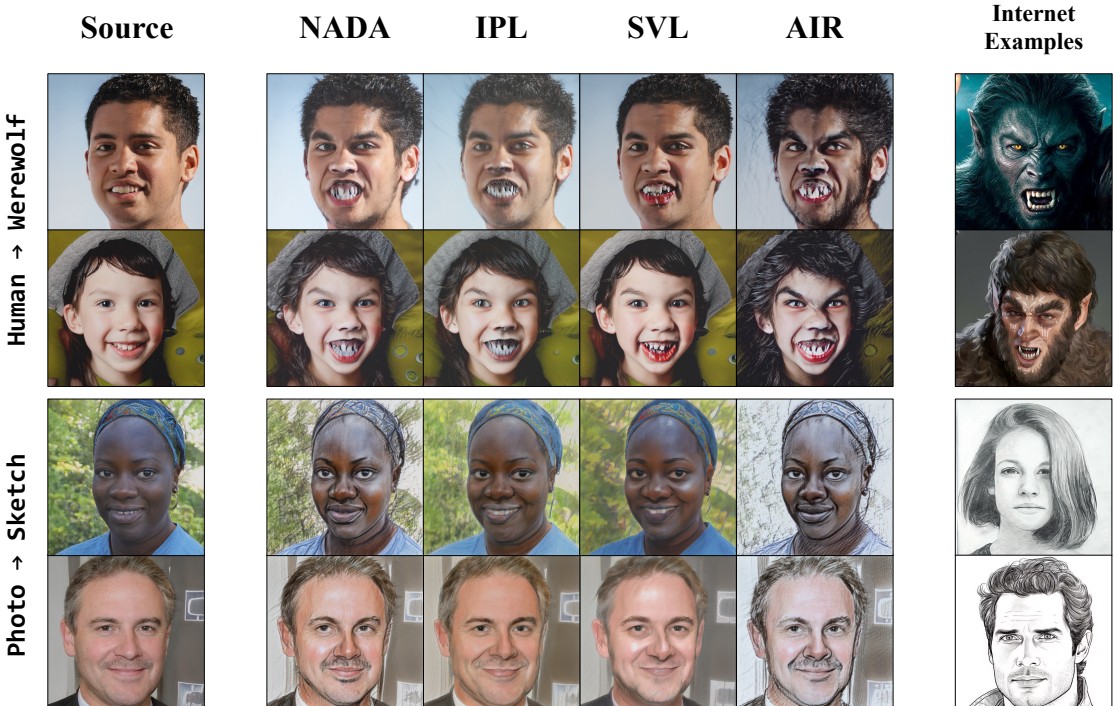

Figure 9: Qualitative comparisons of uGM-3 methods. The source model is StyleGAN-2 pretrained on FFHQ for both settings.

addressing offset misalignment through iterative anchor sampling, achieving the most consistent results across both adaptation tasks.

**cGM-1.** Tab 8 reports the results for task cGM-1. CbC achieves the lowest FID on both Food101 and AFHQ datasets, outperforming ADA and DiffAug. This demonstrates that class-based contrastive strategies are particularly effective in improving generative quality and inter-class separability. While DiffAug improves stability compared to ADA, it still faces challenges with augmentation leakage, and ADA performs worst under constrained data due to overfitting tendencies.

Table 9: Quantitative comparisons of representative methods for task cGM-2.

| Method | Flowers | | AFHQ | |
|---|---|---|---|---|
| | FID ($\downarrow$) | LPIPS ($\uparrow$) | FID ($\downarrow$) | LPIPS ($\uparrow$) |
| FIGR | 190.12 | 0.063 | 211.54 | 0.076 |
| MatchingGAN | 143.35 | 0.163 | 148.52 | 0.151 |
| LoFGAN | 78.83 | 0.387 | 113.01 | 0.489 |
| WaveGAN | 42.17 | 0.399 | 30.35 | 0.508 |
| AGE | 45.96 | 0.431 | 28.04 | **0.558** |
| F2DGAN | **38.26** | **0.433** | **25.24** | 0.546 |

Table 10: FID comparisons of representative methods for task cGM-3. The experimental setting follows that in VPT.

| Method | C101 | Flowers | DTD |
|---|---|---|---|
| MineGAN | 102.4 | 132.1 | 87.4 |
| cGANTransfer | 89.6 | 61.6 | 70.3 |
| VPT (Non-Autoregressive) | **72.7** | 57.2 | **66.1** |
| VPT (Autoregressive) | 76.0 | **56.1** | 92.7 |

**cGM-2.** Tab. 9 presents the quantitative comparison for task cGM-2. F2DGAN achieves the best overall results on both Flowers and AFHQ datasets, with the lowest FID and highest LPIPS, confirming the benefit of feature-to-distribution alignment for balanced fidelity and diversity. WaveGAN and AGE also perform competitively, with AGE showing the strongest diversity on AFHQ. Earlier methods such as FIGR, MatchingGAN, and LoFGAN lag behind due to their limited ability to capture complex intra-class variation and maintain global coherence.

**cGM-3.** Tab. 10 provides the FID results for task cGM-3. The VPT framework achieves the best performance, especially in its non-autoregressive form, which records the lowest FID scores on C101 and DTD

and strong performance on Flowers. This validates the effectiveness of progressive visual prompting in cross-domain class-conditioned generation. cGANTransfer performs moderately well but lacks adaptive prompting mechanisms, while MineGAN shows the weakest performance due to limited fine-tuning efficiency under domain shifts.

Table 11: Quantitative comparisons of representative methods for task IGM evaluated on SinDDM dataset.

| Method | LPIPS (↑) | SIFID (↓) |
|---|---|---|
| SinGAN | 0.187 | 0.155 |
| ConSinGAN | 0.155 | 0.095 |
| GPNN | 0.107 | **0.054** |
| SinDDM | **0.219** | 0.343 |

Table 12: Quantitative comparisons of representative methods for task SGM evaluated on the DreamBooth dataset.

| Method | DINO (↑) | CLIP-I (↑) | CLIP-T (↑) | Tuning-free |
|---|---|---|---|---|
| Real Images | 0.774 | 0.885 | – | – |
| DreamBooth | **0.668** | **0.803** | **0.305** | ✗ |
| CustomDiffusion | 0.643 | 0.790 | **0.305** | ✗ |
| Textual Inversion | 0.569 | 0.780 | 0.255 | ✗ |
| BLIP-Diffusion | 0.594 | 0.779 | 0.300 | ✓ |
| ELITE | **0.621** | 0.771 | 0.293 | ✓ |
| MoMA | 0.618 | **0.803** | 0.348 | ✓ |

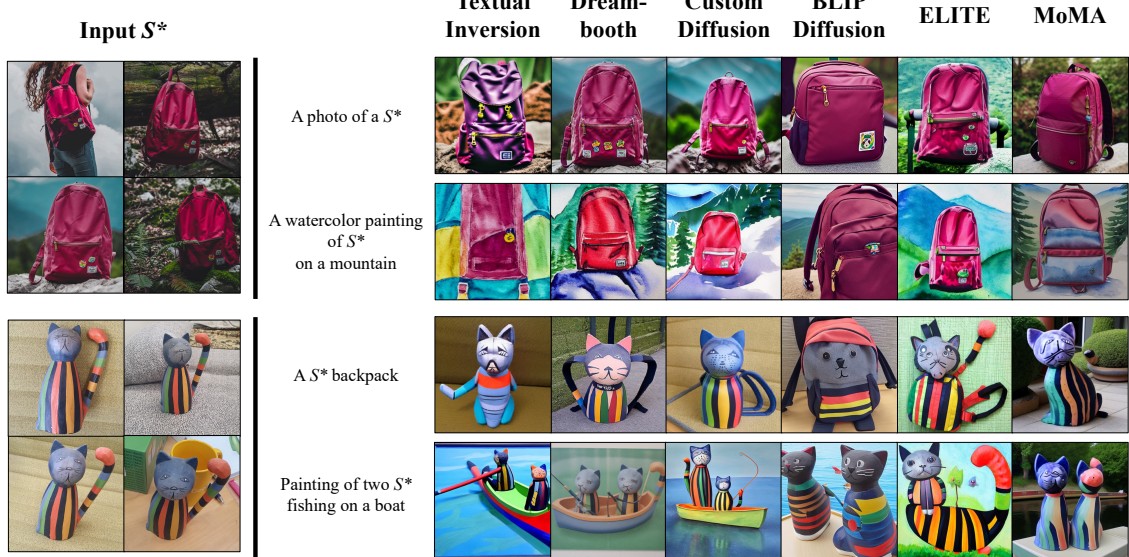

Figure 10: Qualitative comparisons of SGM methods.

**IGM.** Tab. 11 summarizes results for the image-specific generation task (IGM). SinDDM achieves the highest LPIPS score, indicating superior diversity, while GPNN attains the lowest SIFID, reflecting higher visual fidelity. The contrast between these methods highlights the trade-off between realism and diversity. SinGAN and ConSinGAN perform less effectively, producing less diverse and realistic outputs, suggesting weaker generalization compared to patch-based or diffusion-based approaches.

**SGM.** Among subject-driven generation methods, as shwon in Tab. 12 and Fig. 10, DreamBooth achieves the highest fidelity across metrics but requires fine-tuning, making it computationally heavy and less flexible. Custom Diffusion maintains comparable performance with reduced fine-tuning cost, while Textual Inversion trades off fidelity for efficiency. In contrast, tuning-free methods such as BLIP-Diffusion, ELITE, and MoMA achieve competitive performance without optimization, with MoMA performing best in balancing fidelity, diversity, and efficiency through multimodal modeling. This illustrates a clear evolution from high-cost tuning-based personalization toward more efficient yet expressive tuning-free paradigms.

## 7 Discussion

Here, we present an analysis the research landscape, discuss the research gap and future directions in GM-DC.

### 7.1 Analysis of the Research Landscape

In this work, we propose a **taxonomy of eight different tasks for GM-DC** ( Fig. 1, Tab. 2) based on the problem setups of GM-DC publications. Our investigation of the literature focusing on each task (Fig. 3) reveals that a significant portion of the works (up to 84%) concentrate on unconditional generation, either through training from scratch or adapting from a pre-trained model. Additionally, zero-shot unconditional generation is beginning to attract more attention. Similarly, adaptation for in-domain classes has garnered considerable interest for conditional generation. Meanwhile, conditional generation for out-of-domain classes via adaptation has not been explored adequately. Furthermore, subject-driven generation, which enables more control over content generation, is an emerging task. We anticipate increasing interest on this task as recent text-to-image generative models become more accessible.

We further present a **taxonomy of approaches for GM-DC** (Fig. 1, Tab. 3) as our another contribution. Our study reveals that transfer learning is a predominant solution for GM-DC, capable of tackling a large number of tasks (specifically, 5 out of 8 tasks, as indicated in Tab. 3 and Fig. 1), while effectively handling all data constraints including limited data, few-shot, and zero-shot. Moreover, ≈54% of the studies propose new methods based on transfer learning (Fig. 3). More than 12% of the studies propose methods based on other approaches that are compatible to transfer learning, *e.g.* data augmentation. These methods could be used with transfer learning-based methods to improve performance. The primary challenges in transfer learning are selection and preservation of source knowledge useful for generating high-quality and diverse target domain samples. Adaptation-aware approach (Zhao et al., 2022a; 2023a) could be a sound direction in this aspect where they consider both source and target domains (the adaptation process) for knowledge preservation. Language-guided approaches (Gal et al., 2022b; Ruiz et al., 2023; Kumari et al., 2023b; Liu et al., 2025a) are gaining increasing attention due to their ability to facilitate zero-shot generation through appropriate application of vision-language models during the transfer learning phase. Visual prompt tuning (Sohn et al., 2023) is a recent method, which guides the generation of target domain samples by generating visual tokens.

Data augmentation (Karras et al., 2020a; Tran et al., 2021; Wang et al., 2023d) remains a potent technique in GM-DC where it boosts performance under limited data by increasing coverage of the data distribution through various transformations. Multi-task objectives (Yang et al., 2021a; Tseng et al., 2021; Huang et al., 2022) which incorporate additional learning objectives are usually complementary to data augmentation. Various network architecture designs (Liu et al., 2021; Li et al., 2022b) that aim to prevent overfitting or preserve the feature maps are also shown to be significantly effective for GM-DC. Given that generative models tend to exhibit biases in capturing frequency components, enhancing the frequency awareness in these models is an emerging direction for GM-DC (Yang et al., 2022c). Meta-learning (Clouâtre & Demers, 2019) enables generative models to learn inter-task knowledge from seen classes, and then handle new generation tasks from unseen classes usually without fine-tuning (Gu et al., 2021; Hong et al., 2022a). Internal patch-distribution modeling (Shaham et al., 2019; Nikankin et al., 2023) effectively trains a generative model from scratch using a single reference image (scene) to produce novel scene compositions.

Regarding the types of generating models, our study shows that around 68% of the GM-DC works focus on GANs (Fig. 3). This preference can be attributed to the extensive research in GANs. Recently, there has been a growing interest in DMs (30%) and VAEs (2%), particularly VQ-VAE, driven by the success of DMs (Ramesh et al., 2022; Saharia et al., 2022) and transformer-based token prediction methods in generative modeling (Chang et al., 2022; Esser et al., 2021). We anticipate increasing attention directed toward DMs and VQ-VAEs. Furthermore, our survey reveals an interesting trend: around 71% of the works focus on addressing the challenging task of few-shot learning, while 26% concentrate on limited data scenarios. While only 3% of works address zero-shot learning, we expect growing interest due to recent advancements in vision-language models (Kwon & Ye, 2023; Li et al., 2023c).

### 7.2 Trends of Works across Categories

In this section, we discuss the trends of works across categories. Overall, as illustrated in Fig. 11, Transfer Learning has consolidated as the predominant direction, while most other categories remain relatively minor. Specifically, Transfer Learning accounts for 29% of works in 2021 and increasing to 77% in 2024. By

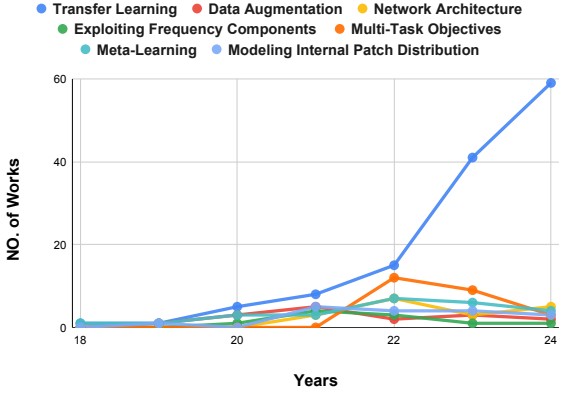

Figure 11: Trends of works in all categories.

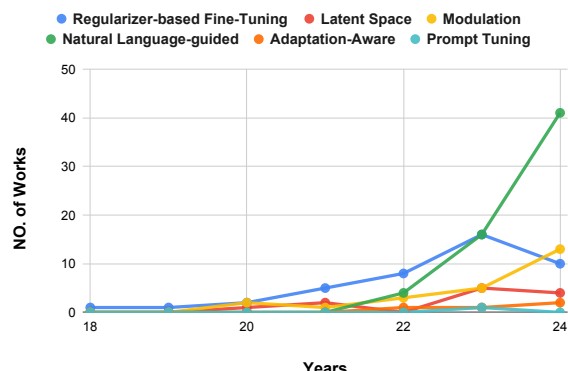

Figure 12: Trends of works in Transfer Learning.

contrast, Data Augmentation, Network Architecture, Exploiting Frequency Components, Meta-Learning, and Modeling Internal Patch Distribution all maintain relatively low percentages of works, generally below 15%. Only Multi-Task Objectives experienced a temporary peak, reaching 24% in 2022, but subsequently declined.

Given its rapid growth and central role, we provide a more detailed analysis of Transfer Learning. As shown in Fig. 12, within this category, we observe a gradual shift in emphasis over time. Earlier works primarily adopted Regularizer-based Fine-Tuning (40–63% between 2020–2021). More recently, Natural Language-guided approaches have increased substantially, reaching 59% in 2024. This development is closely related to the rise of multimodal foundation models such as CLIP and the introduction of new tasks like Subject-Driven Generation, which rely on language as a flexible control signal. Modulation methods also show an increasing trend, growing from 11% to 19%, supported by the adoption of parameter-efficient fine-tuning (PEFT) approaches such as LoRA. Prompt Tuning and Adaptation-Aware methods have also appeared in recent years, though their adoption remains limited.

## 7.3 Research Gap and Future Directions

While GM-DC has witnessed steady progress, several important research gaps remain that limit its advancement and broader applicability. Addressing these gaps requires moving beyond incremental improvements on existing setups and instead rethinking the foundations of data-constrained generative modeling. In particular, future work should consider leveraging foundation models more effectively, grounding zero-shot generation for evolving concepts, enabling transfer to distant target domains, developing holistic evaluation, and adopting data-centric strategies. The following subsections detail these directions.

### 7.3.1 Harnessing the power of foundation models

As previously discussed, transfer learning is a prominent and highly effective solution for GM-DC. Nevertheless, the majority of existing literature uses pre-trained StyleGAN2 (FFHQ) or BigGAN (ImageNet) networks as source models. A potential future direction for GM-DC is to explore the capabilities of foundation models (Bommasani et al., 2021), *i.e.* large models trained using massive amounts of data. In particular, text-image generation models including DALL·E-2 (Ramesh et al., 2022) (≈3.5B parameters), Imagen (Saharia et al., 2022) (≈4.6B parameters) and Stable Diffusion 3.5 (Esser et al., 2024) (≈8.1B parameters) encode knowledge regarding a wide range of concepts for high-quality, diverse image generation. Leveraging such foundation models for GM-DC is relatively under-explored.

### 7.3.2 Grounding zero-shot image generative capabilities

Recent studies have demonstrated the feasibility of zero-shot image generation for well-known concepts, *e.g.* "Tolkien Elf" (Gal et al., 2022b). However, grounding zero-shot image generation models to generate evolving/ new semantic concepts remains a relatively unexplored and challenging area. For instance, how to generate an image depicting "Mass for the Beginning of the Petrine Ministry of the Bishop of Rome," an event that occurred in May 2025, that related images may not be captured by existing models. This requires strategies that allow continual learning, semantic concept editing, and the incorporation of temporal contexts.

### 7.3.3 Knowledge transfer for distant/ remote target domains

Knowledge transfer has received significant attention in GM-DC research. Many works concentrate on utilizing pre-trained knowledge of a source domain to enhance learning in the target domain, as evident from the statistics in Fig. 1 and Fig. 3. However, we remark that exploring knowledge transfer for modeling target domains which are distant/ remote from the source domains still remains largely unexplored. This problem is challenging, as demonstrated in our experiment to transfer knowledge from Human Faces → Flowers (Fig. 6), which clearly demonstrates the complexity of the task. We urge more investigation in knowledge transfer for modeling distant/ remote target domains in GM-DC research.

### 7.3.4 Holistic evaluation of GM-DC

Evaluation of GM-DC presents multiple challenges including difficulties in estimating real data statistics under low-data regimes, lack of unified framework for human evaluation of GM-DC samples, and heavy reliance on particular (pre-trained) feature extractors to quantify the capabilities of GM-DC. In particular, developing holistic evaluation frameworks integrating both objective measurements and subjective judgements tailored for different tasks is essential for understanding GM-DC capabilities. Advancing holistic evaluation is important for GM-DC methods to be applied in a variety of real-world scenarios.

### 7.3.5 Data-centric approaches for GM-DC

We remark that data-centric approaches (Whang et al., 2023) for advancing GM-DC have been relatively overlooked in the literature. Majority of GM-DC methods focus on advancing training procedures based on a given set of training samples, but little attention has been put on how GM-DC performance may be affected by characteristics of the given training samples. Particularly, for GM-DC problems, where a domain is described using limited training samples, the characteristics of the samples can have noticeable impact on performance of GM-DC methods, as hinted in our analysis (see Fig. 7). We suggest greater emphasis on data collection, curation and pre-processing for GM-DC advancement.

## 7.4 Beyond Image Generation

Existing GM-DC works focus on image generation primarily. There are a few works to study other data types. Zhu et al. (2023) studies *3D shape generation* under few-shot target data (10-shot) utilizing pre-trained 3D generative models and optimization adaptation to retain the probability distributions of pairwise adapted samples. CLIP-Sculptor (Sanghi et al., 2023; Kim & Chun, 2023; Kim et al., 2023) leverages CLIP guidance for zero-shot *3D generation*. Wang et al. (2023a); Yang et al. (2024) studies few-shot *font generation* which aims to transfer the source domain style to the target domain. In particular, they introduce a content fusion module and a projected character loss to improve the quality of skeleton transfer for few-shot font generation. Careil et al. (2023) explores the problem of few-shot *semantic image generation* where the objective is to generate realistic images based on semantic segmentation maps. Their approach employs transfer learning on both GANs and DMs for few-shot semantic image synthesis. Couairon et al. (2023) further extend it in a zero-shot manner on Stable Diffusion (Rombach et al., 2022).

# 8 Conclusion

Generative Modeling under Data Constraints (GM-DC) is an important research area. This survey delves into this field by meticulously examining research papers in this area, encompassing different types of generative models including VAEs, GANs, and Diffusion Models. Drawing from this analysis, we identify several challenges encountered in GM-DC, including those related to training, data selection, and model evaluation. Moreover, we propose two taxonomies to categorize works related to GM-DC: a task taxonomy that identifies the variety of generation tasks, and an approach taxonomy that categorizes the extensive list of solutions for these tasks. We present a Sankey diagram to illuminate the interactions between different GM-DC tasks, approaches, and methods. Additionally, we present an organized review of existing GM-DC works and discuss research gaps and future research directions. Our aspiration is that this survey not only could offer valuable insights to researchers but also help spark further advancements in GM-DC.

### Ethics Statement

Generative models could be mis-used to disseminate mis- and disinformation due to their ability to generate realistic content. In particular, advanced generative models could be mis-used by malicious users to fabricate deepfake images, portraying individuals engaging in actions they never actually performed. Advances in GM-DC could exacerbate the situation as it becomes possible to generate realistic content with less data. We advocate for ethical and responsible usage of GM-DC methods and studying of mitigation techniques (Mirsky & Lee, 2021; Chandrasegaran et al., 2021; 2022b; Zhao et al., 2023b; Wen et al., 2023; Doloriel & Cheung, 2024).

### Acknowledgments

This research is supported by the National Research Foundation, Singapore under its AI Singapore Programmes (AISG Award No.: AISG2-TC-2022-007); The Agency for Science, Technology and Research (A*STAR) under its MTC Programmatic Funds (Grant No. M23L7b0021). This research is supported by the National Research Foundation, Singapore and Infocomm Media Development Authority under its Trust Tech Funding Initiative. Any opinions, findings and conclusions or recommendations expressed in this material are those of the author(s) and do not reflect the views of National Research Foundation, Singapore and Infocomm Media Development Authority.

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
