# OpenReview forum: "A Survey on Generative Modeling with Limited Data, Few Shots, and Zero Shot"
_TMLR — Accepted by TMLR_

### Review · Reviewer_PUNt · 2025-08-28

**Summary Of Contributions:**

This paper presents a method for generative modeling under data constraint (GM-DC), including limited-data, few-shot, and zero-shot settings. The authors analyzed the key challenges of GM-DC and proposed two taxonomies: including tasks and approaches. The authors also analyzed the research landscape and future directions.

Strengths:
 I am impressed by the detailed description, comprehensive analysis of this survey. I believe this survey can be used as a useful guide for both young researchers and also experts.
- The first comprehensive survey of GM-DC methods, covering more than 200 papers.
- Novel Sankey diagram to visualize the research landscape and relation between GM-DC papers
- Great organization of the paper, providing easy to access of the content

Weaknesses:
I don't see a major weakness of this paper. But I suggest the following:
- Add more visual results and comparison between papers
- Short summary of different categories of methods and the trend of the progress

**Audience:**

Yes

**Audience Explanation:**

The authors provided a comprehensive survey of GM-DC papers, covering machine learning and computer vision venues. Therefore, I believe the survey is of interest to the whole machine learning community, especially for those working on generative models.

**Claims And Evidence:**

Yes

**Claims Explanation:**

This survey paper is comprehensive and covers almost all papers of GM-DC as far as I am concerned. The authors provided good taxonomy of existing papers and categorized the methods and architectures.

**Requested Changes:**

The survey is almost complete. I suggest the authors add more visualization examples of different tasks, and add a discussion of the trends of each category.

---

> ### Author Response · Authors · 2025-10-31
> **Thank You for the Thoughtful and Supportive Feedback**
>
> We sincerely thank the reviewer for their time and valuable comments.
>
> $ $
>
>
> >**Q1**: Short summary of different categories of methods and the trend of the progress.
>
> **A1**:
> Thank you for your feedback. We kindly note that we have included a summary of categories in Sec. 5 of the submitted manuscript. We have added in section 7.2 of the revised manuscript for discussion of the trends of categories as follows:
>
> In this section, we discuss the trends of works across categories.  Overall, Transfer Learning has consolidated as the predominant direction, while most other categories remain relatively minor. Specifically, Transfer Learning accounts for 29% of works in 2021 and increasing to 77% in 2024. By contrast, Data Augmentation, Network Architecture, Exploiting Frequency Components, Meta-Learning, and Modeling Internal Patch Distribution all maintain relatively low percentages of works, generally below 15%. Only Multi-Task Objectives experienced a temporary peak, reaching 24% in 2022, but subsequently declined.
>
> Given its rapid growth and central role, we provide a more detailed analysis of Transfer Learning. Within this category, we observe a gradual shift in emphasis over time. Earlier works primarily adopted Regularizer-based Fine-Tuning (40–63% between 2020–2021). More recently, Natural Language-guided approaches have increased substantially, reaching 59% in 2024. This development is closely related to the rise of multimodal foundation models such as CLIP and the introduction of new tasks like Subject-Driven Generation, which rely on language as a flexible control signal. Modulation methods also show an increasing trend, growing from 11% to 19%, supported by the adoption of parameter-efficient fine-tuning (PEFT) approaches such as LoRA. Prompt Tuning and Adaptation-Aware methods have also appeared in recent years, though their adoption remains limited.
>
>
> $ $
>
>
> >**Q2**: Add more visualization examples of different tasks.
>
>
> **A2**:
> We thank the reviewer for this valuable suggestion. We incorporated additional visualization examples for several tasks in the Sec. 6.2 of revised manuscript. We chose these tasks because they either include a substantial number of works (e.g., uGM-2, SGM) or represent rapidly growing areas (uGM-3), which offer a representative view of current performance. The visual results were produced using either the official implementations (listed below) or faithful reproductions that strictly follow the methodological details described in the original papers.
>
> Available official codes:
>
> TGAN: https://github.com/yaxingwang/Transferring-GANs
>
> CDC: https://github.com/utkarshojha/few-shot-gan-adaptation
>
> DCL: https://github.com/yunqing-me/A-Closer-Look-at-FSIG
>
> AdAM: https://github.com/yunqing-me/AdAM
>
> RICK: https://github.com/yunqing-me/RICK
>
> NADA: https://github.com/rinongal/StyleGAN-nada
>
> IPL: https://github.com/Picsart-AI-Research/IPL-Zero-Shot-Generative-Model-Adaptation
>
> Textual Inversion: https://github.com/rinongal/textual_inversion
>
> Custom Diffusion: https://github.com/adobe-research/custom-diffusion
>
> BLIP Diffusion: https://github.com/salesforce/LAVIS/tree/main/projects/blip-diffusion
>
> ELITE: https://github.com/csyxwei/ELITE
>
> MoMA: https://moma-adapter.github.io/

---

### Review · Reviewer_KW2s · 2025-09-17

**Summary Of Contributions:**

This paper presents a survey of Generative Modeling under Data Constraint (GM-DC), reviewing over 230 papers across limited data, few-shot, and zero-shot scenarios. The main contributions include: (1) two novel taxonomies organizing GM-DC tasks into 8 categories and approaches into 7 categories, (2) a comprehensive analysis of challenges including overfitting, frequency bias, and incompatible knowledge transfer, (3) statistical analysis of publication trends and model distributions, (4) identification of research gaps and future directions, and (5) visual representation through Sankey diagrams showing task-approach-method relationships.

Strengths:
- Comprehensive scope covering multiple generative model types (GANs, diffusion models, VAEs)
- Well-structured organizational framework with clear taxonomies
- Valuable practical insights about domain proximity and knowledge transfer challenges
- Extensive bibliography demonstrating thorough literature coverage
- Good visual presentation and statistical analysis of the field

Weaknesses
- Limited technical depth and critical analysis beyond categorization
- Lack of systematic methodology for paper selection and validation of taxonomies
- Minimal empirical evaluation or quantitative comparison of approaches

**Audience:**

Yes

**Audience Explanation:**

Generative modeling under data constraints addresses practically important scenarios across many domains (medical imaging, artistic applications, etc.). The comprehensive organization of this rapidly evolving field provides a useful reference for researchers. The identification of research gaps and future directions also serves the community well.
However, interest may be tempered by:

- The descriptive rather than analytical nature of the work
- Limited actionable insights beyond organization
- The rapidly evolving nature of the field potentially dating some analyses quickly
- The heavy focus on GANs (68%) when diffusion models are increasingly prominent

**Broader Impact Concerns:**

The paper includes an appropriate ethics statement acknowledging potential misuse of generative models for creating deepfakes and misinformation. The authors advocate for responsible usage and studying mitigation techniques.

**Claims And Evidence:**

Yes

**Claims Explanation:**

The paper's main claims are generally supported as follows:

- The extensive bibliography (233 papers) and comparison with previous surveys support this claim reasonably well.
- While the taxonomies are more comprehensive than previous work, the novelty is incremental rather than fundamental. The task categorization follows fairly obvious distinctions (conditional vs unconditional, with/without pre-trained models).
- The discussion of challenges like domain proximity and incompatible knowledge transfer is supported by cited examples and experiments (Figures 6-7), though many of these challenges are already known in the community.

However, the evidence has several weaknesses:
- Limited empirical validation beyond a few illustrative experiments
- Arbitrary definitions (e.g., "limited data" as 50-5000 samples) without proper justification

**Requested Changes:**

- Provide clear inclusion/exclusion criteria for paper selection and systematic search protocol. Validate taxonomies through multiple reviewers or user studies.
- Include systematic comparison of approach effectiveness where possible, meta-analysis of performance metrics, and statistical analysis of which methods work best for different scenarios.
- Move beyond surface-level descriptions to analyze why certain approaches succeed or fail, identify fundamental limitations, and provide deeper insights into method design principles.
- Provide principled justification for data constraint ranges (limited/few-shot/zero-shot) based on literature consensus or empirical analysis rather than arbitrary thresholds.

---

> ### Author Response · Authors · 2025-10-31
> **Thank You for the Thoughtful and Supportive Feedback. [Reponse for Reviewer KW2s] Part 1/4**
>
> We sincerely thank the reviewer for their time and valuable comments.
>
> $ $
>
> >**Q1**: Limited technical depth and critical analysis beyond categorization. Move beyond descriptions to analyze why certain approaches succeed or fail, identify fundamental limitations, and provide deeper insights into method design principles.
>
> A1: Thank you for the insightful comment. We apologise that, due to the extensive technical coverage in the paper, some of our critical analyses may have been overlooked. For example, in Section 5, we provide our critical analyses. For Data Augmentation, we note that “a major challenge of these approaches is augmentation leakage, where the generator learns the augmented distribution, such as generating rotated or noisy samples,” while for Multi-Task Objective, we highlight that “the efficient integration of the new objective with the generative learning objective can be challenging under data constraints.” Similar critical analyses are provided for other categories throughout the section.
>
> **Based on the reviewer’s feedback, we have added a dedicated section that summarizes, for each class of approaches, the key factors contributing to their success or failure, their fundamental limitations, and the deeper design principles that emerge from these observations. Comprehensive discussion for all approaches have been added in Sec. 6.1 of the revised manuscript.** Below is one part of our newly-added text in the revised manuscript:
>
>
> $ $
>
> **Transfer Learning.**
>
> *Regularizer-based fine-tuning* remains one of the most widely used strategies for leveraging transfer learning in GM-DC. These approaches tend to succeed when the source and target domains are closely related, because the regularizer constrains optimization to preserve useful semantic and structural priors for stabilizing adaptation. However, they often fail when the domain gap becomes large: the same constraint that stabilizes learning in similar domains can over-restrict adaptation in dissimilar ones, causing the generator to retain source-specific biases that are incompatible with the target distribution and hindering the learning of new features. This exposes a fundamental limitation: regularization strength inherently trades off between knowledge preservation and domain adaptability. Determining the optimal level of regularization requires estimating the transfer distance between domains, which is particularly difficult under limited data. A deeper design principle arising from this insight is that successful GM-DC methods should employ adaptive or data-aware regularization schemes that dynamically balance the reuse of transferable priors with the flexibility needed to capture novel characteristics of the target domain.
>
> *Latent-space* methods provide a lightweight and parameter-efficient strategy for GM-DC. These approaches succeed when adaptation occurs within a compact and well-structured latent representation, allowing the model to reuse the pretrained backbone while updating only a small set of parameters. However, they often fail when the semantic gap between source and target domains becomes large. In such cases, the latent space may lack sufficient expressiveness to represent new concepts, leading to misalignment or entanglement of factors that distort the generated outputs. This highlights a fundamental limitation: adaptation performance is highly dependent on the quality of the pretrained latent representation. A deeper design principle emerging from this insight is that effective latent-space adaptation should include mechanisms for latent disentanglement and representation expansion, maintaining efficiency while enabling the model to flexibly capture novel characteristics of the target domain.
>
> *Modulation-based* approaches have emerged as a practical middle ground for GM-DC. They succeed because they inject new information through lightweight modulation layers or parameters, effectively “writing” new concepts without overwriting pretrained base knowledge. However, they often fail when the required transformation between source and target domains is large or complex, as the limited modulation capacity may underfit distant domain adaptations. Performance also depends strongly on the granularity and location of modulation: inappropriate layer selection can lead to insufficient adaptation or degradation of important pretrained representations. This highlights a fundamental limitation: the method’s success is tied to the hierarchical representation of the specific model and is sensitive to how modulation interacts with that hierarchy. The deeper design principle is that effective modulation-based adaptation must align modulation granularity and placement with the model’s semantic representation hierarchy, ensuring sufficient modulation capacity to inject new concepts effectively while preserving pretrained knowledge.
>
> $ $
>
> (continue in Part 2)

---

> ### Author Response · Authors · 2025-10-31
> **[Reponse for Reviewer KW2s] Part 2/4**
>
> (continue response to Q1 in Part 1)
>
> *Adaptation-aware* methods provide a systematic and data-driven mechanism for selecting and preserving transferable knowledge in GM-DC. They succeed because they explicitly measure the importance or compatibility of model components, such as convolutional kernels or attention heads, often using criteria like Fisher Information. By freezing or pruning components of different importance, these methods retain useful pretrained knowledge while reducing negative transfer from irrelevant or harmful features. However, they often fail when the importance estimation are unreliable—especially under limited data—leading to the removal of components that are crucial for representing target-domain knowledge. This highlights a fundamental limitation: the quality of adaptation depends on the accuracy and stability of importance estimation under data scarcity. The deeper design principle is that effective adaptation-aware strategies should combine robust importance estimation with dynamic update mechanisms, allowing the model to preserve and refine relevant components during adaptation.
>
> *Natural language-guided* approaches have shown strong performance in zero-shot generative modeling and personalization, especially in diffusion-based models where textual prompts enable flexible concept transfer with minimal training data. They succeed because they leverage the rich alignment between text and visual representations learned by large multimodal foundation models such as CLIP. However, these methods often fail when the textual encoder or base model has inadequate instruction-following ability or weak language grounding, leading to unstable or inaccurate generations. They can also amplify or transfer undesired semantic biases inherited from the pretrained text–image space. This highlights a fundamental limitation: performance depends heavily on the alignment quality and bias characteristics of the underlying multimodal model. The deeper design principle is that effective language-guided transfer requires robust semantic alignment and a reliable, bias-aware base model to ensure that linguistic guidance remains precise, consistent, and faithful to the intended target concept.
>
> *Visual prompt-tuning* provides a highly parameter-efficient strategy for GM-DC and effectively mitigates catastrophic forgetting by keeping the pretrained backbone frozen. These methods succeed when the base model already encodes concepts and structures that overlap with the target domain, allowing lightweight visual tokens or prompts to guide generation without extensive retraining. However, they often fail under large domain shifts or fine-grained structural variations, where the limited capacity of guidance tokens cannot adequately steer the generation process toward new or complex target features. This highlights a fundamental limitation: visual prompts operate within the representational bounds of the pretrained model and have limited ability to extend its conceptual space. The deeper design principle is that effective visual prompt-tuning should incorporate mechanisms that enhance steering capacity while preserving the efficiency and stability benefits of frozen-backbone adaptation.
>
>
> $ $
>
> >**Q2**: The heavy focus on GANs (68%) when diffusion models are increasingly prominent.
>
> **A2**:
> We respectfully clarify that the reported 68% reflects the objective distribution of model families in the surveyed literature rather than a preference in our coverage. Earlier works (especially before 2022) are predominantly GAN-based, which raises their aggregate share. Our manuscript follows this distribution while also highlighting recent developments in diffusion models. For example, in the Natural Language-Guided approaches (Sec. 5.1.5) – which is the largest sub-category by number of works in our paper – the text spans approximately two pages, of which about 1.5 pages focus on diffusion-based methods. This emphasis reflects the rapid growth of Subject-Driven Generation, a newly introduced task where methods are typically built on Stable Diffusion.

---

> ### Author Response · Authors · 2025-10-31
> **[Reponse for Reviewer KW2s] Part 3/4**
>
> >**Q3**: Systematic comparison of approach effectiveness, meta-analysis of performance metrics, and statistical analysis of which methods work best for different scenarios.
>
> **A3**:
> We thank the reviewer for this valuable suggestion. We provide a comprehensive quantitative comparison of representative methods for **all GM-DC tasks** in Sec. 6.2 of the revised manuscript. Below is one part of our newly-included text in the revised manuscript:
>
> $ $
>
> Table R1. Quantitative comparisons of representative methods for task uGM-2. The source model is StyleGAN-2 pretrained on FFHQ for both settings.
> | Method  | Target Domain | | | |
> |---|:---|:---|:---|:---|
> | | FFHQ-Baby | | AFHQ-Cat | |
> | | FID (↓) | Intra-LPIPS (↑) | FID (↓) | Intra-LPIPS (↑) |
> | TGAN | 101.58 | 0.517  | 64.68  | 0.490 |
> | FreezeD | 96.25 | 0.518 | 63.60 | 0.492 |
> | EWC | 79.93 | 0.521  | 74.61  | 0.587  |
> | CDC | 69.13 | 0.578  | 176.21  | **0.629** |
> | DCL | 56.48 | 0.580 | 156.82 | 0.616 |
> | AdAM | 48.83  | 0.590  | 58.07 | 0.557 |
> | RICK | **39.39** | **0.608** | **53.27** | 0.569 |
>
> $ $
>
> **uGM-2**. The results in Tab. R1 shows a clear progression from early transfer-based approaches to more adaptation-aware methods. RICK achieves the best overall performance on both target domains, with the lowest FID and highest Intra-LPIPS, indicating superior adaptation and diversity. Earlier approaches such as TGAN and FreezeD perform worse because they rely heavily on fixed discriminator transfer and lack mechanisms to preserve domain-specific knowledge during adaptation. EWC stabilizes learning by constraining parameter changes but can limit adaptation when domain gaps are large, as reflected in its uneven FID results. CDC and DCL improve diversity through correspondence and contrastive learning but are sensitive to similarity metrics and training stability. AdAM bridges these limitations through adaptive module selection, and RICK refines this by prioritizing useful kernels during adaptation, preventing negative transfer and yielding the best trade-off between fidelity and diversity.
>
>
>
>
>
> $ $
>
>
> >**Q4**: Arbitrary definitions (e.g., "limited data" as 50-5000 samples) without proper justification
>
> **A4**
> Thank you for your comment, and apologies that the definition of the data constraint scenarios was not clearly explained. As mentioned in Section 3.3, these definitions are derived from the experimental setups reported in existing research. We conducted a careful empirical analysis of prior works and their experimental configurations to propose these three scenarios. Below is the detailed procedure we followed in defining them:
>  - We began by reviewing prominent and widely cited works in this area, including TGAN, StyleGAN2-ADA, EWC, CDC, FastGAN, DreamBooth, NADA, LofGAN, SinGAN, and others.
>  - For each paper, we carefully examined all experiments and the number of samples used, as reported in their results. This analysis revealed consistent patterns in the data availability assumed across studies.
>  - These patterns suggest three main clusters of data availability setups commonly used in the literature: Limited Data (50–5000 samples), Few-Shot (1–50 samples), and Zero-Shot (no samples from the target domain).
>
> For example, consider the Limited Data scenario. Two of the most well-known works addressing generative learning under data constraints are StyleGAN2-ADA and FastGAN. In ADA, the number of available samples for small datasets (Figure 11) is 1.3k for MetFaces, 1.9k for BreCaHAD, and 5k for AFHQ-Cat/Dog/Wild. Similarly, FastGAN uses between 60 and 1000 samples across 12 datasets (Tables 2 and 3). These works pioneered the setup for small data regimes, and many subsequent studies—often using these as baselines—adopt similar sample ranges (50–5k). This consistent pattern motivated our definition of the Limited Data scenario as 50–5k samples.
>
> A similar trend is observed among works that assume access to only 1–50 samples (e.g., EWC, CDC, AdAM, JoJoGAN, etc.), which we categorize as the Few-Shot scenario. Finally, a newer line of research (e.g., NADA, IPL, AIR) assumes no samples from the target domain, which we classify as the Zero-Shot scenario.
>
> We have included these details in the revised manuscript to clarify the empirical basis and rationale behind our proposed data constraint definitions.
>
> $ $

---

> ### Author Response · Authors · 2025-10-31
> **[Reponse for Reviewer KW2s] Part 4/4**
>
> >**Q5**: Criteria for paper selection and systematic search protocol. Validate taxonomies through multiple reviewers or user studies.
>
> **A5**:
> We thank the Reviewer for the comment. In what follows, we provide detailed information regarding our search strategy and study selection criteria (also included in the updated manuscript):
>
> $ $
>
> **Search Strategy.** Our search strategy is based on an extensive list of keywords related to GM-DC. We began with a set of seed keywords derived from well-known papers in the field and selected several representative works. We then carefully examined these papers and their related work, expanding our initial list to form the main set of keywords used to search for relevant research papers for the GM-DC survey. The final list of 17 keywords is as follows:
>
> Transfer learning for generative models; Transfer learning in GANs/ DMs (2 keywords); Fine-tuning generative models; Fine-tuning GANs/ DMs/ VAEs (3 keywords); Generative model adaptation; Adaptation of GANs/ DMs/ VAEs (4 keywords); Few-shot image generation; Few-shot adaptation of generative models; Data-efficient generative modeling; Training GANs / DMs with limited data (2 keywords).
>
> To ensure comprehensive coverage, we searched for these keywords across seven major repositories commonly used for machine learning and computer vision research:
> Google Scholar,
> OpenReview,
> CVF Open Access,
> IEEE Xplore Digital Library,
> ACM Digital Library,
> ScienceDirect (Elsevier), and
> SpringerLink.
>
> $ $
>
> **Study Selection Criteria.** Among the collected papers, we applied three main criteria for inclusion, focusing on the problem setup, modality, and task type:
> - Problem Setup: We examined all experimental results (both qualitative and quantitative) in each paper to ensure that at least one of them satisfies a data-constrained setup (0~5000 availabe samples), i.e., the proposed approach can operate under such conditions.
>  - Modality: Our survey focuses exclusively on the image modality; therefore, we discarded studies dealing with other modalities such as sculpture, 3D mesh, or point cloud data.
>  - Task Type: We strictly focused on the image generation task and excluded papers addressing other tasks (e.g., image editing).\footnote{Note that some recent works, such as DreamBooth, can handle both image generation and image editing tasks. Since these methods support image generation and not only editing, they are included in our study.}
>
> $ $
>
> **Second round of Search.** After the initial selection, we conducted a second round of review by carefully examining the related work and experimental setups of the selected papers to ensure that all relevant studies were included. In this step, we also added some works that did not explicitly mention the keywords in their title or abstract but nonetheless satisfied all inclusion criteria.

---

### Review · Reviewer_Lsqt · 2025-10-17

**Summary Of Contributions:**

This paper presents a comprehensive and needed survey of the field of Generative Modeling under Data Constraints (GM-DC), covering the sub-topics of limited-data, few-shot, and zero-shot learning. The authors have reviewed an extensive body of literature (over 230 papers) to provide a structured and insightful overview of this rapidly evolving area.

**Key Strengths**:
- The primary contribution is the introduction of novel taxonomies that categorize GM-DC tasks and methodological approaches. This provides an very interesting organizational framework that clarifies the relationships between different research threads.
- The paper does a good job in identifying and analyzing core challenges, such as overfitting, frequency bias, and incompatible knowledge transfer, supporting its claims with experimental illustrations.
- The use of good visualization tools (Sankey diagram) to map the landscape of tasks, approaches, and methods is a significant strength, offering a clear and intuitive visualization of the field's structure.
- The survey concludes with a nice discussion of open problems and future research directions, providing a clear roadmap for researchers.

**Weaknesses:**
- The paper's primary weakness is its density. While comprehensive, the sheer volume of work surveyed, particularly in Section 5, can be overwhelming for readers not already familiar with the domain. The structure of this section could be improved for better readability.

**Audience:**

Yes

**Audience Explanation:**

The topic of generative modeling, especially with limited data, is of high interest to a broad segment of the machine learning community. This survey would be valuable to several groups within the TMLR audience:
- Researchers new to the field will find this paper to be a good entry point, providing a structured overview of the key concepts, challenges, and pointers to existing methods.
- Experienced researchers in generative modeling will benefit from the comprehensive literature review, and the insightful discussion of open problems.
- Practitioners working in domains where data is scarce may find the survey a useful guide to the state-of-the-art methods applicable to their problems.

**Broader Impact Concerns:**

The paper includes an "Ethics Statement" that appropriately acknowledges the potential for misuse of generative models, particularly in the creation of deepfakes and the spread of disinformation.

**Claims And Evidence:**

Yes

**Claims Explanation:**

The claims made in the paper are well-supported by evidence. The central contributions (the taxonomies and the analysis of the field) are substantiated by a thorough review of over 230 relevant papers. The statistical claims regarding trends in the field (e.g., the prevalence of transfer learning, the rise of diffusion models) are backed by nice data visualizations. Furthermore, the authors do not just state the key challenges; they provide concrete examples and even include experimental results (Figs. 5, 6, 7) to demonstrate the difficulties of modeling distant domains and the impact of sample selection. The logical flow is sound, and the conclusions drawn in each section seem to be a direct consequences of the evidence presented.

**Requested Changes:**

The paper is of high quality, but some structural changes and minor edits could significantly improve its clarity and readability.

A few suggestions to strengthen the work.

- While the core concepts ("Limited data," "Few-Shot," "Zero-Shot") are defined in Section 3.2, it would be beneficial to briefly introduce them in the main introduction to provide immediate context for the reader.
- The current "Related Work" section primarily compares this survey to others. It could be strengthened by providing a more general connection to the full spectrum of generative modeling methods, giving readers a broader context for where GM-DC fits in.
- Section 5, "Comprehensive Review," is quite dense. The authors should consider splitting it into multiple sections with more descriptive titles to improve navigation. Similarly, Section 5.1.1 ("Regularizer-based Fine-Tuning") is very long; breaking it into smaller, titled paragraphs would enhance readability.
- Section 6.2 Introduction: It would be helpful to add an introductory paragraph to Section 6.2 to frame the discussion on future directions before diving into the specific sub-points.
- Table 3 Organization: Table 3 is very packed with information. The authors should consider splitting it into multiple, more manageable tables (e.g., one table per major approach) to improve its usability as a reference.

Minor Edits:
- Citation Clarity: Some citations could be more precise. For example, the citation for Diffusion Models (page 8, Kingma et al. 2021) could be confusing. For foundational topics, it is helpful to cite the seminal work(s) or note the key contributing papers.
- Grammar: On page 3: "The more recent text-to-image generative model is trained..." should be "More recent text-to-image generative models are trained...".
- Acronyms: A final check to ensure all acronyms are defined on first use would be beneficial.

---

> ### Author Response · Authors · 2025-10-31
> **Thank You for the Thoughtful and Supportive Feedback**
>
> We sincerely thank the reviewer for their time and valuable comments.
>
> $ $
>
> >**Q1**: Introduce core concepts ("Limited data," "Few-Shot," "Zero-Shot") in the main introduction and other minor edits.
>
> **A1**:
> Thank you for your feedback. We have done a final check and incorporated the suggested edits in the final revised manuscript.
>
> $ $
>
> >**Q2**: The current "Related Work" section primarily compares this survey to others. It could be strengthened by providing a more general connection to the full spectrum of generative modeling methods, giving readers a broader context for where GM-DC fits in.
>
> **A2**:
> Thank you. We added a paragraph in Sec. 2 Related Work that provides broader generative modeling context for GM-DC and includes representative survey citations. For convenience, the added text is as follows:
>
> **Generative Modeling.**
> A broad line of research in machine learning has focused on developing powerful generative models that can synthesize high-quality and diverse data samples. Key paradigms include variational autoencoders, generative adversarial models, flow-based Models, diffusion models. Several surveys provide comprehensive overviews of these approaches and their advancements. These works highlight the remarkable progress in fidelity, diversity, and controllability. However, they typically assume access to abundant training data, and their performance can degrade significantly when trained with limited samples, manifesting in problems such as overfitting, mode collapse, or reduced generalization. Consequently, while these surveys form the foundation of modern generative modeling, they do not directly address the unique challenges of learning under data-constrained settings, which are the primary focus of this work.
>
> $ $
>
> >**Q3**: Section 5, "Comprehensive Review," is quite dense. Consider splitting it into multiple sections with more descriptive titles to improve navigation. Similarly, Section 5.1.1 ("Regularizer-based Fine-Tuning") is very long; breaking it into smaller, titled paragraphs would enhance readability.
>
> **A3**:
> Thank you for the suggestion. While we keep Sec. 5 as a single “Comprehensive Review” to preserve continuity, we improved navigation by adding concise bold lead-in phrases and reorganizing paragraphs so that each paragraph group works by a **single underlying idea** (i.e., a fine-grained category). We believe these lead-ins label the idea explicitly, effectively splitting long sections into smaller, titled units that clarify the rationale behind the grouping and make scanning easier.
>
> $ $
>
> >**Q4**: Section 6.2 Introduction: It would be helpful to add an introductory paragraph to Section 6.2 to frame the discussion on future directions before diving into the specific sub-points.
>
> **A4**:
> Thank you for the feedback. We have added an introductory paragraph in the revised manuscript as follows:
>
> While GM-DC has witnessed steady progress, several important research gaps remain that limit its advancement and broader applicability. Addressing these gaps requires moving beyond incremental improvements on existing setups and instead rethinking the foundations of data-constrained generative modeling. In particular, future work should consider leveraging foundation models more effectively, grounding zero-shot generation for evolving concepts, enabling transfer to distant target domains, developing holistic evaluation, and adopting data-centric strategies. The following subsections detail these directions.
>
> $ $
>
>
> >**Q5**: Table 3 Organization: Tab. 3 is very packed with information. Consider splitting it into multiple, more manageable tables (e.g., one table per major approach) to improve its usability as a reference.
>
> **A5**:
> Thank you for your valuable suggestion. Table 3 is intended as an overview of the proposed taxonomy for approaches as well as the comprehensive review in Sec. 5, so we believe keeping it as one summarized table best preserves comparability at a glance.
> That said, we recognize the readability concern. In the revision, we have adjusted the citation text color to reduce visual crowding so that the main content receives more attention.

---

### Decision · Action_Editor_6MZg · 2025-12-12

**Recommendation:** Accept as is

**Additional Comments:**

This paper presents an extremely thorough survey of generative models under data constraints. The authors also provide a useful taxonomy of the discussed methods, along with challenges in the area. The paper is clearly a high quality survey on a topic of increasing relevance that will be useful for the community, and I thus recommend acceptance and a survey certification.

**Audience:**

Yes

**Audience Explanation:**

Yes, reviewers unanimously agree.

**Claims And Evidence:**

Yes

**Claims Explanation:**

Yes, reviewers unanimously agree.